# FinEvo: From Isolated Backtests to Ecological Market Games for Multi-Agent Financial Strategy Evolution

## Abstract

Conventional financial strategy evaluation relies on isolated backtests in static environments. Such evaluations assess each policy independently, overlook correlations and interactions, and fail to explain why strategies ultimately persist or vanish in evolving markets. We shift to an ecological perspective, where trading strategies are modeled as adaptive agents that interact and learn within a shared market. Instead of proposing a new strategy, we present **FinEvo**, an ecological game formalism for studying the evolutionary dynamics of multi-agent financial strategies. At the individual level, heterogeneous ML-based traders—rule-based, deep learning, reinforcement learning, and large language model (LLM) agents—adapt using signals such as historical prices and external news. At the population level, strategy distributions evolve through three designed mechanisms—selection, innovation, and environmental perturbation—capturing the dynamic forces of real markets. Together, these two layers of adaptation link evolutionary game theory with modern learning dynamics, providing a principled environment for studying strategic behavior. Experiments with external shocks and real-world news streams show that FinEvo is both stable for reproducibility and expressive in revealing context-dependent outcomes. Strategies may dominate, collapse, or form coalitions depending on their competitors—patterns invisible to static backtests. By reframing strategy evaluation as an ecological game formalism, FinEvo provides a unified, mechanism-level protocol for analyzing robustness, adaptation, and emergent dynamics in multi-agent financial markets, and may offer a means to explore the potential impact of macroeconomic policies and financial regulations on price evolution and equilibrium.

## 1 Introduction

Financial markets are among the most complex and adaptive systems, where outcomes depend not only on external signals but also on the collective behavior of participants. Most evaluation paradigms abstract away such interactions. Deep learning emphasizes predictive accuracy, from attention-based architectures such as TimeMixer (Wang et al., 2024) to foundation-style models like Chronos (Ansari et al., 2024). Reinforcement learning extends to sequential decision-making, with recent advances in *offline Reinforcement learning(RL)* for data-efficient policy learning (Kumar et al., 2020) and *risk-sensitive RL* for robustness under uncertainty (Duan et al., 2016). Meanwhile, LLM-based agents extend trading logic with reasoning and tool use: FinCon frames trading as structured financial reasoning with CVaR (Yu et al., 2024), FinGPT adapts foundation models for forecasting and decision support(Liu et al., 2023), and TradingAgents explore general-purpose planning and execution capabilities (Xiao et al., 2024).

Yet most existing approaches optimize in isolation, neglecting the ecological interdependencies that shape real-world market performance. Rather than proposing new trading strategies or predictive models, our goal is to develop a principled ecological game formalism—serving as a "financial wind tunnel" to stress-test systemic risks and market shocks within a controlled environment(see Appendix H).

Simulation infrastructures such as ABIDES (Byrd et al., 2020) and Mesa (Masad et al., 2015) support market microstructure and agent-based experimentation, yet they operate primarily as execution environments without modeling population-level adaptation or long-horizon strategic dynamics. PyMarketSim (Mascioli et al., 2024) further offers high-fidelity order-book execution for deep RL agents, but remains centered on single-agent optimization rather than collective ecological behavior. TraderTalk (Vidler & Walsh, 2024) explores LLM-driven behavioral trading agents, yet focuses on bilateral interactions without a framework for systemic market evaluation.

Beyond these platforms, GPU-accelerated limit order book simulators such as JAX-LOB (Frey et al., 2023) and fast agent-based frameworks for trading latency and RL (Belcak et al., 2021) advance high-fidelity training on historical order-flow data, while ecosystem-style models such as HFTE (Mahdavi-Damghani, 2017) examine multi-species interactions among high-frequency strategies. However, none of these lines of work offer **a unified evolutionary game-theoretic formulation** for characterizing adaptation, population flows, and systemic impact in heterogeneous trading ecologies—a gap that FinEvo focuses on (see Table 6).

To realize this vision, we introduce **FinEvo**, an ecological game formalism for studying the evolutionary dynamics of multi-agent financial strategies. The agent population spans rule-based, deep learning, reinforcement learning, and large language model (LLM) traders, each operating under parameterized policies and capital constraints, with prices endogenously determined by continuous double-auction clearing. Inspired by the replicator–mutator equation (Nowak, 2006), FinEvo formalizes population dynamics through selection, innovation, and environmental perturbation, capturing adaptation, extinction, and ecosystem-level phenomena such as dominance cycles and regime shifts, enabling principled evaluation of robustness and adaptation under realistic market interaction. We provide a more detailed review of related work in Appendix A.

In summary, our contributions are:

1) We introduce **FinEvo**, an ecological game formalism that integrates heterogeneous traders with evolutionary population dynamics, formalized by the **FinEvo SDE**. Its mechanism decomposition unifies selection, innovation, and environmental perturbation, elevating evaluation from instance-level returns to *mechanism-level* behavior. 2) Under mild conditions, we establish minimal theoretical guarantees for the FinEvo SDE—simplex invariance, positivity, and existence/uniqueness—and derive a macro-level variance decomposition that attributes system volatility to selection, innovation, and perturbation. 3) We demonstrate through experiments with shocks and real-world signals that FinEvo reveals context-dependent performance, robustness, and emergent coalition dynamics that static backtests cannot capture.

Together, these contributions establish FinEvo as a principled basis for studying robustness and adaptation in multi-agent financial markets.

## 2 ECOLOGICAL GAMES: PROBLEM FORMULATION AND EVOLUTIONARY DYNAMICS

We model financial strategy evaluation as an *ecological game*, where a population of $N$ heterogeneous trader agents interact in a shared, dynamically evolving market. Each agent follows a distinct policy and adapts through feedback between market outcomes and external signals. This formulation allows us to study both the performance of individual strategies and the collective dynamics of selection, innovation, and perturbation.

**Agent behavior and market feedback.** At each time step $t$, agent $i$ selects an action $a_{i,t} \sim \pi_{k(i)}(\mathcal{I}_t)$, where $\pi_{k(i)}$ is the policy of type $k$, conditioned on recent prices $\{P_{t-\tau}, \ldots, P_t\}$ and exogenous signals $E_t$. And the action space includes *market buy, market sell, limit buy, limit sell, and hold*.

The market aggregates all actions and shocks $\xi_t$ into the next price $P_{t+1}$, with the detailed matching mechanism given in Appendix G.6,

$$P_{t+1} = F\big(P_t, \{a_{i,t}\}_{i=1}^N, \xi_t\big).$$

Executed trades then update portfolio holdings. Specifically, let $C_{i,t}$ denote the cash holdings and $S_{i,t}$ the assets holdings of agent $i$ at time $t$, while $P_t$ denotes the market-clearing price. $C_{i,t+1} =$

$C_{i,t} - a_{i,t}^{\text{buy}} P_t + a_{i,t}^{\text{sell}} P_t$, and $S_{i,t+1} = S_{i,t} + a_{i,t}^{\text{buy}} - a_{i,t}^{\text{sell}}$. For strategy $\pi_k$, the realized payoff over $[t, t+1]$ is $\hat{f}_{k,t} = \frac{1}{N_{k,t}} \sum_{i:\pi_k(i)=\pi_k} \left[ C_{i,t+1} + S_{i,t+1} P_{t+1} - (C_{i,t} + S_{i,t} P_t) \right]$, representing the average incremental wealth change of its adopters.

**Individual adaptation.** Each strategy maintains an internal parameter state $\Psi_{k,t}$ that governs how it maps information $\mathcal{I}_t$ into actions. This state evolves through a generic adaptation operator

$$\Psi_{k,t+1} = \mathcal{A}\big(\Psi_{k,t}, \hat{f}_{k,t}, E_t\big),$$

where $\mathcal{A}$ abstracts diverse mechanisms of learning and adjustment. In practice, this update may correspond to temporal-difference or policy-gradient rules in reinforcement learning, gradient-based parameter tuning in deep learning models, in-context adaptation in large language models, or heuristic adjustment in rule-based strategies.

Beyond reacting to immediate rewards, traders typically attempt to estimate a *long-term value* of their strategies. To capture this, we associate with each $\pi_k$ a forward-looking value process $V_k(t)$, modeled as $dV_k(t) = \lambda_k \big( f_k(E_t) - V_k(t) \big) dt + \nu_k dB_k(t)$, where $f_k(E_t)$ denotes an environment-conditioned target payoff, $\lambda_k > 0$ controls the adjustment speed, and $\nu_k dB_k(t)$ introduces stochastic exploration. This Ornstein–Uhlenbeck process represents convergence of long-term value estimates toward external signals while retaining variability due to uncertainty, serving as a forward-looking anchor that guides subsequent population-level dynamics.

**Population evolution (FinEvo SDE).** Let $X_t = \big(x_1(t), \ldots, x_K(t)\big)^\top \in \Delta^{K-1}$ denote the population distribution over $K$ strategies, with $\sum_k x_k(t) = 1$. The evolution of $X_t$ can be abstractly represented by an operator

$$x_{t+1} = \mathcal{G}\big(x_t, V(t), m_t, \xi_t\big),$$

where $V(t) = (V_1(t), \ldots, V_K(t))^\top$ collects the forward-looking values estimated at the individual level, $m_t$ denotes the innovation distribution, and $\xi_t$ represents stochastic perturbations.

Concretely, we formulate $\mathcal{G}$ as a continuous-time stochastic differential equation, the **FinEvo SDE**, which integrates the three designed mechanisms—selection, innovation, and perturbation—into a unified dynamic:

$$dX_t = b(X_t, t)\, dt + \Sigma(X_t, t)\, dW_t$$

$$= \underbrace{\beta \mathrm{diag}(X_t) \big( V(t) - \bar{V}(t)\mathbf{1} \big)}_{\textbf{Selection}} dt + \underbrace{\mu \, (m_t - X_t)}_{\textbf{Innovation}} dt + \underbrace{\gamma \, \mathrm{diag}(X_t)\, \sigma\, P(X_t)\, dW_t}_{\textbf{Perturbation}},$$

where $\bar{V}(t) = X_t^\top V(t)$ is the population mean payoff, $P(X_t) = I - \mathbf{1}X_t^\top$ projects perturbations onto the simplex tangent space, $\sigma = \mathrm{diag}(\sigma_1, \ldots, \sigma_K)$, $\gamma$ denotes the global perturbation scaling factor and $W_t$ is a $K$-dimensional Brownian motion. The target distribution $m^t \sim \mathrm{Dir}(\alpha_1, \ldots, \alpha_K)$ models social influence and random experimentation, ensuring $m^t \in \Delta^{K-1}$ and maintaining diversity across strategies.

This formulation preserves the simplex, ensures positivity, and admits invariant measures under mild conditions (Appendix E). While related to replicator–mutator equations, FinEvo grounds selection in payoff signals $V(t)$ and explicitly incorporates innovation and perturbations, yielding a more flexible and robust model of population evolution.

**Independence Assumptions.** The independence assumptions (A1–A3) used in our variance decomposition serve only to obtain a closed-form expression. They are *not* required for the FinEvo SDE itself: the dynamics are unchanged when payoff and environmental shocks are correlated. In the correlated case, the decomposition simply acquires additional covariance terms (see Appendix D).

**Mechanism decomposition.** In discrete approximation, the short-run change in share $\Delta x_k$ is shaped by three forces: $\Delta x_k \approx \beta x_k^t \big( V_k^t - \bar{V}^t \big) + \mu \, (m_k^t - x_k^t) + x_k^t \Big( \gamma \sigma_k \eta_k^t - \gamma \sum_j x_j^t \sigma_j \eta_j^t \Big), \quad \eta_k^t \sim \mathcal{N}(0,1)$.

Correspondingly, their contributions to short-run volatility are

$$\text{Var}(\Delta x_k) \approx \underbrace{\beta^2 (x_k^t)^2 \frac{\nu_k^2}{2\lambda_k}}_{\text{Selection}} + \underbrace{\mu^2 \frac{m_k(1-m_k)}{\alpha_0+1}}_{\text{Innovation}} + \underbrace{\gamma^2 \left( (x_k^t)^2 \sigma_k^2 - 2x_k^t \sigma_k \sum_j x_j^t \sigma_j + \left( \sum_j x_j^t \sigma_j \right)^2 \right)}_{\text{Perturbation}}.$$

Together, these decompositions show how selection, innovation, and perturbation jointly shape both the direction and volatility of evolutionary adaptation in financial markets.Detailed proof can be seen in the Appendix C.

## 3 EXPERIMENTS AND ANALYSIS OF EVOLUTIONARY DYNAMICS

### 3.1 EXPERIMENTAL SETUP

To study how financial strategies evolve in an ecological setting, we build a multi-agent simulation environment. Within this environment, we test two scenarios: one with artificial shocks and another with real-world news events.We simulate a synthetic market with news shocks, multi-indicator redistribution, and eco-evolutionary updates (see Appendix G.6.1). A full discussion of the motivation for these scenarios is provided in Appendix G.7.

**Agent Population**   The market is composed of 20 trader agent archetypes, each with distinct behaviors, information sources, and decision-making logic. For clarity, we group these agents into four broad categories: (1) **Rule-based Agents**, which include traditional strategies such as trend-following, mean-reversion (Jegadeesh & Titman, 1993; De Bondt & Thaler, 1985), and noise trading, along with fundamental (informational) traders; we further endow them with heterogeneous trading styles (e.g., *aggressive* and *conservative*) to reflect real-world diversity. (2) **Deep Learning Agents**, supervised models such as *Informer* and *TimeMixer* (Zhou et al., 2021; Wang et al., 2024), which use neural networks to forecast price direction from market indicators. (3) **Reinforcement Learning (RL) Agents**, adaptive strategies (e.g., *DoubleDQN*, *PPO*) (Van Hasselt et al., 2016; Schulman et al., 2017) that learn policies online by interacting with the market to maximize cumulative returns. (4) **Large Language Model (LLM) Agents**, which parse textual information such as news and announcements to infer market sentiment and generate trading signals (e.g., *FinCon*, *TradingAgents*) (Yu et al., 2024; Xiao et al., 2024),with GPT-4o-mini(Hurst et al., 2024) serving as the backbone language model. Details of each agent's design are presented in Appendix F.2.1. Abbreviations are adopted throughout the main text, with the full mapping summarized in Appendix F.1, and complete experimental parameter configurations can be found in Appendix G.6.2.

**Simulation Scenarios**   Each scenario is evaluated through Monte Carlo experiments, with **128 independent runs** per configuration. We consider two settings: (1) **Artificial Shocks**, where predefined perturbations are injected into the market to examine system stability and resilience (see Appendix G.6.4); and (2) **Real-World News**, simulating July 2023–July 2025 with actual historical news streams(e.g., GDELT and Reuters Markets; details in Appendix G.6.5). The market operates as a continuous double auction from 09:00 to 17:00 (slightly longer than real trading hours to allow market states to converge) each day, enabling event-driven and LLM agents to process real information and interact under realistic intraday constraints.

**Analysis Metrics**   Our evaluation framework follows a micro–meso–macro structure (detailed in Appendix G.6.3). At the **micro level**, we assess individual performance and diversity through population proportion (market shares of each agent), agent win rate (frequency with which an agent's return is ranked within the top three performers across evaluation trials), and financial metrics including return with confidence interval, Sharpe ratio, maximum drawdown, and turnover. At the **meso level**, we capture population diversity and interaction structures via concentration (HHI), strategy entropy, modularity (community strength), synergy vs. antagonism (correlation heatmaps), co-occurrence frequency, and mutual-information networks reflecting dependency and alliance reconfiguration. At the **macro level**, we evaluate system volatility and regime shifts by decomposing volatility into performance pressure ($V_{\text{select}}$), innovation/social mixing ($V_{\text{innovation}}$), and environmental Perturbation ($V_{\text{perturbation}}$),as well as measuring phase changes , which captures shifts in the dominant strategy cluster. A formal definition is provided in Appendix G.6.3. For visualization, raw

time-series data (e.g., population shares and volatility metrics) are smoothed using a Savitzky–Golay filter. In addition, following standard practice in empirical finance, we report *Excess Kurtosis*, *Skewness*, *Sharpe Dispersion*, and *Vol-of-Vol* to quantify fat tails, return asymmetry, cross-strategy performance dispersion, and volatility clustering. These metrics, widely used to characterize Stylized Facts of real markets, provide an external validation of the realism of the endogenous dynamics generated by FinEvo.

## 3.2 SIMULATION UNDER ARTIFICIAL SHOCKS

We test how the market ecology adapts under large external artificial disturbances. In our synthetic market, we inject news shocks between 12:00 and 13:30, considering both a *strongly positive* and a *strongly negative* scenario. This setting allows us to examine adaptation at the micro, meso, and macro levels.

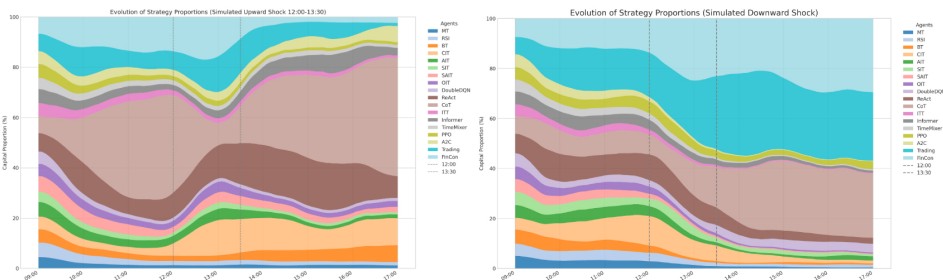

Figure 1: Agent population shares over time. Left: positive shock; Right: negative shock.

**Micro: Market Share Dynamics.** Figure 1 shows that responses are strongly asymmetric. Positive shocks increase diversity by reducing CoT's dominance and redistributing market share across multiple agents (e.g., CIT, ReAct), whereas negative shocks reinforce concentration, leading to LLM-dominated equilibria (FinCon, ReAct). Thus, shocks drive regime-dependent transitions in dominance concentration, highlighting dynamics not captured by static benchmarks.

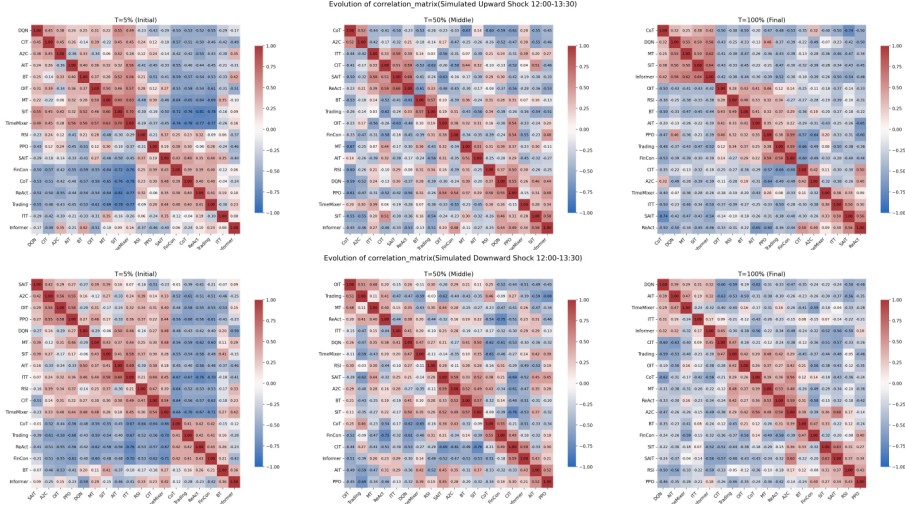

Figure 2: Correlation matrices over time under positive (top) and negative (bottom) shocks. Red = cooperation, Blue = competition.

**Meso: Interaction Structures and Stability.** Figure 2 tracks the reorganization of alliances. Before the shock, large red clusters indicate coordination groups where many strategies move together. During the shock, these clusters fragment as blue bands appear, reflecting stronger competition and divergence. After the shock, new cooperative clusters form, but their structures diverge: pluralistic

in the positive case, concentrated around LLMs in the negative case. This contrast highlights the path-dependent nature of adaptation.

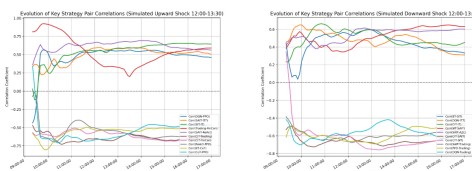 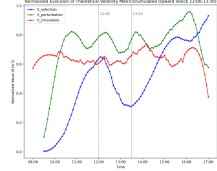 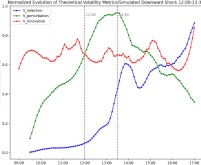

(a) Strategy-pair correlations under shocks.      (b) Volatility decomposition under shocks.

Figure 3: Responses under external shocks: Left = positive shock, Right = negative shock. In (b), Blue = selection, Red = innovation, Green = perturbation.

Zooming into individual pairs (Figure 3a), we observe three consistent patterns: (1) some pairs maintain persistent cooperation, (2) others maintain persistent competition, and (3) shocks induce only localized reorganization. Thus, while alliances shift, the broader cooperation–competition backbone remains intact, providing structural stability.

**Macro: Volatility Decomposition.** Figure 3b decomposes volatility into three components: selection ($V_{\text{selection}}$), innovation ($V_{\text{innovation}}$), and perturbation ($V_{\text{perturbation}}$). Across both scenarios, shocks trigger a spike in perturbation, destabilizing alliances. Afterward, selection dominates: in the positive case, it supports a pluralistic equilibrium, while in the negative case, it drives convergence toward LLM dominance. Innovation remains steady, acting as a background force that preserves diversity rather than driving short-term change.

**Summary.** External shocks reshape the market at all levels: they break monopolies or reinforce concentration (micro), fragment and reorganize alliances (meso), and amplify volatility before selection restores order (macro). Overall, adaptation is governed by the interplay of selection, innovation, and perturbation. Shocks amplify perturbation in the short run, but long-run equilibria are primarily shaped by selection, with innovation maintaining diversity.

For completeness, we also analyze the evolution of co-occurrence matrices and mutual-information networks under shocks, which provide complementary evidence of robustness from the perspective of networked interactions. The detailed results are reported in Appendix G.1.

### 3.3 Empirical Analysis of Intra-Day Evolutionary Dynamics

After examining the adaptive dynamics under simulated shocks, we next turn to *real-world* data and analyze the **intra-day evolutionary dynamics** within a single trading day. This allows us to validate whether the eco-evolutionary patterns observed in controlled experiments persist under actual market conditions.

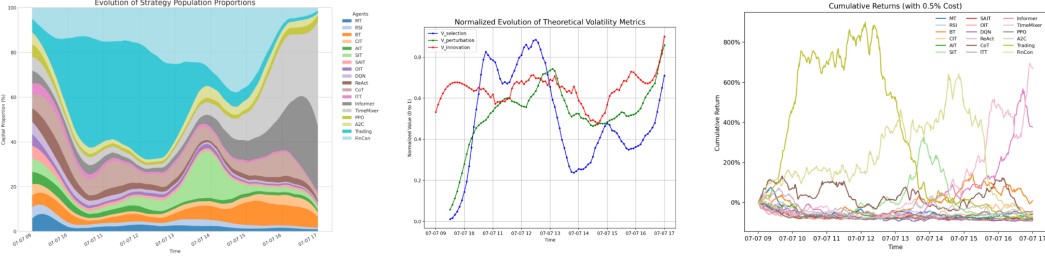

Figure 4: Intra-day evolution on July 7, 2025: (left) agent population proportions, (middle) theoretical volatility metrics, and (right) cumulative returns under a 0.5% transaction cost (including slippage).

**Intra-Day Adaptation and Profitability.** Figure 4 shows that intra-day dynamics follow a multi-phase cycle: strong selection initially concentrates capital in dominant strategies, sustained inno-

Table 1: Performance metrics by agent (Return with 95% CI, Sharpe, MaxDrawdown, Turnover;2025-07-07, N=128). Positive returns are shaded in gray.

| Agent | Return | Sharpe | MaxDD | Turn. | Agent | Return | Sharpe | MaxDD | Turn. | Agent | Return | Sharpe | MaxDD | Turn. |
|---|---|---|---|---|---|---|---|---|---|---|---|---|---|---|
| MT | -0.832 ± 0.064 | -2.016 | -0.908 | 0.157 | ReAct | -0.685 ± 0.025 | 15.972 | -0.978 | 0.207 | Informer | 4.333 ± 0.022 | 26.928 | -0.771 | 0.362 |
| RSI | -0.849 ± 0.023 | -3.546 | -0.888 | 0.122 | CoT | -0.560 ± 0.046 | 5.118 | -0.815 | 0.434 | TimeMixer | 6.738 ± 0.031 | 30.392 | -0.790 | 0.463 |
| BT | 0.034 ± 0.010 | 10.804 | -0.802 | 0.309 | ITT | -0.896 ± 0.066 | -1.228 | -0.910 | 0.119 | PPO | -0.431 ± 0.033 | 6.641 | -0.859 | 0.123 |
| CIT | -0.544 ± 0.032 | 6.265 | -0.844 | 0.163 | Trading | -0.746 ± 0.048 | -2.244 | -0.979 | 0.890 | A2C | -0.733 ± 0.004 | 0.847 | -0.837 | 0.197 |
| AIT | -0.807 ± 0.026 | -3.121 | -0.838 | 0.126 | FinCon | -0.685 ± 0.045 | -2.605 | -0.958 | 0.845 | DQN | -0.885 ± 0.034 | 13.436 | -0.955 | 0.139 |
| SIT | -0.673 ± 0.057 | 0.624 | -0.948 | 0.334 | SAIT | -0.866 ± 0.010 | -0.933 | -0.909 | 0.099 | OIT | -0.915 ± 0.032 | 2.265 | -0.942 | 0.130 |

vation later restores diversity and balance, and heightened perturbation near the close disrupts alliances before a new equilibrium emerges. Informer and TimeMixer ultimately prevail, highlighting how evolutionary pressures jointly shape regime transitions within a single trading day. Beyond survival, a natural question is which agents convert adaptation into profitability under market frictions. Table 1 reports averages over the same rounds (Return with 95% CI, Sharpe, MaxDrawdown, Turnover). Most agents lose, but a few achieve positive returns (shaded) despite large drawdowns, enabled by diversity-preserving innovation. Higher turnover further indicates that profitable agents trade more actively, highlighting a market ecology that is both selective and tolerant.

**Dynamic Alliance Reconfiguration.** Beyond individual outcomes, we next examine the structural layer of interactions. Figure 5 illustrates the intra-day evolution of agent-agent correlations, showing a dynamic interplay between *synergistic alliances* and *strategic antagonisms*. The market initially features multiple competing clusters with strong internal coordination, which are subsequently fragmented and reorganized under rising environmental perturbation. Toward the close, these transient shifts consolidate into a new regime characterized by restructured alliances and altered power balances. We further report the intra-day evolution of co-occurrence matrices and mutual-information networks in Appendix G.1.

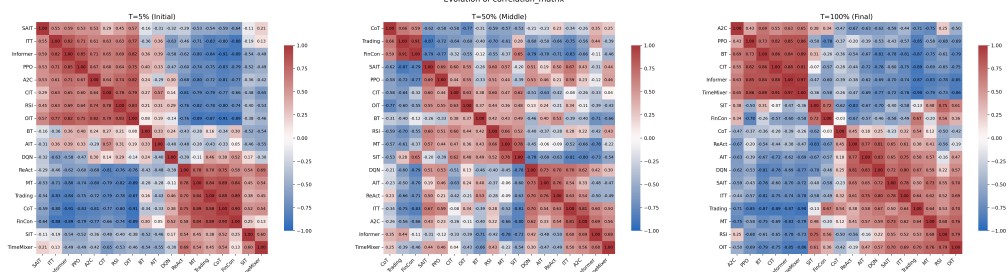

Figure 5: Dynamic evolution of intra-day correlation matrices on July 7, 2025. Red blocks represent *synergistic alliances*, while blue blocks indicate *strategic antagonism*.

**Performance Rankings in Bull vs. Bear Markets.** Finally, moving from intra-day fluctuations to market cycles, we evaluate whether these patterns persist across long-term bull and bear regimes. We approximate January–October 2022 as a *bear market phase*, objectively defined by persistent downward dynamics in the synthetic environment, and October 2024–July 2025 as a *bull market phase*, characterized by sustained recoveries and new highs. Figure 6 compares podium finishes and top-3 win rates across these regimes.

Our analysis shows that agent performance is highly **regime-dependent**: bull markets produce a concentrated dominance by a few strategies, whereas bear markets foster a more dispersed competitive landscape. Notably, **LLM-based agents consistently rank higher across both regimes**, suggesting that their ability to integrate diverse informational cues and adapt flexibly confers a significant robustness advantage over strategies based on fixed rules or purely quantitative signals.

**Evolutionary Game Modeling vs. Agent-Based Simulation in Market Environments** Table 2 compares the performance of FinEvo, ABIDES, and PyMarketSim during bull and bear markets. FinEvo exhibits a dynamic, adaptive environment where agents continuously adjust strategies, promoting greater diversity and competition. In contrast, ABIDES and PyMarketSim are more rigid, with fewer strategies dominating over time, limiting adaptability and diversity. This demonstrates

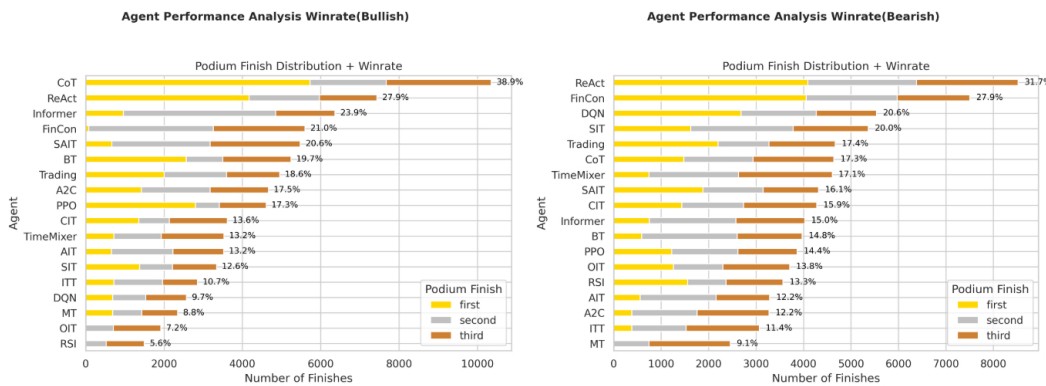

Figure 6: Agent performance rankings in bull vs. bear markets.

Table 2: Comparison of Ecological Metrics Across Evolutionary Game Modeling and Agent-Based Simulation

| Platform & Phase | Entropy$_{mean}$ | HHI$_{mean}$ | Top1$_{mean}$ | AvgAbsCorr$_{mean}$ | Modularity$_{mean}$ | PhaseChanges$_{mean}$ |
|---|---|---|---|---|---|---|
| FinEvo (bull) | $0.732 \pm 0.016$ | $0.226 \pm 0.032$ | $0.453 \pm 0.024$ | $0.441 \pm 0.019$ | $0.091 \pm 0.028$ | $3 \pm 0.437$ |
| FinEvo (bear) | $1.167 \pm 0.036$ | $0.175 \pm 0.029$ | $0.241 \pm 0.067$ | $0.376 \pm 0.039$ | $0.063 \pm 0.018$ | $4 \pm 0.348$ |
| ABIDES (bull) | $0.516 \pm 0.045$ | $0.539 \pm 0.011$ | $0.746 \pm 0.031$ | $0.653 \pm 0.046$ | $0.153 \pm 0.023$ | $1 \pm 0.439$ |
| ABIDES (bear) | $0.623 \pm 0.011$ | $0.481 \pm 0.012$ | $0.615 \pm 0.045$ | $0.541 \pm 0.022$ | $0.132 \pm 0.032$ | $1 \pm 0.344$ |
| PyMarketSim (bull) | $0.539 \pm 0.005$ | $0.564 \pm 0.019$ | $0.773 \pm 0.046$ | $0.662 \pm 0.039$ | $0.159 \pm 0.034$ | $1 \pm 0.332$ |
| PyMarketSim (bear) | $0.662 \pm 0.037$ | $0.461 \pm 0.015$ | $0.719 \pm 0.023$ | $0.575 \pm 0.007$ | $0.144 \pm 0.019$ | $1 \pm 0.329$ |

FinEvo's strength in capturing market evolution, while the other platforms show more static behavior.

### 3.4 ABLATION STUDY

**Ablation on Evolutionary Pressures**    To disentangle the roles of the three evolutionary pressures (selection, innovation, perturbation), we remove each component in turn and compare the resulting population dynamics (Figure 7). Without $\mathbf{V}_{selection}$ (left), competitive reinforcement vanishes and all agents remain at comparable proportions, producing a static, inefficient ecology. Without $\mathbf{V}_{innovation}$ (middle), diversity collapses as FinCon and Trading rapidly dominate, leading to an oligopolistic lock-in. Without $\mathbf{V}_{perturbation}$ (right), dynamics become smoother but rigid: FinCon and Trading still dominate, while adaptive reorganization is muted. Overall, selection drives efficiency, innovation sustains diversity, and disturbances from the environment can amplify shocks within the system, preventing premature lock-in and enabling regime shifts..

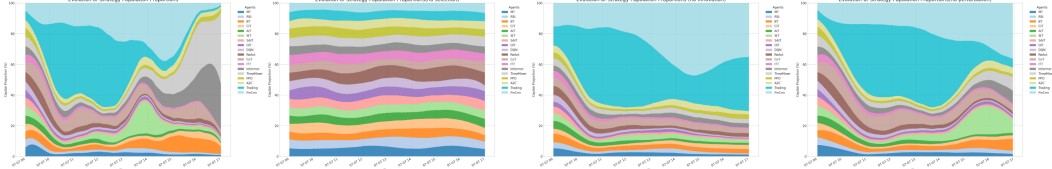

Figure 7: Ablation study of evolutionary pressures. From left to right: removing *selection*, *innovation*, and *perturbation*.

We complement the single-day dynamics in Figure 7 with aggregated statistics over the full evaluation period (Table 3). These global metrics provide a more robust picture: removing **selection** yields inflated entropy and frequent phase transitions, indicating unstable fragmentation; removing **innovation** collapses diversity and locks the ecology into an oligopoly; removing **perturbation** leads to higher concentration but fewer regime shifts, reflecting rigidity. Together, these results substantiate the earlier case study: selection promotes efficiency, innovation preserves diversity, and perturbation enables adaptive reorganization.

Table 3: Ablation study results with global ecological metrics (mean $\pm$ 95% CI).

| Setting | Entropy$_{mean}$ | HHI$_{mean}$ | Top1$_{mean}$ | AvgAbsCorr$_{mean}$ | Modularity$_{mean}$ | PhaseChanges$_{mean}$ |
|---|---|---|---|---|---|---|
| FinEvo | $0.828 \pm 0.14$ | $0.189 \pm 0.012$ | $0.344 \pm 0.015$ | $0.384 \pm 0.021$ | $0.074 \pm 0.009$ | $4 \pm 0.663$ |
| FinEvo (w/o selection) | $1.793 \pm 0.25$ | $0.081 \pm 0.108$ | $0.074 \pm 0.009$ | $0.139 \pm 0.018$ | $0.121 \pm 0.012$ | $14 \pm 0.767$ |
| FinEvo (w/o innovation) | $0.535 \pm 0.13$ | $0.477 \pm 0.014$ | $0.531 \pm 0.033$ | $0.553 \pm 0.024$ | $0.023 \pm 0.007$ | $1 \pm 0.483$ |
| FinEvo (w/o perturbation) | $0.719 \pm 0.04$ | $0.245 \pm 0.011$ | $0.376 \pm 0.017$ | $0.474 \pm 0.019$ | $0.036 \pm 0.008$ | $3 \pm 0.591$ |

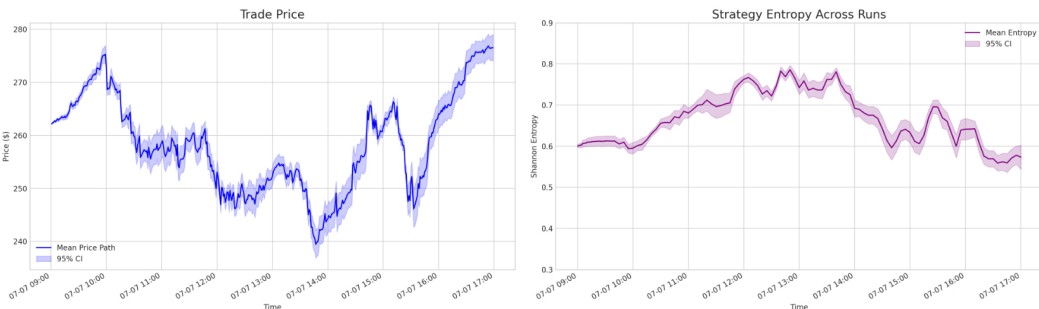

Figure 8: Robustness to random seeds (July 7, 2025). Left: trade price trajectories; Right: strategy entropy. Curves show means; shaded bands indicate 95% confidence intervals across 128 runs.

**Ablation on Robustness(Random seeds).** We conducted an ablation study based on the simulation carried out on July 7, 2025, with 128 rounds, in order to evaluate the robustness of our framework under different random seeds. As shown in Figure 8, we report the cross-run mean of trade price trajectories and strategy entropy, together with the shaded bands representing 95% confidence intervals across runs.

Despite minor run-to-run variations, the core patterns remain consistent: (i) on the price side, the market exhibits a stable upward trend following external shocks with bounded fluctuations; (ii) on the strategy side, the entropy remains persistently high, indicating sustained diversity and heterogeneity. The relatively narrow shaded regions suggest that different random seeds lead to similar dynamics, thereby demonstrating the stability and robustness of our framework under stochastic perturbations. In addition, we report trade price trajectories under simulated exogenous shocks, which further demonstrate the robustness of our framework(see Appendix G.1) for detailed results.

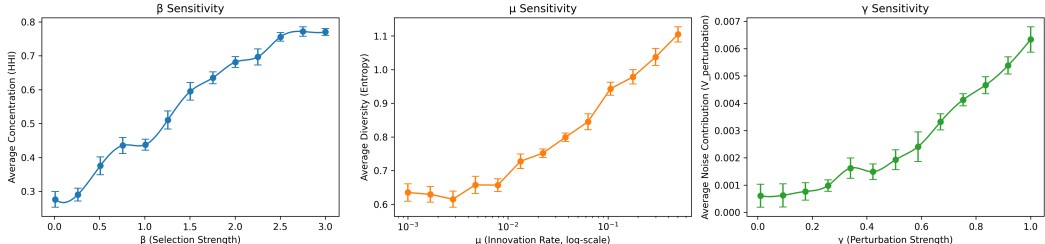

Figure 9: Sensitivity analysis on evolutionary parameters (simulation of July 7, 2025). Left: effect of selection strength $\beta$ on market concentration . Middle: effect of innovation rate $\mu$ on strategy diversity . Right: effect of perturbation intensity $\gamma$ on volatility contribution .Error bars indicate 95% CI across 128 runs.

**Ablation on Parameter Sensitivity.** We further conduct sensitivity experiments on the three key parameters of our evolutionary framework: the selection strength $\beta$, the mutation rate $\mu$, and the environmental perturbation intensity $\gamma$. We report system-level outcomes using three complementary metrics: (i) average concentration (Herfindahl–Hirschman Index, HHI), (ii) average diversity (Shannon entropy), and (iii) average perturbation contribution $V_{perturbation}$. Figure 9 illustrates the results. As $\beta$ increases, the average concentration rises significantly, suggesting that stronger selection pressure accelerates the dominance of a few strategies. In contrast, higher $\mu$ leads to increased diversity,

confirming that mutation and social mixing mechanisms help maintain heterogeneity. Meanwhile, increasing $\gamma$ monotonically elevates $V_{\text{perturbation}}$, demonstrating the independent contribution of exogenous perturbations to volatility. We additionally conducted a $\pm 20\%$ perturbed-parameter robustness test; the results (Appendix G.3) show that all stylized-fact metrics remain stable, confirming that FinEvo's dynamics are not sensitive to initial agent parameters.

**Ablation on Market Size $N$.** We examine the robustness of FinEvo with respect to the market size $N$, i.e., the number of agents participating in the ecology. Table 4 reports stylized-fact and performance metrics under four market sizes $N \in \{60, 80, 100, 120\}$, for both bull and bear regimes (128 runs per setting).

Table 4: Ablation on market size $N$ (128 runs). Stylized-fact metrics remain stable across $N$. Values are mean $\pm$ 95% CI.

| Regime | $N$ | Ex. Kurtosis$_{\text{mean}}\uparrow$ | Skewness$_{\text{mean}}\downarrow$ | Sharpe Disp.$_{\text{mean}}\uparrow$ | Vol-of-Vol$_{\text{mean}}\uparrow$ |
|---|---|---|---|---|---|
| Bullish | 120 | $3.143 \pm 0.141$ | $-1.539 \pm 0.132$ | $1.673 \pm 0.056$ | $0.009 \pm 0.0004$ |
| | 100 | $3.163 \pm 0.138$ | $-1.532 \pm 0.129$ | $1.669 \pm 0.061$ | $0.008 \pm 0.0004$ |
| | 80 | $3.178 \pm 0.165$ | $-1.541 \pm 0.158$ | $1.657 \pm 0.073$ | $0.008 \pm 0.0005$ |
| | 60 | $3.159 \pm 0.178$ | $-1.516 \pm 0.162$ | $1.659 \pm 0.067$ | $0.007 \pm 0.0005$ |
| Bearish | 120 | $4.326 \pm 0.169$ | $-1.805 \pm 0.152$ | $1.759 \pm 0.091$ | $0.012 \pm 0.0006$ |
| | 100 | $4.259 \pm 0.172$ | $-1.793 \pm 0.149$ | $1.732 \pm 0.089$ | $0.013 \pm 0.0007$ |
| | 80 | $4.115 \pm 0.175$ | $-1.765 \pm 0.161$ | $1.741 \pm 0.138$ | $0.013 \pm 0.0007$ |
| | 60 | $4.251 \pm 0.197$ | $-1.772 \pm 0.183$ | $1.725 \pm 0.119$ | $0.011 \pm 0.0007$ |

Across all values of $N$, FinEvo consistently maintains heavy-tailed PnL (high excess kurtosis), strong downside asymmetry (negative skewness), and comparable Sharpe dispersion and Vol-of-Vol, with mean returns fluctuating only within a narrow band.

### 3.5 STYLIZED-FACT VALIDATION

A key question is whether FinEvo's endogenous price formation—despite Gaussian exogenous shocks—can reproduce classical market stylized facts (Cont, 2001). To address this, we compare stylized-fact metrics across **FinEvo**, **ABIDES**, and **PyMarketSim**, running 128 simulations in both bull and bear regimes and computing standard econometric measures (excess kurtosis, skewness, Sharpe dispersion, and Vol-of-Vol). FinEvo matches the heavy tails, negative skewness, and elevated volatility-of-volatility more closely than the baselines(Table 5), and this behavior is robust: a correlated, heavy-tailed shock variant yields similar stylized-fact metrics (Appendix G.4, Table 9).We also provide a small cross-asset robustness check in Appendix G.5.

Table 5: Comparison of stylized-fact metrics across simulators (128 runs). Values report mean $\pm$ 95% CI.

| Simulator / Regime | Excess Kurtosis$_{\text{mean}}\uparrow$ | Skewness$_{\text{mean}}\downarrow$ | Sharpe Dispersion$_{\text{mean}}\uparrow$ | Vol-of-Vol$_{\text{mean}}\uparrow$ |
|---|---|---|---|---|
| FinEvo (bull) | $3.163 \pm 0.138$ | $-1.532 \pm 0.129$ | $1.669 \pm 0.061$ | $0.008 \pm 0.0004$ |
| FinEvo (bear) | $4.259 \pm 0.172$ | $-1.793 \pm 0.149$ | $1.732 \pm 0.089$ | $0.013 \pm 0.0007$ |
| ABIDES (bull) | $0.315 \pm 0.041$ | $-0.209 \pm 0.036$ | $0.675 \pm 0.028$ | $0.004 \pm 0.0002$ |
| ABIDES (bear) | $0.562 \pm 0.053$ | $-0.474 \pm 0.047$ | $0.836 \pm 0.033$ | $0.007 \pm 0.0003$ |
| PyMarketSim (bull) | $0.446 \pm 0.044$ | $-0.186 \pm 0.032$ | $0.512 \pm 0.026$ | $0.003 \pm 0.0002$ |
| PyMarketSim (bear) | $0.591 \pm 0.058$ | $-0.438 \pm 0.041$ | $0.906 \pm 0.034$ | $0.005 \pm 0.0003$ |

## 4 CONCLUSION AND FUTURE WORK

In this work, we introduce an ecological game formalism for modeling market ecology, which unifies selection, innovation, and environmental perturbation into a single mathematical formulation. We implement the framework in a multi-agent financial market environment and, through large-scale simulations, demonstrate its ability to capture the evolution of stability, concentration, and diversity across different market phases, thereby helping to narrow the gap between theoretical analysis and empirical dynamics. Overall, our study establishes a three-layer mapping from mechanisms to metrics to behaviors, highlighting both the interpretability and robustness of the proposed framework.FinEvo is designed with modular components that naturally support further extensions. A brief discussion of potential future directions is provided in Appendix I.

## 5 ETHICS STATEMENT

FinEvo provides an ecological game formalism for studying the evolution of financial strategies. The datasets used consist of publicly available financial market data and synthetic news streams, which do not involve personal or sensitive information; thus, privacy concerns are minimal. To reduce potential risks, our framework is intended for scientific investigation of market dynamics and policy evaluation, rather than for the development of deployable trading algorithms.

## 6 REPRODUCIBILITY STATEMENT

We are committed to ensuring the reproducibility of our results. All datasets used in this work are publicly available (see Appendix G.6.5), and we will release our source code, model implementations, and configuration files upon publication. To guarantee robustness and stability, each experiment was repeated with 128 Monte Carlo runs, and we report averaged results along with 95% confidence intervals (see Appendix G.6.1). Detailed descriptions of model architectures F.2.1, hyperparametersG.6.2, and evaluation matrics G.6.3 are provided in the main text and appendix to facilitate independent verification.

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

## A    RELATED WORK

### A.1    FINANCIAL AGENTS

The concept of *financial agents* abstracts market participants as decision-making entities with heterogeneous rules, objectives, and information sets. This framing emphasizes (i) heterogeneity — agents differ in behavioral assumptions such as trend-following, mean-reversion, or news sensitivity; and (ii) interaction — outcomes depend not only on external data but also on the actions of other participants. By treating strategies as agents, prior research provides a unifying abstraction that accommodates diverse modeling approaches.

Within this perspective, several families have emerged. **Rule-based agents** include momentum, mean-reversion, breakout, and pairs trading rules (Brock et al., 1992; Gatev et al., 2006), as well as extensions with overlays for volatility or stop-loss control (Harvey et al., 2018). **Data-driven agents** employ modern sequence models: TCN and N-BEATS for long-horizon forecasting (Oreshkin et al., 2019), Informer and FEDformer for efficient attention (Zhou et al., 2021; 2022), and foundation-style models such as TimeGPT and Chronos for zero-shot transfer (Garza et al., 2023; Ansari et al., 2024). **News- and multimodal agents** integrate text, prices, and sentiment, from FinBERT for financial language (Huang et al., 2023) to multimodal benchmarks such as FinMME (Luo et al., 2025). **LLM-based agents** extend trading logic with reasoning and tool use, exemplified by FinCon, FinMem, and FinGPT (Yu et al., 2024; 2025; Liu et al., 2023), as well as hybrid systems that combine forecasting models with LLM reasoning (Gan et al., 2025).

These developments illustrate how financial agents have evolved into adaptive, data-rich decision makers. Yet most evaluations still focus on individual performance rather than population-level adaptation, motivating our ecological perspective where heterogeneous agents interact and co-adapt within shared markets.

### A.2    REINFORCEMENT LEARNING

Reinforcement learning (RL) provides a complementary perspective to our ecological formulation in FinEvo. Both view markets as sequential decision processes under uncertainty: RL formalizes policy optimization over state–action trajectories (Sutton et al., 1998), while FinEvo frames the dynamics of strategy populations as stochastic replicator–mutator processes. The parallel is clear: maximizing long-horizon return in RL (Mnih et al., 2015; Schulman et al., 2017) mirrors survival and dominance of strategies in evolutionary games, though RL emphasizes a single adaptive policy while FinEvo studies heterogeneous populations.

RL draws on several traditions. Dynamic programming and optimal control inspire value-based methods (Bellman, 1966; Puterman, 2014), bandit and online learning emphasize regret minimization (Auer et al., 2002; Bubeck et al., 2012), and control-as-inference links planning to probabilistic modeling (Levine, 2018). These abstractions parallel FinEvo's payoff-driven selection, innovation, and environmental perturbation modeling.

Methodological advances further connect the two domains. Scalable actor–critic algorithms such as A3C (Mnih et al., 2016) highlight the role of parallelism and sample efficiency in training adaptive policies. Offline RL and off-policy evaluation (Kumar et al., 2020; Uehara et al., 2020) echo our concern with reproducible evaluation under counterfactual conditions. Risk-sensitive RL (Mihatsch & Neuneier, 2002) aligns with our treatment of shocks as ecological stressors. Preference- and feedback-driven RL (Christiano et al., 2017) resembles our innovation mechanism, where external signals redirect evolutionary trajectories. Meanwhile, sequence-decision formulations such as decision transformers (Chen et al., 2021) illustrate how supervised learning and RL converge, analogous to FinEvo's integration of forecasting with adaptation.

From this vantage point, RL and FinEvo can be seen as two instantiations of a common paradigm: learning under uncertainty with delayed feedback, balancing exploitation, exploration, and robustness. Yet, RL typically optimizes an isolated policy, leaving open how heterogeneous strategies evolve collectively under market pressures. FinEvo addresses this gap by extending RL principles to agent populations, bridging individual-level optimization with ecological dynamics.

### A.3 ECOLOGICAL GAMES

Ecological game theory extends classical game theory by modeling the adaptation and survival of strategy populations rather than equilibrium among rational individuals (Hofbauer & Sigmund, 1998; Sandholm, 2010). Originating in biology, dynamics such as the replicator equation (Taylor & Jonker, 1978) capture how successful strategies propagate, while mutation and drift introduce continual innovation.

This perspective has influenced machine learning through *evolutionary algorithms* (Bäck et al., 1997; Deb et al., 2002), *multi-agent evolutionary games* (Tuyls et al., 2006), and *population-based training* (Jaderberg et al., 2017). Recent work emphasizes open-endedness, as in AlphaStar's league training (Vinyals et al., 2019), autocurricula for exploration (Leibo et al., 2019), and open-ended strategy ecology (Grill et al., 2020). All highlight that robustness arises from continual adaptation within diverse populations.

In finance, ecological models explain heterogeneous traders' interactions (Lux & Marchesi, 1999; Hommes, 2006) and are instantiated in agent-based markets (LeBaron, 2006; Farmer & Foley, 2009). More recently, high-fidelity platforms like ABIDES (Byrd et al., 2020; Amrouni et al., 2021) provide infrastructure for market microstructure research, though they lack formal guarantees on population dynamics. Closer in spirit, the HFTE model (Mahdavi-Damghani, 2017) proposes a multi-species predator–prey ecosystem for high-frequency quantitative strategies, but relies on fixed topologies and genetic algorithms rather than a stochastic replicator–mutator dynamic. FinEvo complements this line of work by providing an explicit evolutionary game-theoretic SDE that links heterogeneous strategy populations with CDA microstructure. FinEvo advances beyond infrastructure by introducing a principled ecological formalism with stochastic replicator–mutator dynamics, enabling reproducible and theory-grounded evaluation of robustness and adaptation.

| Dimension | ABIDES | Mesa | PyMarketSim | TraderTalk | FinEvo (ours) |
|---|---|---|---|---|---|
| **Goal / Target** | Market microstructure simulation | General-purpose agent-based modeling | RL benchmark for trading strategies | LLM-driven bilateral trading behavior | **Mechanism-level evaluation of evolving financial ecosystems** |
| **Theoretical Guarantees** | None | None | Limited (training stability only) | None | **Simplex invariance, positivity, OU-process convergence** |
| **Mechanism Layering** | Agent-level only | Agent-level only | Agent-policy interactions | Dialogue-driven actions | **Selection / Innovation / Perturbation decomposition** |
| **Research Use** | Stress-test specific microstructure | Generic agent dynamics demos | RL strategy benchmarking | LLM role-playing for traders | **Systematic evaluation, stress-test, mechanism attribution** |
| **Reproducibility** | Depends on configuration | High (but non-financial focus) | Medium (sensitive to seeds) | Low (LLM variability) | **High, with cross-seed robustness and theoretical consistency** |

Table 6: Comparison of FinEvo with existing agent-based financial market platforms.

## B FROM FINEVO SDE TO DISCRETE SIMULATION UPDATE

This appendix derives the discrete-time update rule implemented in our simulator from the continuous-time *FinEvo SDE*. The goal is to make explicit how theoretical dynamics map to a practical update.

## B.1 CONTINUOUS-TIME FORMULATION

Let $X_t = (x_1(t), \ldots, x_K(t))^\top \in \Delta^{K-1}$ be the population shares with $\sum_k x_k(t) = 1$. The FinEvo dynamics are

$$dX_t = \underbrace{\beta \operatorname{diag}(X_t)(V(t) - \bar{V}(t)\mathbf{1})}_{\text{Selection}} dt + \underbrace{\mu (m_t - X_t)}_{\text{Innovation}} dt + \underbrace{\gamma \operatorname{diag}(X_t) \sigma P(X_t) dW_t}_{\text{Perturbation}}, \quad (1)$$

where $\bar{V}(t) = X_t^\top V(t)$, $m_t \in \Delta^{K-1}$ is the innovation distribution, $\sigma = \operatorname{diag}(\sigma_1, \ldots, \sigma_K)$, and $P(X) = I - \mathbf{1}X^\top$ projects perturbation onto the simplex tangent space.

## B.2 STEP 1: SELECTION–PERTURBATION FLOW

Consider first equation 1 without innovation. For each coordinate, define $y_k = \log x_k$. By Itô's lemma, to leading order on a short interval with $V(t)$ frozen,

$$dy_k \approx \beta (V_k - \bar{V}) dt + \gamma \sigma_k dB_k,$$

where $B_k$ are independent Brownian motions (after projection). Integrating over $\Delta t$ gives

$$y_k(t + \Delta t) - y_k(t) \approx \beta (V_k - \bar{V}) \Delta t + \gamma \sigma_k \sqrt{\Delta t} \xi_k, \quad \xi_k \sim \mathcal{N}(0, 1). \quad (2)$$

Exponentiating equation 2 yields the multiplicative-weights update

$$\hat{x}_k(t + \Delta t) = \frac{x_k(t) \exp(\beta V_k(t)\Delta t + \gamma \sigma_k \sqrt{\Delta t} \xi_k)}{\sum_j x_j(t) \exp(\beta V_j(t)\Delta t + \gamma \sigma_j \sqrt{\Delta t} \xi_j)}. \quad (3)$$

Normalization ensures $\hat{X}_{t+\Delta t} \in \Delta^{K-1}$.

## B.3 STEP 2: INNOVATION INJECTION

Reintroducing the innovation term in equation 1, a forward Euler step gives

$$X_{t+\Delta t} = (1 - \mu\Delta t) \hat{X}_{t+\Delta t} + \mu\Delta t\, m_t. \quad (4)$$

Thus the population is a convex combination of the selection–perturbation update and the innovation distribution.

## B.4 FINAL DISCRETE UPDATE

For simulation we take $\Delta t = 1$, yielding

$$x_k^{t+1} = (1 - \mu) \frac{x_k^t \exp(\beta V_k^t + \gamma \sigma_k \xi_k^t)}{\sum_j x_j^t \exp(\beta V_j^t + \gamma \sigma_j \xi_j^t)} + \mu m_k^t, \quad \xi_k^t \sim \mathcal{N}(0, 1). \quad (5)$$

Equation equation 5 is the update used in practice. It preserves positivity and the simplex by construction.

## B.5 PAYOFF DYNAMICS

Each payoff $V_k$ evolves as an Ornstein–Uhlenbeck (OU) process,

$$dV_k(t) = \lambda_k(\bar{f}_k - V_k(t)) dt + \nu_k dB_k(t), \quad (6)$$

whose exact discretization is

$$V_k^{t+1} = V_k^t e^{-\lambda_k \Delta t} + \bar{f}_k(1 - e^{-\lambda_k \Delta t}) + \nu_k \sqrt{\tfrac{1-e^{-2\lambda_k \Delta t}}{2\lambda_k}} \zeta_k^t, \quad \zeta_k^t \sim \mathcal{N}(0, 1). \quad (7)$$

## B.6 SUMMARY

- Selection $\to$ exponential weighting by payoff $V_k$, - Perturbation $\to$ log-normal shocks via $\sigma_k \xi_k^t$, - Innovation $\to$ convex mixing with $m_t$, - Payoffs $V_k$ follow OU discretization equation 7.

Together, equation 5 and equation 7 provide the discrete-time simulator consistent with the continuous-time FinEvo SDE.

## C  MECHANISM DECOMPOSITION

We aim to compute the second moment of the perturbation term under the per-strategy perturbation intensities $\sigma_k$ and a global scale $\gamma$. With the tangent (mean-subtracted) construction used in the main text, the per-component discrete perturbation is

$$\mathrm{N}_k^t = \gamma\, x_k^t \Big( \sigma_k\, \eta_k^t - \sum_j x_j^t \sigma_j\, \eta_j^t \Big), \qquad \eta_k^t \sim \mathcal{N}(0,1) \text{ i.i.d. across } k, t.$$

We compute

$$\mathbb{E}\big[ (\mathrm{N}_k^t)^2 \big] = \gamma^2 (x_k^t)^2\, \mathbb{E}\left[ \Big( \sigma_k\, \eta_k^t - \sum_j x_j^t \sigma_j\, \eta_j^t \Big)^2 \right].$$

Expanding the square and using independence with unit variance, $\mathbb{E}[\eta_i^t \eta_j^t] = \delta_{ij}$, we obtain

$$\mathbb{E}\left[ \Big( \sigma_k\, \eta_k^t - \sum_j x_j^t \sigma_j\, \eta_j^t \Big)^2 \right] = \sigma_k^2 - 2\sigma_k \sum_j x_j^t \sigma_j\, \mathbb{E}[\eta_k^t \eta_j^t] + \sum_{j,\ell} x_j^t x_\ell^t \sigma_j \sigma_\ell\, \mathbb{E}[\eta_j^t \eta_\ell^t]$$

$$= \sigma_k^2 - 2 x_k^t \sigma_k^2 + \sum_j (x_j^t)^2 \sigma_j^2.$$

Therefore,

$$\mathbb{E}\big[ (\mathrm{N}_k^t)^2 \big] = \gamma^2 (x_k^t)^2 \left( \sigma_k^2 - 2 x_k^t \sigma_k^2 + \sum_j (x_j^t)^2 \sigma_j^2 \right). \tag{8}$$

This per-strategy expression replaces the uniform-$\sigma$ formula $\sigma^2 (x_k^t)^2 \big( 1 - 2 x_k^t + \|x^t\|_2^2 \big)$. It reduces to the uniform case by setting all $\sigma_k \equiv \sigma$ and $\gamma \equiv 1$, yielding $\mathbb{E}\big[ (\mathrm{N}_k^t)^2 \big] = \sigma^2 (x_k^t)^2 \big( 1 - 2 x_k^t + \|x^t\|_2^2 \big)$.

### C.1  INDEPENDENCE AND VARIANCE ADDITIVITY.

To justify the variance decomposition used in the mechanism analysis, we separate the discrete update into three *centered* (zero-mean) random fluctuation components around a deterministic drift:

$$\Delta x_k^t = \underbrace{S_k^t}_{\text{selection fluct.}} + \underbrace{I_k^t}_{\text{innovation fluct.}} + \underbrace{N_k^t}_{\text{perturbation fluct.}} + \text{(deterministic drift).}$$

Only the centered fluctuations contribute to short-run variance. We work *conditional on $X_t$* and impose the following assumptions.

**Assumptions.**

(A1) (*Selection fluctuations*) Write the payoff signal as $V_k^t = \bar{V}_k^t + \zeta_k^t$, where $\bar{V}_k^t := \mathbb{E}[V_k^t \mid X_t]$ and $\zeta_k^t$ is the centered fluctuation. Define

$$S_k^t := \beta x_k^t \big( \zeta_k^t - \bar{\zeta}^t \big), \qquad \bar{\zeta}^t := \sum_j x_j^t \zeta_j^t,$$

so that $\mathbb{E}[S_k^t \mid X_t] = 0$. We assume $\zeta^t$ has diagonal conditional covariance $\mathrm{Cov}(\zeta_k^t, \zeta_j^t \mid X_t) = \delta_{kj}\, \nu_k^2 / (2\lambda_k)$.

(A2) (*Innovation fluctuations*) Let $m_t \sim \mathrm{Dir}(\alpha)$ be independent of $(\zeta^t, \eta^t)$ given $X_t$, with mean $m := \mathbb{E}[m_t]$ and centered fluctuation $\tilde{m}_t := m_t - m$. Define

$$I_k^t := \mu\, \tilde{m}_k^t,$$

so that $\mathbb{E}[I_k^t \mid X_t] = 0$ and $\mathrm{Var}(I_k^t \mid X_t) = \mu^2\, \frac{m_k(1-m_k)}{\alpha_0 + 1}$.

(A3) (*Perturbation fluctuations*) The environmental shocks $\eta^t = (\eta_1^t, \ldots, \eta_K^t)^\top$ are i.i.d. standard normal and independent of $(\zeta^t, m_t)$ given $X_t$. Define

$$N_k^t := \gamma \, x_k^t \Big( \sigma_k \eta_k^t - \sum_j x_j^t \sigma_j \eta_j^t \Big),$$

so that $\mathbb{E}[N_k^t \mid X_t] = 0$ and $\mathrm{Var}(N_k^t \mid X_t)$ is given in Eq. (8).

**Lemma (variance additivity).** Under (A1)–(A3),

$$\mathrm{Var}(\Delta x_k^t \mid X_t) = \mathrm{Var}(S_k^t \mid X_t) + \mathrm{Var}(I_k^t \mid X_t) + \mathrm{Var}(N_k^t \mid X_t).$$

*Proof.* By construction, $\mathbb{E}[S_k^t \mid X_t] = \mathbb{E}[I_k^t \mid X_t] = \mathbb{E}[N_k^t \mid X_t] = 0$. Independence in (A2)–(A3) implies the cross-covariances vanish: $\mathrm{Cov}(S_k^t, I_k^t \mid X_t) = 0$, $\mathrm{Cov}(S_k^t, N_k^t \mid X_t) = 0$, $\mathrm{Cov}(I_k^t, N_k^t \mid X_t) = 0$. Therefore,

$$\mathrm{Var}(\Delta x_k^t \mid X_t) = \mathrm{Var}(S_k^t + I_k^t + N_k^t \mid X_t) = \sum_{\bullet \in \{S, I, N\}} \mathrm{Var}(\bullet_k^t \mid X_t). \qquad \square$$

**Remarks.** (i) The deterministic drift (e.g., $\beta x_k^t (\bar{V}_k^t - \bar{V}^t)$ and $\mu(m_k - x_k^t)$) affects the conditional mean but not the centered variance. (ii) If one wishes to allow correlated payoff and environment shocks, replace (A1)–(A3) by a joint Gaussian with block covariance; the variance then includes explicit cross terms $\mathrm{Cov}(S_k^t, N_k^t \mid X_t)$, etc. Our empirical results use (A1)–(A3), which are standard in evolutionary/replicator SDEs and lead to the closed forms reported in the main text.

## C.2 PARAMETER SUMMARY.

| Symbol | Description | Typical usage/role |
|---|---|---|
| $x_k^t$ | Proportion of agents using $\pi_k$ at time $t$ | $\sum_k x_k^t = 1$ |
| $f_k^t$ | Empirical mean payoff of strategy $k$ at $t$ | Observed fitness |
| $\beta$ | Selection intensity | Tuning parameter |
| $\gamma_{kj}$ | Ecological effect of $j$ on $k$ | Set to 0 in baseline; $> 0$ competition |
| $\sigma_k$ | perturbation intensity for strategy $k$ | Calibrated or fixed |
| $\eta_k^t$ | Standard normal noise | Simulation draw |
| $\mu$ | Innovation probability | Tuning parameter |
| $m_k$ | Innovation distribution | User-defined |
| $\Delta t$ | Time step (usually 1) | Simulation round |

## C.3 CONCLUSION.

This derivation demonstrates that the agent-level elimination, selection, and replacement mechanism implemented in our simulation is a numerically consistent discretization of the generalized Lotka–Volterra stochastic ecology, augmented with innovation. The result is a unified framework with theoretical foundations in evolutionary dynamics and practical applicability for complex agent-based market studies.

## D VARIANCE DECOMPOSITION UNDER CORRELATED PAYOFF AND ENVIRONMENTAL SHOCKS

In this section, we extend the variance analysis in Appendix C.1 to the case where payoff shocks and environmental shocks are correlated. We emphasize that correlation does *not* alter the FinEvo SDE itself; only the decomposition gains additional covariance terms.

**FinEvo SDE**

$$dX_t = \underbrace{\beta \, X_t \odot (V_t - \bar{V}_t)}_{\text{Selection}} dt + \underbrace{\mu(m_t - X_t)}_{\text{Innovation}} dt + \underbrace{\sigma \, P(X_t) \, dB_t}_{\text{Perturbation}}, \tag{C.1}$$

where $P(X_t)$ is the tangent-space projection ensuring simplex invariance, and $B_t \in \mathbb{R}^K$ is a vector-valued Brownian motion.

## D.1 ALLOWING CORRELATED SHOCKS

We now assume the Brownian increments satisfy

$$\mathrm{Cov}(dB_t) \;=\; \Sigma \, dt, \tag{9}$$

where $\Sigma$ is a symmetric positive semidefinite correlation matrix:

$$\Sigma = \begin{pmatrix} 1 & \rho_{12} & \cdots & \rho_{1K} \\ \rho_{21} & 1 & \cdots & \rho_{2K} \\ \vdots & & \ddots & \vdots \\ \rho_{K1} & \cdots & \rho_{K(K-1)} & 1 \end{pmatrix}.$$

This relaxation affects only the variance decomposition, not the SDE dynamics.

## D.2 VARIANCE OF A SINGLE COMPONENT

For any strategy population component $X_{k,t}$, Itô's lemma yields:

$$\mathrm{Var}(dX_{k,t}) = \underbrace{\mathrm{Var}\big[\beta X_{k,t}(V_{k,t} - \bar{V}_t)\big]}_{\text{Selection}} + \underbrace{\mathrm{Var}[\mu(m_{k,t} - X_{k,t})]}_{\text{Innovation}}$$
$$+ \underbrace{\sigma^2 \big(P\Sigma P^\top\big)_{kk}}_{\text{Perturbation}} + \underbrace{2\sigma \, \mathrm{Cov}\big(\beta X_{k,t}(V_{k,t} - \bar{V}_t), \, (P\, dB_t)_k\big)}_{\text{Selection–Perturbation Cross Term}}. \tag{C.2}$$

## D.3 FULL MATRIX VARIANCE

For the full vector process $X_t$, we have:

$$\mathrm{Var}(dX_t) = \beta^2 \, \mathrm{Var}\big[X_t \odot (V_t - \bar{V}_t)\big] \;+\; \mu^2 \, \mathrm{Var}(m_t - X_t)$$
$$+ \; \sigma^2 \, P\Sigma P^\top \;+\; 2\beta\sigma \, \mathrm{Cov}\big(X_t \odot (V_t - \bar{V}_t), \, P\, dB_t\big). \tag{C.3}$$

## D.4 RELATION TO THE UNCORRELATED CASE

Under independence assumptions (A1–A3) in Appendix C.1, we have

$$\Sigma = I, \qquad \mathrm{Cov}\big(\text{selection}, \, dB_t\big) = 0,$$

and Equations (C.2)–(C.3) reduce to the clean additive decomposition in the main text. When shocks are correlated, the FinEvo SDE itself remains unchanged, and only additional covariance terms appear in the variance decomposition, consistent with the remark in Appendix C.1.

# E MINIMAL THEORETICAL GUARANTEES

## E.1 CONTINUOUS-TIME FINEVO SDE (FOR REFERENCE).

We explicitly write the continuous-time dynamics underlying our discrete updates:

$$dX_i = X_i \left( f_i(X,t) - \sum_j X_j f_j(X,t) \right) dt$$
$$+ \mu \, (m_i - X_i) \, dt \tag{10}$$
$$+ \sigma_i X_i \left( dW_i - \sum_j X_j \, dW_j \right), \qquad i = 1, \dots, K.$$

where $X(t) \in \Delta^K := \{x \in \mathbb{R}^K_{\geq 0} : \sum_i x_i = 1\}$ are strategy shares, $m \in \mathrm{int}(\Delta^K)$ with $\min_i m_i > 0$, $\mu \geq 0$, and $\{W_i\}$ are independent Brownian motions. The diffusion is *tangent* to $\Delta^K$.

### E.2 Assumptions.

(i) $f_i(\cdot, t)$ are locally Lipschitz with linear-growth bounds in $X$, uniformly on compact time sets; (ii) $m \in \text{int}(\Delta^K)$, $\mu \geq 0$, $\sigma_i \geq 0$; (iii) Brownian motions $\{W_i\}$ are independent. When we couple with exogenous value processes $V_i(t)$, we assume $V_i$ are OU with unique stationary laws and bounded moments; the discrete OU approximation is Eq. equation 7.

**Lemma 1 (Well-posedness)** *Under the above assumptions, the SDE equation 10 admits a unique strong solution on $[0, \infty)$ for any $X(0) \in \Delta^K$.*

**Proposition 1 (Simplex invariance and positivity)** *Let $X(0) \in \Delta^K$. Then the solution to equation 10 satisfies $\sum_i X_i(t) \equiv 1$ for all $t \geq 0$ and $X_i(t) \geq 0$ almost surely. If, in addition, $X(0) \in \text{int}(\Delta^K)$, $\mu > 0$, and $\min_i m_i > 0$, then $X_i(t) > 0$ for all $t > 0$ almost surely (the boundary is non-absorbing).*

*Proof sketch.* Summing equation 10 over $i$ cancels both the selection drift and the tangent diffusion; the innovation term sums to $\mu(\sum_i m_i - \sum_i X_i) = 0$, hence mass is conserved. Multiplicative diffusion vanishes on the boundary, implying non-negativity. With $\mu > 0$ and $\min_i m_i > 0$, the drift $\mu(m_i - X_i)$ points strictly inward. $\square$

**Proposition 2 (Invariant measure and quasi-stability)** *Consider the joint Markov process $(X(t), V(t))$ with $X$ governed by equation 10 and $V$ OU. If $\mu > 0$ and $\min_i m_i > 0$, then at least one invariant probability measure exists on $\Delta^K \times \mathbb{R}^K$. Moreover, in the small-perturbation/weak-innovation regime ($\sigma \to 0$, $\mu \to 0$ with $\mu = O(\sigma^2)$), invariant measures concentrate near the attractor set of the deterministic replicator–mutator ODE $\dot{X}_i = X_i(f_i - \sum_j X_j f_j) + \mu(m_i - X_i)$, yielding quasi-stable ecological compositions.*

*Proof sketch.* OU has a unique stationary law with bounded moments. By Prop. 1, $X(t)$ remains in the compact simplex; the joint process is Feller and tight, so Krylov–Bogoliubov yields existence of an invariant measure. Small-perturbation concentration follows from standard Freidlin–Wentzell arguments for coupled systems. $\square$

### E.3 Discrete-time counterpart (consistency).

Our simulation update implements exponential selection followed by innovation/mutation. The selection-normalization step is Eq. equation 3, and the innovation mixing is the final update Eq. equation 5. Together, these define a positivity-preserving, simplex-invariant map. In particular,

$$\sum_k x_k^{t+1} = 1, \quad x_k^{t+1} \geq 0, \quad x_k^{t+1} > 0 \text{ for } t \geq 1 \text{ if } \mu > 0 \text{ and } \min_k m_k > 0.$$

Coupled value signals use the OU discretization in Eq. equation 7. Under $\Delta t \to 0$, the discrete scheme is a first-order consistent approximation to Eq. equation 10.

## F Agent Library and Initialization.

### F.1 Agent Name Mapping

Table F.1 summarizes the mapping between full agent names and their abbreviations used throughout the paper. Agents are organized by category (rule-based, informed, reinforcement learning, time-series models, and LLM-based traders). This mapping provides a concise reference to ensure consistency across figures, tables, and analyses in the main text.

### F.2 Agent library

This section describes the full design of each agent archetype in our framework, including their trading objectives, information sources, decision rules, and execution mechanisms. By detailing both classical strategies (e.g., technical and informed traders) and modern adaptive agents (e.g., deep learning, reinforcement learning, and LLM-based models), the agent library highlights the heterogeneity of behaviors underlying FinEvo's ecological simulations.

Table 7: Agent Name Mapping

| Category | Full Name | Abbreviation |
|---|---|---|
| Technical Traders | MomentumTrader | MT |
| | RSITrader | RSI |
| | BreakoutTrader | BT |
| Informed Traders | ConservativeInformedTrader | CIT |
| | AggressiveInformedTrader | AIT |
| | SpreadAwareInformedTrader | SAIT |
| | ScaledInformedTrader | SIT |
| | OpportunisticInformedTrader | OIT |
| Reinforcement Learning | DoubleDQNTrader | DoubleDQN |
| | PPO_Trader | PPO |
| | A2C_Trader | A2C |
| Time-Series Models | iTransformer_Trader | ITT |
| | Informer_Trader | IT |
| | TimeMixer_Trader | TimeMixer |
| LLM-based Agents | LLM_ReAct_Trader | ReAct |
| | LLM_CoT_Trader | CoT |
| | LLM_Trading_Trader | Trading |
| | LLM_FinCon_Trader | FinCon |

### F.2.1 INFORMATIONAL TRADERS AGENT

- **Trading Objective**
  Exploit discrepancies between the market price $P_t$ and an oracle-provided fundamental value $P_t^*$ to generate profitable trades without long-term directional bias.

- **Signal Computation**
  At each time $t$, compute the relative deviation

$$\delta_t = \frac{P_t - P_t^*}{P_t^*}, \quad P_t = \frac{P_t^b + P_t^a}{2}$$

- **Decision Rule**
  Given a deviation threshold $\theta > 0$, the agent's trading decision $a_t^i \in \{\text{buy}, \text{sell}, \text{hold}\}$ is defined as:

$$a_t^i = \begin{cases} \text{buy}, & \text{if } \delta_t < -\theta \text{ and } h_t^i \leq 0 \\ \text{sell}, & \text{if } \delta_t > \theta \text{ and } h_t^i > 0 \\ \text{hold}, & \text{otherwise} \end{cases}$$

  where $h_t^i$ denotes agent $i$'s current inventory position at time $t$.

- **Execution Mechanism**
  Depending on variant, the agent submits orders as follows:

  - *Conservative*: place **limit orders** at

$$P^{\text{buy}} = P_t^b + \epsilon, \quad P^{\text{sell}} = P_t^a - \epsilon$$

  - *Aggressive*: place **market orders** immediately at the best ask/bid.
  - *Opportunistic*: place **market orders** of size

$$q_t = \begin{cases} \lambda\, q, & \text{if liquidity is low} \\ q, & \text{otherwise} \end{cases}$$

  where $q$ is the base trade quantity and $\lambda > 1$ is an amplification factor.

### F.2.2 MARKET MAKER AGENT

- **Trading Objective**
  Maintain liquidity in the limit order book by continuously submitting limit orders on both bid and ask sides across multiple price levels, aiming to profit from the bid-ask spread.

- **Signal Computation**
  At each decision time $t$, observe the best bid and ask prices:
  $$P_t^b, \quad P_t^a$$
  Calculate the mid-price and half-spread:
  $$P_t^m = \frac{P_t^b + P_t^a}{2}, \quad s_t = \frac{P_t^a - P_t^b}{2}$$
  Sample the total order size per side from a predefined range:
  $$q_t \sim [q_{\min}, q_{\max}]$$
  Determine the number of price levels $L$ to quote on each side, sampled randomly.

- **Decision Rule**
  For each level $i = 0, 1, \ldots, L - 1$, compute limit order prices:
  $$P_{t,i}^b = P_t^m - s_t - i, \quad P_{t,i}^a = P_t^m + s_t + i$$
  Allocate order sizes at each level according to weight vector $w = (w_1, \ldots, w_L)$, satisfying
  $$\sum_{i=1}^{L} w_i = 1, \quad q_{t,i} = w_i \cdot q_t$$

- **Execution Mechanism**

  - **Polling mode (subscribe=False)**: Query current market spread and place symmetric limit orders around the mid-price with uniform size at each level.
  - **Subscription mode (subscribe=True)**: Subscribe to order book updates, randomly select $L \in \{1, \ldots, 5\}$ levels, and allocate volumes per the predefined weight vector $w$ from the levels quote dictionary.

### F.2.3 NOISE TRADER AGENT

- **Trading Objective**
  Operates without access to fundamental information or technical indicators. Makes randomized decisions to simulate irrational or uninformed trading behavior.

- **Decision Rule at time $t$**
  With equal probability $p = 0.5$, the agent executes one of two actions:

  - *Aggressive Execution (Market Order)*:
    Submits a market buy or sell order with equal probability:
    $$q_t \sim \mathcal{U}\{10, 20, ..., 100\}, \quad a_t = \begin{cases} \text{MarketBuy}(q_t), & \text{prob. } 0.5 \\ \text{MarketSell}(q_t), & \text{prob. } 0.5 \end{cases}$$

  - *Passive Execution (Limit Order)*:
    Submits a limit buy or sell order offset from the mid-price:
    $$P_t^m = \frac{P_t^b + P_t^a}{2}, \quad \epsilon_t \sim \mathcal{U}(0.001, 0.005)$$
    $$P_t^{\text{limit}} = \begin{cases} \lfloor P_t^m (1 - \epsilon_t) \rfloor, & \text{if buy} \\ \lceil P_t^m (1 + \epsilon_t) \rceil, & \text{if sell} \end{cases}$$
    $$a_t = \begin{cases} \text{LimitBuy}(q_t, P_t^{\text{limit}}), & \text{prob. } 0.5 \\ \text{LimitSell}(q_t, P_t^{\text{limit}}), & \text{prob. } 0.5 \end{cases}$$

- **Next Decision Time**
  The next wake-up time is sampled from:
  $$t_{\text{next}} = t + \Delta t, \quad \Delta t \sim \mathcal{U}(55, 65) \text{ seconds}$$

### F.2.4 EVENT-DRIVEN NEWS REACTION AGENT

- **Trading Objective**
  React only to significant market news events (e.g., macroeconomic shocks or asset-specific ratings) by placing immediate market orders in response to the sentiment direction of the news.

- **Information Source**
  Subscribes to an *EventBus* that broadcasts discrete news events. Each event includes:
  - `event_type` $\in \{$POSITIVE, NEGATIVE, UPGRADE, DOWNGRADE, $\ldots\}$
  - `content`: textual description (ignored by logic)

- **Decision Logic on Event Reception**
  At time $t$, when a news event $e_t$ is received:

$$\text{Action}_t = \begin{cases} \text{MarketBuy}(q), & \text{if event\_type} \in \{\text{POSITIVE}, \text{UPGRADE}\} \\ \text{MarketSell}(q), & \text{if event\_type} \in \{\text{NEGATIVE}, \text{DOWNGRADE}\} \\ \text{NoAction}, & \text{otherwise} \end{cases}$$

  where $q$ is the fixed order quantity (e.g., $q = 10$ shares).

- **Timing and Wake-up Frequency**
  The agent wakes up every $\Delta t = 10$ seconds to maintain kernel activity and ensure one-time subscription to the event stream, but trading decisions are purely event-driven.

- **Execution Mechanism**
  All trades are submitted as **market orders** immediately upon signal reception to ensure fastest reaction to major events.

### F.2.5 REINFORCEMENT LEARNING TRADER AGENT

- **Trading Objective**
  The RL-based agent aims to maximize cumulative portfolio returns by interacting with the environment in an online manner. At each time step, it observes the current market state, selects an action (buy/sell/hold), and updates its policy parameters through continual learning (online fine-tuning).

- **State Representation**
  The market state $s_t$ is represented as follows for different learning schemes:

$$\begin{aligned} \textbf{Q-Learning:} \quad & s_t = (x_t^{(1)},\ x_t^{(2)},\ x_t^{(3)}) \text{ (discretized)} \\ \textbf{PPO:} \quad & s_t = \left[ x_t^{(1)},\ x_t^{(2)},\ \ldots,\ x_t^{(n)} \right] \text{ (continuous and normalized)} \end{aligned}$$

  1

- **Action Policy**
  The agent selects actions from a discrete space:

$$\mathcal{A} = \{0 = \text{HOLD},\ 1 = \text{BUY},\ 2 = \text{SELL}\}$$

  Each action triggers a market order of fixed quantity $Q$ in the corresponding direction.

- **Reward Function**
  The agent is rewarded based on changes in portfolio value:

$$r_t = PV_t - PV_{t-1} - \lambda \cdot \mathbb{I}(a_t \in \{\text{BUY}, \text{SELL}\})$$

  where $PV_t$ is portfolio value and $\lambda$ is a trade penalty coefficient.

- **Learning Mechanism**

  1. **Q-Learning (Tabular, Online)**
     The agent maintains a Q-table $Q(s, a)$ and updates it using the standard Bellman equation:

$$Q(s_t, a_t) \leftarrow Q(s_t, a_t) + \alpha \left[ r_t + \gamma \max_{a'} Q(s_{t+1}, a') - Q(s_t, a_t) \right]$$

     Parameters $(\alpha, \gamma)$ denote the learning rate and discount factor, respectively. Updates occur online every $N$ steps from recent experience buffer $D$.

2. **Proximal Policy Optimization (PPO)**

   The agent learns a stochastic policy $\pi_\theta(a|s)$ using clipped surrogate objective:

   $$L^{\text{PPO}} = \mathbb{E}_t\left[\min\left(r_t(\theta)A_t,\ \text{clip}(r_t(\theta), 1-\epsilon, 1+\epsilon)A_t\right)\right]$$

   where $r_t(\theta) = \frac{\pi_\theta(a_t|s_t)}{\pi_{\theta_{\text{old}}}(a_t|s_t)}$ is the probability ratio, and $A_t$ is the advantage function. The agent collects $N$ steps of interaction and performs $K$ epochs of gradient updates for online fine-tuning.

### F.2.6 Technical Trader Agent

- **Trading Objective**

  Execute market orders based on common technical analysis signals derived from historical price data, aiming to capture trends or mean reversion.

- **State Observation**

  At each decision time $t$, the agent observes a recent price window:

  $$S_t = \{P_{t-n}, \ldots, P_t\}$$

  where $P_t$ is the latest trade price.

- **Trading Quantity**

  Fixed order size $Q$ for all trades.

- **Trading Strategies and Decision Rules**

  The agent implements one of the following strategies:

  1. *Momentum Agent (Moving Average Crossover)*

     Compute short-term and long-term moving averages:

     $$MA_t^{short} = \frac{1}{N_s}\sum_{i=t-N_s+1}^{t} P_i, \quad MA_t^{long} = \frac{1}{N_l}\sum_{i=t-N_l+1}^{t} P_i$$

     The trading action $a_t$ is:

     $$a_t = \begin{cases} \text{BUY}, & \text{if } MA_t^{short} > MA_t^{long} \text{ and position} \leq 0 \\ \text{SELL}, & \text{if } MA_t^{short} < MA_t^{long} \text{ and position} \geq 0 \\ \text{HOLD}, & \text{otherwise} \end{cases}$$

     Market order of size $Q$ placed in the direction of $a_t$.

  2. *RSI Agent (Mean Reversion)*

     Calculate the Relative Strength Index (RSI):

     $$RSI_t = 100 - \frac{100}{1 + RS}, \quad RS = \frac{\text{Average Gain}}{\text{Average Loss}}$$

     The trading action $a_t$ is:

     $$a_t = \begin{cases} \text{BUY}, & RSI_t < \theta_{\text{low}} \\ \text{SELL}, & RSI_t > \theta_{\text{high}} \\ \text{HOLD}, & \text{otherwise} \end{cases}$$

     Market order of size $Q$ placed in the direction of $a_t$ (buy low RSI, sell high RSI).

  3. *Breakout Agent (Channel Breakout)*

     Define breakout levels:

     $$R_t = \max(P_{t-L}, \ldots, P_{t-1}), \quad S_t = \min(P_{t-L}, \ldots, P_{t-1})$$

     The trading action $a_t$ is:

     $$a_t = \begin{cases} \text{BUY}, & P_t > R_t \\ \text{SELL}, & P_t < S_t \\ \text{HOLD}, & \text{otherwise} \end{cases}$$

     Market order of size $Q$ placed following $a_t$.

### F.2.7 DEEP LEARNING AGENT

- **Trading Objective**
  Predict the future direction of asset prices using a deep neural network trained on market indicators, and make trading decisions accordingly.

- **Inputs and Prediction**
  At each decision time $t$, the agent receives a vector of processed market indicators (e.g., RSI, MACD, spread, etc.), denoted by $x_t$. The agent uses a parameterized neural network $f_\theta(x_t)$ to produce logits corresponding to three classes: price *decrease*, *no change*, or *increase*.
  $$\hat{y}_t = f_\theta(x_t) \in \mathbb{R}^3, \quad \text{with action}_t = \arg\max_i \hat{y}_{t,i}$$

- **Decision Rule**
  Based on the predicted class:
    - If action$_t$ = increase: place a market buy order of size $Q$
    - If action$_t$ = decrease: place a market sell order of size $Q$
    - Otherwise: hold

- **Online Supervised Learning**
  The agent continuously collects experience tuples $(x_t, y_t)$, where $y_t$ is the realized future price direction after a fixed prediction horizon. These are stored in a training buffer, and used to fine-tune the model parameters $\theta$ periodically via stochastic gradient descent:
  $$\theta \leftarrow \theta - \eta \cdot \nabla_\theta \mathcal{L}(f_\theta(x), y)$$
  where $\mathcal{L}$ is the cross-entropy loss.

- **Learning Cycle**
  The agent alternates between:

  1. **Prediction:** Use the current model $f_\theta$ to make trading decisions
  2. **Labeling:** Wait until the prediction horizon expires to label old observations
  3. **Fine-Tuning:** Train $f_\theta$ on recent labeled samples from the buffer

### F.2.8 TIMEMIXER TRADING AGENT

- **Trading Objective**
  Predict the short-term price direction of the traded asset using a deep learning model (TimeMixer), and place market orders accordingly. The model is continuously fine-tuned online with newly observed data to improve predictive performance.

- **Information Source**
  Periodically requests a vector of standardized market indicators from a *DataCenterAgent*. These indicators may include measures of momentum, liquidity, order book imbalance, and normalized position holdings.

- **Decision Logic on Indicator Reception**
  At time $t$, when an indicator vector $x_t$ is received, the TimeMixer model outputs a trading signal $\hat{y}_t \in \{\texttt{Buy}, \texttt{Sell}, \texttt{Hold}\}$, leading to the following actions:
    - $\hat{y}_t = \texttt{Sell}$: predicted price decrease $\Rightarrow$ place $\texttt{MarketSell}(q)$
    - $\hat{y}_t = \texttt{Hold}$: predicted no significant change $\Rightarrow$ take $\texttt{NoAction}$
    - $\hat{y}_t = \texttt{Buy}$: predicted price increase $\Rightarrow$ place $\texttt{MarketBuy}(q)$

  where $q$ denotes the fixed order quantity.

- **Labeling and Online Training**
  Each observation $x_t$ is stored and labeled after a prediction horizon $\Delta T$ based on realized price movement. If the relative change exceeds an upper threshold $\theta_1$, the label is "up"; if it falls below a lower threshold $\theta_2$, the label is "down"; otherwise the label is "neutral". Experiences $(x_t, y_t)$ are stored in a replay buffer. At fixed intervals, the agent samples minibatches from the buffer and fine-tunes the model parameters $\theta$ via stochastic gradient descent.

- **Timing and Wake-up Frequency**
  The agent wakes up at stochastic intervals $\Delta t$ to request new indicators, and may schedule short-term wakeups while awaiting responses to maintain activity.

- **Execution Mechanism**
  All trades are submitted as **market orders** with fixed size to ensure immediate execution. The agent dynamically updates its cash and holdings according to executed trades.

### F.2.9 ITRANSFORMER TRADING AGENT

- **Trading Objective**
  Employ a simplified iTransformer model for supervised short-term price prediction. The agent uses predicted signals to submit market orders, while continuously fine-tuning the model with streaming market data.

- **Information Source**
  Periodically requests a vector of technical indicators (e.g., momentum, order book imbalance, liquidity spread, position features) from a *DataCenterAgent*. These indicators are normalized and transformed into input features for the iTransformer.

- **Decision Logic on Indicator Reception**
  Upon receiving an indicator vector $x_t$, the iTransformer outputs a trading signal $\hat{y}_t \in \{\texttt{Buy}, \texttt{Sell}, \texttt{Hold}\}$:

  - $\hat{y}_t = \texttt{Buy} \Rightarrow$ place $\texttt{MarketBuy}(q)$
  - $\hat{y}_t = \texttt{Sell} \Rightarrow$ place $\texttt{MarketSell}(q)$
  - $\hat{y}_t = \texttt{Hold} \Rightarrow$ take $\texttt{NoAction}$

  where $q$ denotes the fixed trade quantity.

- **Labeling and Online Training**
  Each observation is assigned a label after a prediction horizon $\Delta T$, based on realized price change relative to past levels. If the change exceeds an upper threshold $\theta_1$, the label is "Buy"; if below a lower threshold $\theta_2$, the label is "Sell"; otherwise it is "Hold". These labeled samples are stored in a replay buffer. At regular intervals, minibatches are sampled and the model parameters are fine-tuned using stochastic gradient descent.

- **Timing and Wake-up Frequency**
  The agent schedules wake-ups at randomized intervals to request new indicators. During waiting periods, short-term wakeups are used to retry or maintain activity.

- **Execution Mechanism**
  All trades are executed as **market orders** with fixed size to ensure immediate execution. The portfolio of cash and holdings is dynamically updated after each trade.

### F.2.10 INFORMED TRADER AGENTS

- **Trading Objective**
  Informed traders compare the market price with the fundamental value (as provided by an oracle). If the relative deviation exceeds a predefined threshold, they place orders according to their specific strategy.

- **Information Source**
  The agent observes the fundamental value from the oracle and retrieves the best bid and ask quotes from the order book maintained by the *ExchangeAgent*. From this information, the deviation ratio is computed:

$$d_t = \frac{p_t^{\text{market}} - p_t^{\text{fundamental}}}{p_t^{\text{fundamental}}}.$$

- **Decision Logic**
  The base class (`InformedTraderAgent`) provides a general mechanism for computing deviation and submitting orders. Subclasses override the `take_action()` method to implement distinct trading styles:

– **Conservative Informed Agent**
  Places **limit orders** when deviation exceeds the threshold, aiming to trade passively. Buys when the market is undervalued; sells when overvalued, but only if current positions allow.

– **Aggressive Informed Agent**
  Reacts with **market orders** immediately once deviation exceeds the threshold. Executes a buy when the asset is undervalued and a sell when overvalued.

– **Opportunistic Informed Agent**
  Evaluates available liquidity before trading. If liquidity on the opposite side of the book is thin, scales up the order size; otherwise uses standard quantity. Orders are submitted as market orders.

– **Scaled Informed Agent**
  Adjusts order size proportionally to the magnitude of deviation. Larger deviations lead to larger trade quantities, up to a capped multiple. Trades are placed using limit orders.

– **Spread-Aware Informed Agent**
  Chooses between limit and market orders depending on the bid-ask spread. If the spread is wide, prefers limit orders to capture better prices; if narrow, executes with market orders for immediacy.

• **Timing and Wake-up Frequency**
  The agent wakes up at randomized intervals during the market session to re-evaluate deviation, update its position, and potentially submit orders.

• **Execution Mechanism**
  Execution style depends on the subclass: passive (limit orders), aggressive (market orders), or hybrid (spread-aware or scaled). All order messages are routed through the exchange and logged for state tracking.

#### F.2.11 TECHNICAL TRADER AGENTS

• **Trading Objective**
  Technical traders base their trading decisions on historical price patterns and derived technical indicators. They aim to capture short-term price trends, reversals, or breakout movements according to their respective strategy.

• **Information Source**
  Each agent observes recent trade prices from the market and computes technical indicators such as moving averages, RSI, or price channel levels. These indicators are used to determine market trends or overbought/oversold conditions.

• **Decision Logic**
  The agents implement different trading styles:

  – **Momentum Agent**
    Uses a dual moving average crossover strategy. Buys when the short-term average crosses above the long-term average (uptrend) and sells when the short-term average crosses below the long-term average (downtrend). Orders are placed as market orders.

  – **RSI Agent**
    Applies the Relative Strength Index (RSI) for mean-reversion trading. Buys when the market is oversold (RSI below threshold) and sells when overbought (RSI above threshold). This agent also uses market orders for execution.

  – **Breakout Agent**
    Follows a channel breakout strategy. If the current price exceeds the recent high (resistance), the agent buys; if it drops below the recent low (support), it sells. Orders are executed immediately using market orders.

• **Timing and Wake-up Frequency**
  Agents wake up at randomized intervals to observe the market, compute indicators, and evaluate trading opportunities. The randomness introduces desynchronization among agents, avoiding artificial simultaneity.

- **Execution Mechanism**
  All agents place market orders based on signals derived from their respective indicators. Trade execution is logged, and positions are updated accordingly. Agents track cash, stock holdings, and portfolio value as part of their internal state.

### F.2.12 LLM-BASED TRADING AGENTS

LLMs integrate market indicators and news information to generate trading actions using different reasoning mechanisms.

- **LLM ReAct Trader**
  **Trading Objective:** Incorporate market indicators and news text into a ReAct reasoning process to produce immediate trading actions while logging interpretable reasoning.
  **State Observation:** At each decision time $t$, the agent observes a vector of processed market indicators $x_t$ and recent news $n_t$.
  **Decision Rule:** The agent performs ReAct reasoning (Reason $\rightarrow$ Act) to select an action $a_t \in \{\text{BUY, SELL, HOLD}\}$:
  $$a_t = \text{ReAct}(x_t, n_t)$$
  Market orders of fixed quantity $Q$ are placed according to $a_t$.
  **Logging:** Reasoning steps and intermediate analysis are stored for interpretability.

- **LLM CoT Trader**
  **Trading Objective:** Generate a chain-of-thought (CoT) reasoning process using indicators and news to improve decision quality.
  **State Observation:** At time $t$, observe $x_t$ and $n_t$.
  **Decision Rule:** Construct a sequence of reasoning steps:
  $$\text{CoT}_t = \{s_1, s_2, \ldots, s_k\}, \quad a_t = \text{FinalDecision}(\text{CoT}_t)$$
  Market orders of quantity $Q$ are executed based on the final action.

- **LLM Trading Trader**
  **Trading Objective:** Produce a structured trading plan (action + quantity) using indicators and news information.
  **State Observation:** Observe $x_t$ and $n_t$ at decision time $t$.
  **Decision Rule:** Generate a trading plan:
  $$\text{Plan}_t = \{\text{action}_t, \text{quantity}_t\} = \text{LLMTradingModel}(x_t, n_t)$$
  The agent executes market orders following the proposed plan.

- **LLM FinCon Trader**
  **Trading Objective:** Generate trading actions under financial constraints (e.g., maximum position limits), integrating market indicators and news.
  **State Observation:** At time $t$, observe $x_t$, $n_t$, and current holdings $h_t$.
  **Decision Rule:** Compute a constrained action:
  $$a_t, q_t = \text{ConstrainedDecision}(x_t, n_t, h_t, Q, h_{\max})$$
  Market orders are placed only if they respect financial constraints. Reasoning and planned quantity are logged.

## G    EXPERIMENTAL DETAILS AND RESULTS

### G.1    SIMULATION UNDER ARTIFICIAL SHOCKS

**Trading Price**    The left panel shows the average price trajectory under an upward shock injected at 13:00, while the right panel illustrates the dynamics under a downward shock injected at 12:00. In both cases, the blue line represents the mean path of simulated trade prices, and the shaded region denotes the variability across runs, measured as 95% CI above and below the mean. The upward shock leads to a persistent price increase, whereas the downward shock triggers a sharp drop followed by gradual decline, closely resembling stylized market reactions. These results further confirm that our framework produces plausible market responses under extreme exogenous disturbances, reinforcing the validity of the proposed ecological dynamics model.

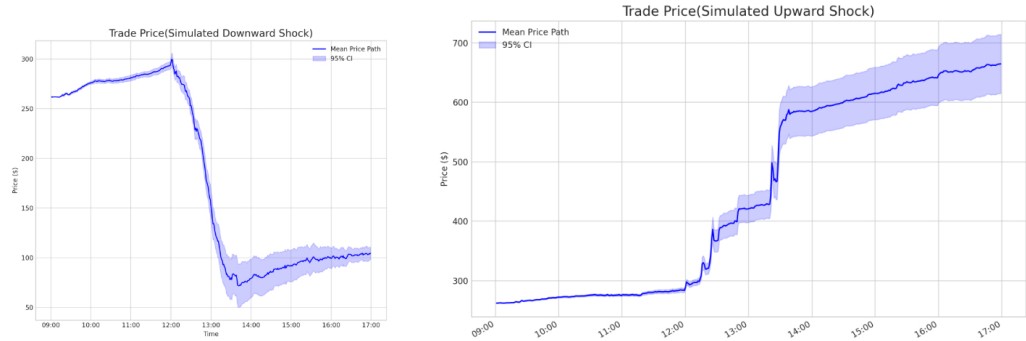

Figure 10: Impact of simulated exogenous shocks on trade prices. The blue line indicates the mean transaction price path, and the shaded area represents the $\pm 3\sigma$ confidence band across simulation runs.

### SHOCK-INDUCED NETWORK DYNAMICS

To complement the main analysis, we further investigate how external shocks reshape the structural interactions between strategies. Figures 11 and 12 present the evolution of co-occurrence matrices and mutual-information networks under a positive shock, while Figures 13 and 14 show the corresponding results under a negative shock. These views provide complementary evidence of robustness and reveal how alliances reorganize in response to shocks.

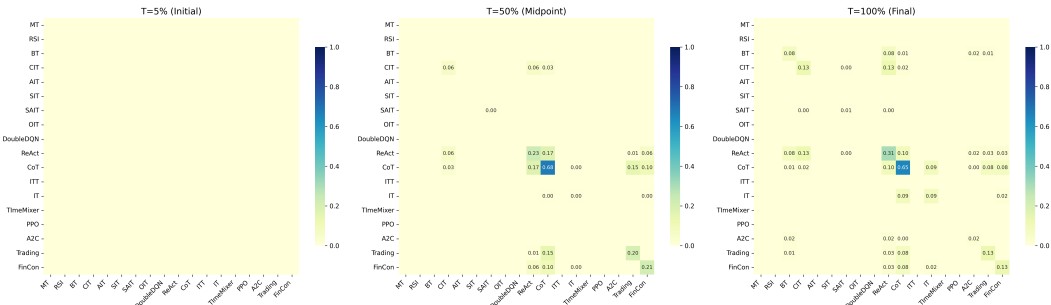

Figure 11: Evolution of strategy co-occurrence matrices under a positive shock. Values indicate the frequency with which pairs of strategies exceed the dominance threshold across different stages (initial, midpoint, final).

### INTRADAY NETWORK EVOLUTION (JULY 7, 2025)

To examine how ecological structures evolve within a normal trading day (without exogenous shocks), we track (i) the *co-occurrence* of dominant strategies and (ii) the *mutual-information (MI)* network among strategies on July 7, 2025. Time is normalized to the trading session ($T=0\%, 25\%, 50\%, 75\%, 100\%$). In the co-occurrence view, an entry $(i, j)$ records the frequency with which strategies $i$ and $j$ are simultaneously dominant (share $> 12\%$). In the MI network, node size reflects the dominance of a strategy and edge thickness encodes co-dependence strength.

**Detailed analysis.** **(Early,** $T=0\% - 25\%$**)** Co-occurrence is negligible and the MI network is sparse, indicating a fragmented ecology with weak coordination. Most strategies operate independently.

**(Mid-day,** $T=50\%$**)** The first *co-dominant pairs* emerge in the co-occurrence matrix, and the MI network becomes *hub-centric*: one or two strategies act as connectors linking otherwise disjoint clusters. This marks a regime shift from exploratory behavior to coordinated specialization.

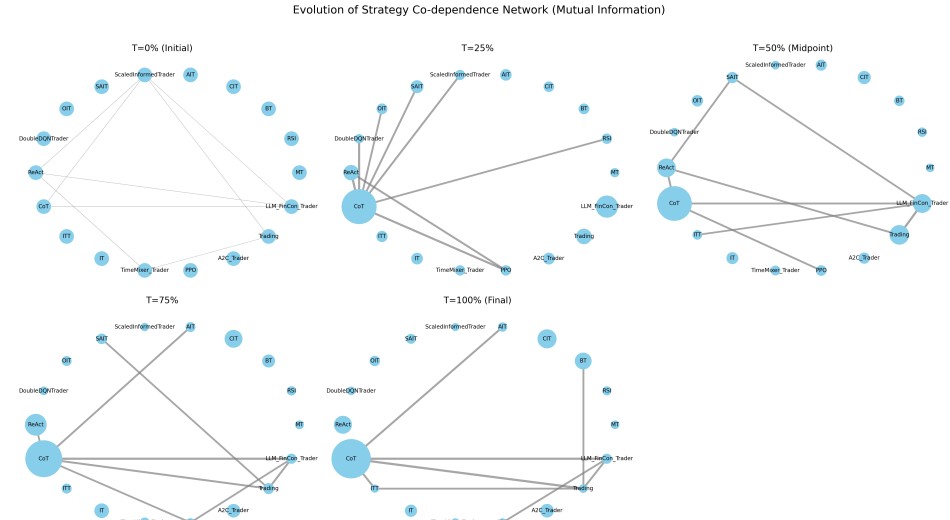

Figure 12: Evolution of mutual-information networks under a positive shock. Node size reflects strategy dominance, and edge thickness encodes co-dependence strength.

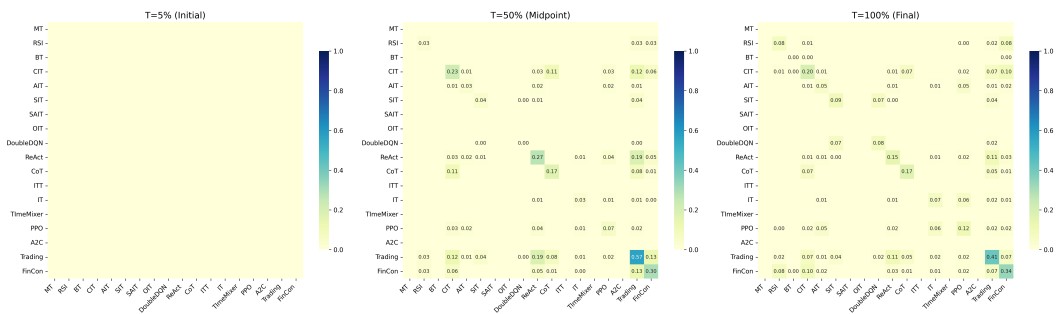

Figure 13: Evolution of strategy co-occurrence matrices under a negative shock. Compared to the positive case, the network re-concentrates more quickly, with fewer but stronger alliances emerging.

(**Late,** $T{=}75\%-100\%$) A small set of alliances persists and strengthens, while peripheral strategies decouple. The MI network shows *selective pruning*: superfluous links vanish and high-weight ties concentrate around the main hub(s). This coincides with the end-of-day re-concentration seen in population shares.

**Takeaways.** Across the day, the ecology moves from *diffuse* (low co-occurrence, sparse MI) to *organized* (stable alliances, hub-centric MI), then *stabilizes* toward the close. This intraday reorganization is consistent with our macro findings: exploration predominates early, while selection gradually consolidates alliances into a small, persistent backbone.

## G.2 ADDITIONAL INTRADAY TRAJECTORIES ACROSS MULTIPLE REAL-MARKET DAYS

In the main text (Figure 6), we evaluated FinEvo on two long real-market periods: *January–October 2022* (bear regime) and *October 2024–July 2025* (bull regime). These experiments already aggregate ecosystem statistics over **many months** of real data. However, due to space constraints, only a single representative intraday trajectory was shown in the main text. Therefore, we provide **additional intraday population-evolution trajectories** from multiple dates across both bull and bear markets. These plots demonstrate that FinEvo's ecological dynamics are not specific to a single trading day: dominance shifts, strategy turnover, and phase-change patterns remain consistent across a wide range of trading conditions.

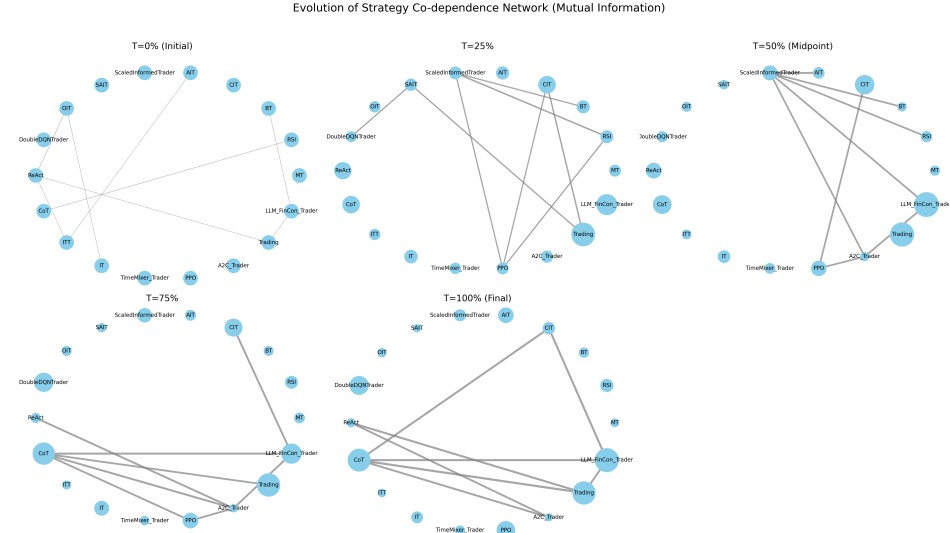

Figure 14: Evolution of mutual-information networks under a negative shock. The results highlight the collapse of pluralistic alliances and the rise of concentrated hubs dominated by LLM agents.

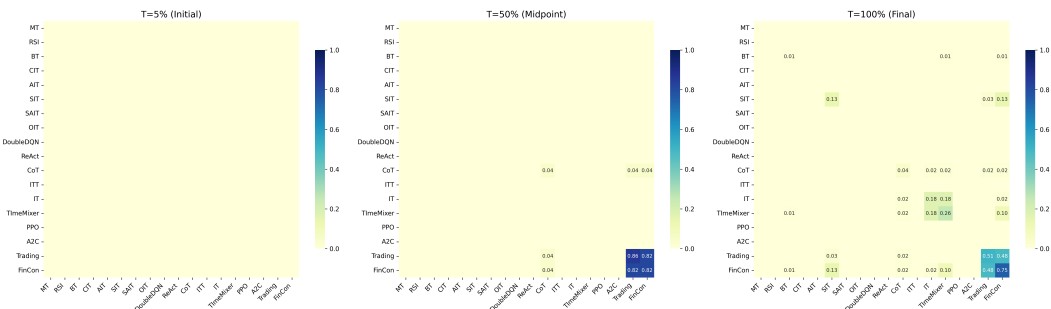

Figure 15: Intraday evolution of **co-occurrence** matrices on July 7, 2025 (dominance threshold >12%). Early in the day ($T=0\%$) the matrix is near-empty; mid-day ($T=50\%$) a few pairs begin to co-dominate; by the close ($T=100\%$) the structure consolidates around a small number of persistent alliances while many strategies remain peripheral.

These results confirm that FinEvo's real-data performance is **robust across days, months, and market regimes**,

### G.3    ROBUSTNESS TO PERTURBATIONS OF INITIAL AGENT PARAMETERS

To assess whether FinEvo's ecological dynamics depend sensitively on the initial agent-level hyperparameters, we conducted a *Perturbed-Parameter* robustness experiment. For each Monte Carlo run, all key hyperparameters (e.g., RL learning rates, window lengths for rule-based agents, and LLM prompting parameters) were randomly perturbed by $\pm20\%$ around their baseline values, sampled from a uniform distribution. This produces substantially different heterogeneous initializations for every simulation.

Across both bull and bear regimes, the perturbed-parameter version closely matches the baseline across all stylized-fact metrics. This confirms that heavy tails, downside asymmetry, Sharpe dispersion, and volatility clustering are **stable emergent properties of FinEvo's evolutionary dynamics**, not artifacts of specific hyperparameter choices.

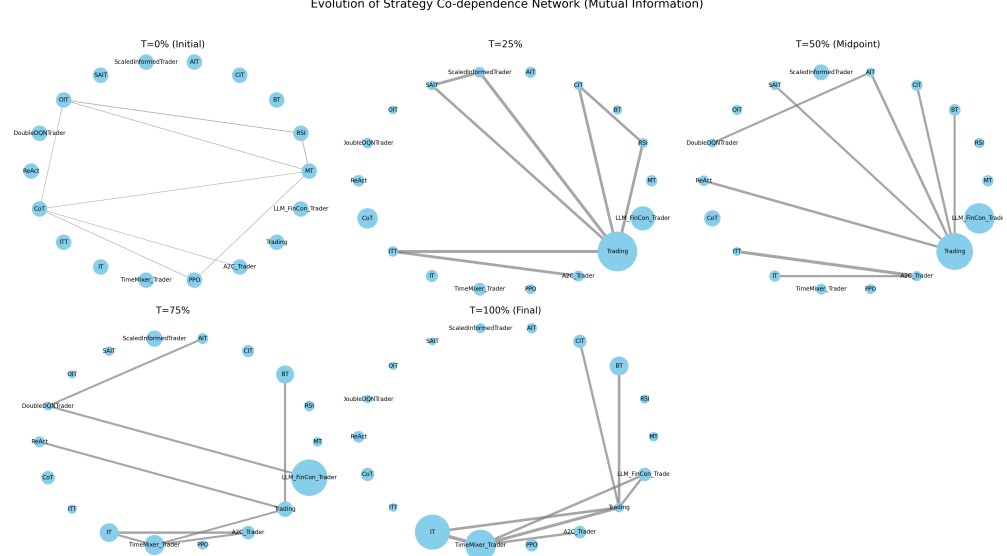

Figure 16: Intraday evolution of the **mutual-information** network on July 7, 2025. Node size indicates dominance; edge thickness indicates co-dependence strength. The network transitions from a sparse, weakly connected configuration in the morning to a hub-centric structure by mid-day, followed by selective pruning and stabilization toward the close.

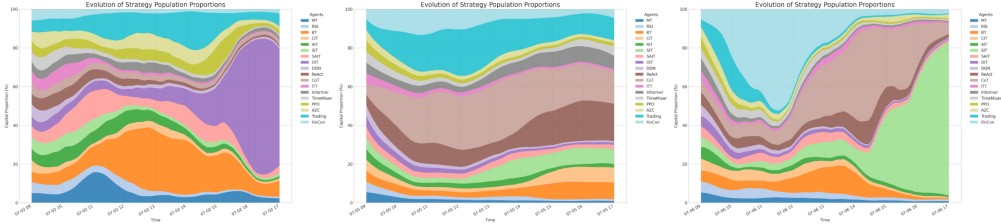

Figure 17: **Intraday population evolution on 2025-07-02, 07-05, and 07-06.**

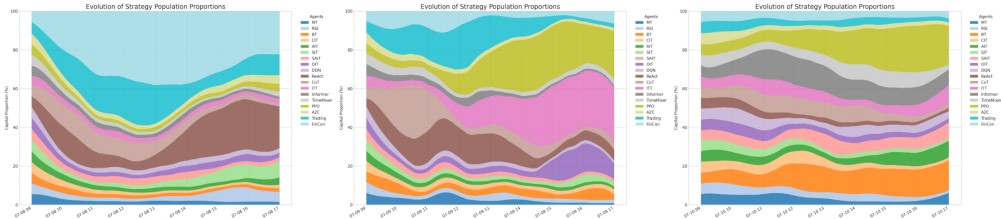

Figure 18: **Intraday population evolution on 2025-07-08, 07-09, and 07-10.**

Table 8: Robustness under perturbed agent parameters (128 runs). Values report mean ± 95% CI. Stylized-fact metrics remain stable under ±20% perturbations.

| Regime | Model | Ex. Kurtosis$_{mean}$↑ | Skewness$_{mean}$↓ | Sharpe Disp.$_{mean}$↑ | Vol-of-Vol$_{mean}$↑ |
|---|---|---|---|---|---|
| Bullish | FinEvo | $3.163 \pm 0.138$ | $-1.532 \pm 0.129$ | $1.669 \pm 0.061$ | $0.008 \pm 0.0004$ |
| | FinEvo (Perturbed) | $3.141 \pm 0.152$ | $-1.596 \pm 0.138$ | $1.638 \pm 0.067$ | $0.008 \pm 0.0004$ |
| Bearish | FinEvo | $4.259 \pm 0.172$ | $-1.793 \pm 0.149$ | $1.732 \pm 0.089$ | $0.013 \pm 0.0007$ |
| | FinEvo (Perturbed) | $4.196 \pm 0.181$ | $-1.692 \pm 0.157$ | $1.684 \pm 0.094$ | $0.012 \pm 0.0008$ |

## G.4 ROBUSTNESS TO CORRELATED, HEAVY-TAILED SHOCKS

In the main text, our analytical results are derived under the standard assumption of independent Gaussian shocks. To verify that our empirical findings are not an artifact of this simplification, we conduct an additional robustness experiment in which payoff and environmental shocks are both *correlated* and *heavy-tailed* Concretely, we modify the noise driving the value processes $V_k(t)$ as follows. At each time step $t$, instead of drawing independent standard normal increments, we construct strategy-level shocks

$$\epsilon_k(t) = \rho Z_{\text{common}}(t) + \sqrt{1 - \rho^2} Z_{k,\text{idio}}(t), \tag{11}$$

where $\rho \in (0, 1)$ controls the correlation strength (we use $\rho = 0.5$ in our experiments), $Z_{\text{common}}(t)$ is a *common macro factor* shared by all strategies, and $Z_{k,\text{idio}}(t)$ is an idiosyncratic component specific to strategy $k$. Both $Z_{\text{common}}(t)$ and $Z_{k,\text{idio}}(t)$ are drawn from a standardized Student-$t$ distribution with $\nu$ degrees of freedom (we use $\nu = 5$), rescaled to have unit variance. This yields positively correlated, heavy-tailed shocks at the strategy level:

$$\text{Cov}\big(\epsilon_k(t), \epsilon_\ell(t)\big) = \rho^2 \quad (k \neq \ell), \tag{12}$$

in contrast to the independent Gaussian setting in the main analysis. We then drive the value process of each strategy with these correlated heavy-tailed shocks:

$$dV_k(t) = \lambda_k\big(m_t - V_k(t)\big)\, dt + \nu_k\, \epsilon_k(t)\, \sqrt{dt}, \tag{13}$$

and, for the environmental shock, we take

$$\eta_t = Z_{\text{common}}(t), \tag{14}$$

so that strategy-level payoffs are explicitly correlated both with each other and with the environment shock. We rerun the main set of simulations under this correlated, heavy-tailed specification using the same market configurations as in Section 3. The resulting markets continue to exhibit (i) fat-tailed and negatively skewed PnL distributions, (ii) pronounced volatility clustering and elevated volatility-of-volatility, and (iii) similar ecological phase changes in the strategy population. Quantitatively, the stylized-fact metrics change only moderately; a detailed comparison is reported in Table 9. These results confirm that the independence/normality assumptions are primarily used for analytical tractability in the SDE derivation and are not critical for the qualitative behavior of FinEvo.

Table 9: Robustness to correlated, heavy-tailed shocks (128 runs). Values report mean $\pm$ 95% CI.

| Regime | Model | Ex. Kurtosis$_{\text{mean}}\uparrow$ | Skewness$_{\text{mean}}\downarrow$ | Sharpe Disp.$_{\text{mean}}\uparrow$ | Vol-of-Vol$_{\text{mean}}\uparrow$ |
|---|---|---|---|---|---|
| Bullish | FinEvo | $3.163 \pm 0.138$ | $-1.532 \pm 0.129$ | $1.669 \pm 0.061$ | $0.008 \pm 0.0004$ |
| | FinEvo (Correlated) | $5.278 \pm 0.151$ | $-1.632 \pm 0.133$ | $1.727 \pm 0.075$ | $0.011 \pm 0.0007$ |
| Bearish | FinEvo | $4.259 \pm 0.172$ | $-1.793 \pm 0.149$ | $1.732 \pm 0.089$ | $0.013 \pm 0.0007$ |
| | FinEvo (Correlated) | $6.399 \pm 0.214$ | $-1.932 \pm 0.175$ | $1.844 \pm 0.101$ | $0.015 \pm 0.0009$ |

## G.5 CROSS-ASSET ROBUSTNESS

**Setup.** FinEvo is designed to be asset-agnostic: the evolutionary SDE and CDA mechanism do not rely on any asset-specific assumption. To examine robustness across heterogeneous markets, we instantiate the same FinEvo ecology under four widely studied equities—AAPL, TSLA, MSFT, and XOM. Across all assets, the evolutionary parameters $(\beta, \mu, \gamma)$, agent library, value-learning dynamics, and market microstructure remain *identical*. The only asset-dependent components are the empirical volatility scale and the asset-specific news schedule used to parameterize exogenous shocks. Thus, each asset represents the *same evolutionary system under different empirical environments*, rather than a separately calibrated model.

**Results.** In our experiments, the same FinEvo configuration produces qualitatively similar ecological signatures across AAPL, TSLA, MSFT, and XOM (Table10), suggesting that the framework is not tightly tied to any single underlying asset. While we only test equities here, the evolutionary SDE and CDA mechanism do not rely on equity-specific assumptions, so in principle the same formalism could be instantiated for other electronically traded markets (e.g., futures, FX, cryptocurrencies) with appropriate data and calibration.

Table 10: Cross-asset robustness of FinEvo. Metrics are reported as mean $\pm$ 95% CI over 128 runs.

| Asset | Ex. Kurtosis | Skewness | Sharpe Disp. | Vol-of-Vol |
|---|---|---|---|---|
| AAPL | $4.354 \pm 0.41$ | $-1.656 \pm 0.18$ | $1.633 \pm 0.12$ | $0.011 \pm 0.0015$ |
| TSLA | $\mathbf{5.578 \pm 0.53}$ | $\mathbf{-2.344 \pm 0.24}$ | $\mathbf{1.975 \pm 0.18}$ | $\mathbf{0.015 \pm 0.0023}$ |
| MSFT | $3.176 \pm 0.35$ | $-0.967 \pm 0.15$ | $1.158 \pm 0.09$ | $0.008 \pm 0.0016$ |
| XOM | $3.741 \pm 0.38$ | $-0.573 \pm 0.12$ | $1.392 \pm 0.17$ | $0.008 \pm 0.0011$ |

## G.6 EXPERIMENT CONFIGURATION

### PRICE MATCHING MECHANISM (CONTINUOUS DOUBLE AUCTION)

**Order book and priority.** We adopt a continuous double auction with price–time priority. At time $t$, let the best bid/ask be $b_t$ and $a_t$ with spread $s_t = a_t - b_t$. Depth at level $\ell$ on the bid/ask sides are $B_t(\ell), A_t(\ell)$, tick size $\delta p$. Orders are queued FIFO within price.

**Order types.** Agents submit *limit orders* $(p, q)$ (positive $q$ for buy, negative $q$ for sell) or *market orders* ($p = \pm\infty$). A limit order is *marketable* if $p \geq a_t$ (buy) or $p \leq b_t$ (sell).

**Matching rule and fills.** Incoming marketable quantity $q > 0$ (buy) consumes the ask ladder $\{(A_t(1), a_t), (A_t(2), a_t + \delta p), \dots\}$ until filled or book exhausted. If the order walks $L$ levels with per-level trades $v_\ell$ at prices $p_\ell$, the volume-weighted execution price (VWAP) is

$$\bar{p}_t \;=\; \frac{\sum_{\ell=1}^{L} v_\ell\, p_\ell}{\sum_{\ell=1}^{L} v_\ell}, \qquad \sum_{\ell=1}^{L} v_\ell \;=\; q_{\text{filled}} \leq q.$$

Any unfilled remainder $\Delta q = q - q_{\text{filled}}$ (for a limit order) is posted at its limit price $p$ on the respective side with timestamp $t$. The *last trade price* $P_t$ is set to the price of the final match within the event; the book then updates depths and best quotes.

**Mid/micro-price and slippage.** Define mid-price $m_t = \frac{a_t + b_t}{2}$ and a depth-weighted micro-price $\tilde{m}_t = \frac{A_t(1)\, b_t + B_t(1)\, a_t}{A_t(1) + B_t(1)}$. For a buy, the instantaneous slippage (vs. pre-trade mid) is

$$\text{Slip}_t^{\text{buy}} \;=\; \bar{p}_t - m_{t^-}, \qquad \text{Slip}_t^{\text{sell}} \;=\; m_{t^-} - \bar{p}_t.$$

**Trading frictions (costs).** We account for realistic frictions via a per-trade proportional cost $c = 0.5\%$ (including slippage/impact setting). For a trade of notional $N_t = |\bar{p}_t\, q_{\text{filled}}|$, the fee component is $c\, N_t$.

**Cash, inventory, and mark-to-market.** For agent $k$, let inventory be $x_{k,t}$ and cash be $C_{k,t}$. A filled *buy* of size $q > 0$ at $\bar{p}_t$ updates

$$C_{k,t} \leftarrow C_{k,t} - \bar{p}_t\, q - c\, |\bar{p}_t q|, \qquad x_{k,t} \leftarrow x_{k,t^-} + q.$$

A *sell* symmetrically yields $C_{k,t} \leftarrow C_{k,t} + \bar{p}_t\, |q| - c\, |\bar{p}_t q|$, $x_{k,t} \leftarrow x_{k,t^-} - |q|$. Mark-to-market portfolio value at event-time $t$ is

$$V_{k,t} \;=\; C_{k,t} + x_{k,t}\, m_t.$$

**PnL and return.** Incremental PnL is $\Delta\text{PnL}_{k,t} = V_{k,t} - V_{k,t^-}$. Cumulative return over a session is

$$\text{Ret}_k \;=\; \frac{V_{k,T} - V_{k,0}}{V_{k,0}}.$$

We report $\text{Ret}_k$ with a 95% confidence interval (CI) across runs: $\text{CI}_{95} = \bar{r} \pm 1.96\, \hat{\sigma}/\sqrt{n}$.

**Sharpe, drawdown, turnover.** Using event- or bar-level returns $\{r_{k,t}\}$, the (annualized) Sharpe is

$$\text{Sharpe}_k \;=\; \frac{\mu(r_{k,t})}{\sigma(r_{k,t})}\, \sqrt{A},$$

with $A$ the annualization factor (e.g., bars-per-year). Max drawdown is $\text{MDD}_k = \max_{t \leq u}\big(1 - \frac{V_{k,u}}{\max_{s \leq t} V_{k,s}}\big)$. Turnover over the session is measured as traded notional over average equity:

$$\text{Turnover}_k = \frac{\sum_t |\overline{p}_t\, q_{k,t}|}{\frac{1}{T}\sum_t V_{k,t}}.$$

**Interpretation.**

- *Price–time priority* ensures fair queueing: better prices fill first; within a price level, earlier timestamps precede later ones.
- *VWAP execution* captures multi-level consumption and embeds microstructure slippage relative to $m_{t^-}$.
- *0.5% cost per trade* penalizes overtrading and brings profitability closer to realistic conditions.
- *Mark-to-market PnL* at mid links inventory risk to contemporaneous liquidity via $m_t$ (or $\tilde{m}_t$ if desired).
- *Sharpe/MDD/Turnover* jointly describe risk-adjusted performance, path risk, and trading intensity, complementing raw returns.

### G.6.1 EXPERIMENT SETUP

All experiments are conducted using the simulation market engine, which provides a continuous double auction market with realistic order book dynamics (see Appendix G.6 for details of the price matching mechanism). We extend the environment with three key components to capture eco-evolutionary dynamics:

1. **News Release Line.** External information shocks are injected via a news-release channel that delivers common signals to all agents at scheduled times. Shocks can be positive or negative and directly affect agent beliefs and trading aggressiveness.

2. **Multi-Metric Distribution Mechanism.** Beyond simple price updates, the simulation tracks and redistributes capital based on multiple metrics including realized PnL, volatility-adjusted returns, and strategy diversity. This allows performance pressure, innovation, and environmental perturbation to jointly influence ecological adaptation.

3. **Evolutionary Rules.** We implement the replicator–mutator dynamics introduced in Section 2. At each epoch (set to 5 minutes of simulated time), agent population shares are updated according to selection strength $\beta$, mutation rate $\mu$, and perturbation intensity $\sigma$. Capital is then reallocated consistently with these updated proportions, ensuring both ecological realism and conservation of market value.

Unless otherwise specified, we run each configuration with 128 Monte Carlo seeds with a 8-hour trading session per day. Transaction costs (including slippage) are set to $0.5\%$ per trade. All agents start with equal initial capital and interact in the same synthetic market environment.

### G.6.2 AGENT PARAMETERS

For clarity, we summarize the design of the infrastructure agents used in our experiments. Since the trading logic of each agent type has already been detailed in Appendix F.2.1, here we only describe their functional roles and key configuration parameters.

**DataCenterAgent.** Acts as the unique global data hub and indicator engine. It collects trades and liquidity snapshots from the exchange, computes a wide set of microstructure and technical indicators (e.g., bid–ask spread, order book imbalance, RSI, MACD, ATR, Bollinger Bands), and maintains historical records via bounded deques to ensure memory efficiency. Key parameters include:

- *wakeup frequency:* 5s (sampling interval for updates).

- *max history length:* 200 ticks.
- *minimum history for service:* 30 ticks.
- *output:* indicator_history.pkl and liquidity_history.pkl.

**EventBusAgent.** Implements a centralized publish/subscribe hub. It manages subscriptions from other agents and forwards broadcast events (e.g., news). The design is lightweight: subscribers are stored in a set, ensuring de-duplication, and events are distributed in real time to all registered agents.

**NewsAgent.** Publishes exogenous events (e.g., news shocks) according to a pre-specified schedule. At each scheduled time, the event is broadcast via the EventBusAgent. The schedule is defined as a sorted list of $(t, \text{event})$ pairs, ensuring reproducibility and precise timing of information shocks.

**NoiseTraderAgent.** A baseline stochastic trader that injects random liquidity into the market. At each wakeup, it places either a market order (immediate execution) or a limit order (near the mid-price with small random offsets), each with 50% probability. **Key parameters:** traded symbol , random order size (10–100 shares), wakeup interval (55–65 seconds, uniformly drawn).

**MarketMakerAgent.** Provides liquidity symmetrically on both sides of the order book. At each wakeup, it cancels its resting orders and posts new ones across multiple price levels, ensuring spread coverage. It can operate in two modes: (i) polling mode (places orders around mid-price using current spread), (ii) subscription mode (subscribes to order book updates and allocates volumes according to a predefined level-quote distribution). **Key parameters:** initial cash = 10M, order size range = [10, 50], wakeup frequency = 5s, number of levels = 1–5, order size split follows DEFAULT_LEVELS_QUOTE_DICT.

**NewsReactionAgent.** An event-driven trader that reacts to exogenous news released by the NewsAgent via the EventBus. Upon receiving a broadcast, it submits a market order consistent with the sentiment (buy if POSITIVE/UPGRADE, sell if NEGATIVE/DOWNGRADE). **Key parameters:** initial cash = 1M, trade size = 10 shares per reaction, wakeup frequency = 10s (used only for subscription maintenance).

**MomentumAgent.** A trend-following trader based on moving average crossovers. It buys when the short-term average exceeds the long-term average, and exits when the reverse occurs. **Key parameters:** initial cash = 1M, short window = 5, long window = 20, trade size = 10, wakeup interval = 55–65s.

**RSIAgent.** A mean-reversion trader using the Relative Strength Index (RSI). It buys when RSI drops below the oversold threshold and sells when it exceeds the overbought threshold. **Key parameters:** initial cash = 1M, RSI window = 14, thresholds = [30, 70], trade size = 10, wakeup interval = 55–65s.

**BreakoutAgent.** A channel breakout trend-follower. It buys when the current price exceeds the recent maximum and sells when it falls below the recent minimum. **Key parameters:** initial cash = 1M, lookback window = 50, trade size = 10, wakeup interval = 55–65s.

**InformedTraderAgent (Base).** This family of agents trades on deviations between market price and fundamental value as observed from the oracle. If deviation exceeds a threshold, subclasses determine execution style. **Key parameters:** initial cash = 1M, trade size = 10, deviation threshold = 2%, wakeup interval = 2–3 min.

- **Conservative Informed.** Places passive limit orders when value deviations are large.
- **Aggressive Informed.** Executes immediate market orders upon detecting deviations.
- **Opportunistic Informed.** Adapts trade size to available liquidity (e.g., larger buys if ask depth is thin).
- **Scaled Informed.** Scales order size proportionally with deviation magnitude (up to 3x baseline).

- **Spread-Aware Informed.** Chooses between limit and market orders depending on the bid–ask spread width.

**DoubleDQNTradingAgent.** An online reinforcement learning trader that implements Double DQN with experience replay. The state space is discretized via RSI, order book imbalance (OBI), and bid–ask spread bins. The agent learns Q-values for three actions (HOLD, BUY, SELL) with periodic target network updates. **Key parameters:** learning rate $= 10^{-3}$, discount $= 0.9$, $\epsilon = 0.1$, buffer size $= 1000$, batch $= 32$, target update $= 50$ steps, trade size $= 100$, wakeup interval $= 2$–$3$ min.

**LLM Agents (CoT, ReAct, Trading).** Our generic LLM-based agents rely on structured prompting to integrate technical indicators, news events, and portfolio states. They differ in reasoning style: *CoT* uses step-by-step deliberation, *ReAct* interleaves reasoning with action, and *Trading* employs a hybrid evidence-aggregation workflow. When a news event arrives, the agents immediately request indicators and issue a trading decision; in the absence of news, they wake up on randomized intervals (5–15 minutes) to avoid synchronization artifacts. Trade sizes are discretized into multiples of 10 units, ensuring reproducibility and comparability across agents. These design choices keep agents responsive to exogenous shocks while preserving heterogeneous but tractable behavior, and robustness checks confirm stability under alternative sampling intervals and trade granularities.

**iTransformerTradingAgent.** A supervised-learning trader using the *iTransformer* architecture. It predicts short-term price direction (down, neutral, up) from streaming indicators, with labels generated via delayed price changes. The model is fine-tuned online using a replay buffer. **Key parameters:** input features $= \{$RSI, MACD, spread, OBI, position$\}$, output $= 3$ actions, buffer size $= 1000$, batch $= 32$, train frequency $= 10$, trade size $= 10$, wakeup interval $= 2$–$3$ min.

**InformerTradingAgent.** A supervised agent based on the *Informer* architecture for time-series forecasting. It takes streaming indicators from the DataCenter and predicts future price directions (down, flat, up). Predictions are used to place trades, while labels are generated ex-post using realized returns, enabling online fine-tuning with a replay buffer. **Key parameters:** input $= \{$RSI, MACD, spread, OBI, position$\}$, prediction horizon $= 10$ min, buffer $= 1000$, batch $= 32$, train freq $= 10$, trade size $= 10$, wakeup interval $= 2$–$3$ min.

**TimeMixerTradingAgent.** A supervised agent using the *TimeMixer* sequence model, similarly trained online. It processes technical features and position state, predicts directional moves, executes trades, and fine-tunes periodically. **Key parameters:** input $= \{$RSI, MACD, spread, OBI, position$\}$, hidden dim $= 64$, 2 layers, buffer $= 1000$, batch $= 32$, train freq $= 10$, trade size $= 10$, wakeup interval $= 2$–$3$ min.

**RL Agents (A2C, PPO).** Our reinforcement learning agents adopt canonical hyperparameters commonly used in the financial RL literature. For instance, the discount factor is fixed at $\gamma = 0.99$, entropy coefficient $0.01$, and value coefficient $0.5$, which follow standard practice in continuous control benchmarks. Training updates are based on short $n$-step trajectories ($n = 20$), balancing responsiveness and stability under noisy market feedback. Action space is deliberately simplified to $\{$Buy, Sell, Hold$\}$ with fixed trade size (10 units), ensuring comparability across agents. We further verified that moderate perturbations of these hyperparameters do not materially change the qualitative dynamics.

**LLM FinCon (Risk-Aware).** In addition to evidence-based reasoning, FinCon incorporates a lightweight risk module. We approximate Conditional Value-at-Risk (CVaR) using the last 50 realized returns and flag a risk alert when estimated CVaR falls below $-2\%$. This threshold is chosen empirically as a conservative proxy for downside fragility. To avoid overfitting to a single specification, we verified robustness across alternative window sizes (20–100) and thresholds ($-1\%$ to $-3\%$), which yield consistent qualitative patterns. When a risk alert is active, trade sizes are scaled down to reflect more cautious behavior.

**Population Manager.** The ecological dynamics are governed by a stochastic replicator–mutator mechanism. We adopt parameters within standard ranges of evolutionary game theory: selection

strength $\beta = 2.0$, mutation rate $\mu = 0.05$, and environmental perturbation $\sigma = 0.1$. Strategy-specific Ornstein–Uhlenbeck parameters $(\lambda, \nu)$ are used to smooth empirical payoffs, analogous to mean-reverting fitness dynamics in ecological modeling. The Dirichlet prior $\alpha = 10$ encourages moderate baseline diversity. Although these values are not calibrated to a specific market, qualitative behaviors—such as coexistence, dominance cycles, and extinction events—are robust under moderate perturbations of $(\beta, \mu, \sigma, \alpha)$, as confirmed by sensitivity checks.

### G.6.3 ANALYSIS METRICS

Our evaluation framework adopts a micro–meso–macro structure, with metrics defined as follows.

**Micro (Individual Performance and Diversity).** We track each agent's financial performance and survival:

- **Return**: $\text{Return}_i = \frac{V_i^{\text{final}} - V_i^{\text{init}}}{V_i^{\text{init}}}$, with $95\%$ confidence intervals via bootstrapping.

- **Sharpe ratio**: $\text{Sharpe}_i = \frac{\mathbb{E}[r_{i,t}]}{\sigma(r_{i,t})}$.

- **Maximum drawdown**: $\text{MaxDD}_i = \max_{t<u} \frac{P_{i,t} - P_{i,u}}{P_{i,t}}$.

- **Turnover**: $\text{Turnover}_i = \frac{\sum_t |q_{i,t}^{buy} + q_{i,t}^{sell}|}{\sum_t q_{i,t}^{pos}}$, measuring trading intensity.

- **Win rate**: fraction of runs in which an agent achieves positive return or high composite score.

**Meso (Population Diversity and Interaction Structures).** We evaluate intermediate-level ecological structure:

- **Concentration (HHI)**: $\text{HHI} = \sum_k x_k^2$, where $x_k$ is the population share of strategy $k$.
- **Strategy entropy**: Based on the distribution of discrete agent actions $\mathcal{A} = \{\text{Market\_buy}, \text{Limit\_buy}, \text{hold}, \text{Limit\_sell}, \text{Market\_sell}\}$, we compute at each time step $t$ the proportion $p_i(t)$ of agents in each action state. The Shannon entropy is then

$$H(p(t)) = -\sum_{i=1}^{5} p_i(t) \log_2 p_i(t).$$

  High $H$ indicates diverse and disordered activity, while low $H$ indicates homogenization (e.g., herding on a single action).
- **Modularity**: graph-based clustering strength in the correlation network of strategies.
- **Synergy vs. antagonism**: average signed correlation between strategy pairs, visualized as heatmaps.
- **Co-occurrence and mutual information**: frequency of joint dominance and dependency structures between strategies, reflecting alliance formation and reconfiguration.

**Macro (System Volatility and Regime Shifts).** We decompose aggregate fluctuations into three components:
$$\text{Var}(\Delta x_k) = V_{\text{selection}} + V_{\text{innovation}} + V_{\text{perturbation}},$$
corresponding to performance-driven selection pressure, innovation and social mixing, and environmental perturbations, respectively. We further quantify regime instability via the number of **phase changes** (frequency of dominant-strategy transitions).

**Phase Change Metric** We provide the formal definition of the *Phase Change* metric used in the empirical analyses.

Let
$$X_t = (x_{1,t}, \dots, x_{K,t})$$
denote the population share vector over the $K$ strategy classes at time $t$. Define the set of dominant strategies as
$$\mathcal{D}(t) = \left\{ k \; : \; x_{k,t} = \max_j x_{j,t} \right\}.$$

**Definition.** A *Phase Change* occurs at time $t$ whenever the identity of the dominant strategy set changes:

$$\boxed{\text{PhaseChange}(t) = 1 \quad \text{iff} \quad \mathcal{D}(t) \neq \mathcal{D}(t^-)}$$

and PhaseChange$(t) = 0$ otherwise.

**Distributional Statistics (System-Level).** Let $\{r_t\}_{t=1}^T$ denote the system-level return series (e.g., aggregate market return). We report four key distributional metrics that are standard in the stylized-fact literature.

- **Excess kurtosis (tail heaviness).** Financial returns are empirically leptokurtic and strongly non-Gaussian: their distributions exhibit more mass in the tails and around the mean than a normal distribution with the same variance (Engle, 1995; Bollerslev, 1986; Engle & Ng, 1993). Excess kurtosis measures how heavy the tails are relative to the Gaussian benchmark.

$$\text{ExKurt}(r) = \frac{1}{T}\sum_{t=1}^T \left(\frac{r_t - \mu_r}{\sigma_r}\right)^4 - 3, \quad \mu_r = \tfrac{1}{T}\sum r_t, \quad \sigma_r^2 = \tfrac{1}{T}\sum (r_t - \mu_r)^2.$$

  A Gaussian distribution has ExKurt $= 0$; positive values indicate fat tails (large shocks occur more often than under a normal model), which correspond to higher crash risk and jump-like behavior in returns.

- **Skewness (asymmetry of the return distribution).** Skewness captures whether large positive moves and large negative moves are equally likely. Many equity markets exhibit negative skewness, reflecting a higher probability of large downside moves (crashes) than large upside jumps (Barberis et al., 1998).

$$\text{Skew}(r) = \frac{1}{T}\sum_{t=1}^T \left(\frac{r_t - \mu_r}{\sigma_r}\right)^3.$$

  Values close to $0$ correspond to symmetric returns; Skew$(r) < 0$ indicates a heavy left tail (downside risk dominates), while Skew$(r) > 0$ indicates a heavy right tail.

- **Sharpe dispersion (heterogeneity of strategic performance).** While excess kurtosis and skewness are computed at the market level, Sharpe dispersion measures how differently individual strategy classes perform within the same environment. Let $S_k$ denote the Sharpe ratio of strategy class $k$, and $\bar{S} = \tfrac{1}{K}\sum_{k=1}^K S_k$ their cross-sectional mean. Then

$$\text{SharpeDisp} = \sqrt{\frac{1}{K}\sum_{k=1}^K (S_k - \bar{S})^2}.$$

  A low SharpeDisp means most strategies earn similar risk-adjusted returns (the environment does not strongly discriminate between them), whereas a high SharpeDisp indicates that the ecosystem sharply separates robust "winners" from fragile "losers" and thus provides a stringent stress test for trading rules.

- **Vol-of-Vol (instability and clustering of volatility).** Volatility in financial markets is time-varying and exhibits persistent clustering: periods of high volatility tend to be followed by high volatility, and calm periods by calm periods (Andersen et al., 2003). To capture this, we first construct a rolling-window estimator of volatility (e.g., with window size $W$):

$$\sigma_t = \sqrt{\frac{1}{W}\sum_{i=t-W+1}^t (r_i - \mu_t)^2}, \quad \mu_t = \tfrac{1}{W}\sum_{i=t-W+1}^t r_i.$$

  We then measure the standard deviation of this volatility process:

$$\text{VoV} = \sqrt{\frac{1}{T-W}\sum_{t=W}^T (\sigma_t - \bar{\sigma})^2}, \qquad \bar{\sigma} = \tfrac{1}{T-W}\sum_{t=W}^T \sigma_t.$$

  Larger VoV indicates more unstable volatility and stronger volatility clustering (frequent transitions between calm and turbulent regimes), whereas smaller VoV corresponds to a more stationary, homoskedastic market.

In heterogeneous-agent and agent-based models, the accepted evaluation philosophy is not to fit price trajectories, but to examine whether the model endogenously generates the Stylized Facts observed in real markets(Lux & Marchesi, 1999; LeBaron, 2006; Hommes, 2006). Theoretical and microstructure studies further emphasize that these non-Gaussian features arise from interaction among market participants, not exogenous noise or curve fitting (Bouchaud et al., 2009). Thus, reproducing Stylized Facts is the most meaningful and widely accepted measure of "closeness" to real financial behavior.

**Economic interpretation.** This metric captures a structural reorganization of the market ecology rather than a statistical effect. It is distinct from (i) volatility clustering, which is a price-level phenomenon, and (ii) strategy regime switching, which typically refers to a micro-level change within a single agent. A Phase Change reflects a macro-level transition in the ecosystem equilibrium, driven by the evolutionary forces of the FinEvo SDE (selection, innovation, perturbation).

**Trading indicators observed by agents.** To ensure realism, agents condition their actions on standard microstructure and technical signals. *Liquidity and flow.* Bid–ask spread, order-book imbalance (OBI), market depth (buy/sell), and tick-rule trade flow imbalance (TFI). *Momentum and oscillators.* ROC, RSI, MACD (line, signal, histogram), Aroon (up/down), KDJ, and Williams-%R. *Volatility and OHLC-based.* ATR, Bollinger Bands (upper/middle/lower), and ADX/DMI (ADX, +DI, −DI). These features ground agent behavior in market observables rather than oracle information.

Overall, this micro–meso–macro design provides complementary views: profitability and diversity of individuals, relational structures within populations, and systemic variance decomposition at the ecology level. Together they enable both fine-grained and aggregate analysis of evolving market games.

### G.6.4 ARTIFICIAL SHOCKS

We inject controlled exogenous shocks by simultaneously perturbing (i) the news stream processed by event-driven/LLM agents and (ii) the latent fundamental-value process observed by informational traders, while (iii) tightening microstructure conditions to mimic liquidity stress. Unless otherwise stated, shocks are scheduled during **12:00–13:30 (local time)** and we report both positive and negative cases.

**News channel.** Let $\lambda_{\text{base}}$ be the baseline Poisson rate for news arrivals and $z_t$ the scalar sentiment payload. During the shock window $[t_s, t_e]$ we amplify both *frequency* and *magnitude*:

$$\lambda_{\text{shock}} = \kappa\, \lambda_{\text{base}}, \qquad z_t \sim \mathcal{N}(\mu_s, \sigma_{\text{news}}^2), \ \ s \in \{+1, -1\},$$

where $s = +1$ denotes a positive shock ($\mu_+ > 0$) and $s = -1$ a negative shock ($\mu_- < 0$). All agents that subscribe to the news bus (event-driven, LLM) ingest these messages in real time; the higher-rate, large-magnitude signals create an immediate directional impulse. **Fundamental channel.** Informational traders observe an oracle-provided *fundamental anchor* $P_t^\star$, which we use to maintain a stable link between transaction prices and real-world valuation. Shocks are injected *on top of* this anchor rather than replacing it. Let $P_{\text{base}}^\star(t)$ denote the baseline fundamental curve (before shocks). At the news-shock start $t_s$ we overlay a deterministic bump aligned with the shock, so the effective anchor becomes

$$\tilde{P}_t^\star = P_{\text{base}}^\star(t)\left(1 + s\, A_f\, e^{-(t-t_s)/\tau_f}\, \mathbf{1}\{t \geq t_s\}\right),$$

where $s \in \{+1, -1\}$ indicates positive/negative shocks, $A_f$ controls impact size (*percentage* of the baseline anchor), and $\tau_f$ sets the decay horizon. Informational traders then act on mispricing relative to the *effective* anchor as in our implementation: with $\delta_t = (P_t - \tilde{P}_t^\star)/\tilde{P}_t^\star$, they buy when $\delta_t < -\theta$ and sell when $\delta_t > \theta$, thereby transmitting fundamental shocks to order flow while preserving the anchor linkage.:contentReference[oaicite:1]index=1

### G.6.5 DATA DETAILS

To ground our intraday simulations in realistic conditions, we construct a multi-source financial dataset that integrates heterogeneous information streams across assets, firms, and macroeconomic

environments. We deliberately cover two contrasting market regimes—the global *bear market* of January–October 2022 and the *bull market* of October 2024–July 2025—to ensure robustness across structurally distinct dynamics. All signals are aligned at sub-daily resolution, enabling high-frequency event-driven trading and ecological adaptation.

**Market events and news.** High-frequency event streams are obtained from **GDELT**, **Reuters Markets**, and the **Bloomberg Terminal**, each providing thousands of timestamped market-relevant headlines per trading day at minute-level granularity. These data form the primary source of exogenous shocks that drive intraday volatility.

**Macroeconomic and policy signals.** Macroeconomic indicators from **FRED** (e.g., GDP, CPI, unemployment, Treasury yields) are modeled as scheduled events and injected at their official release timestamps. **Federal Reserve policy documents** (FOMC minutes, statements, and press conferences) are treated as discrete intraday shocks aligned to release time, providing textual signals about monetary stance and forward guidance.

**Firm-level and analyst data.** Corporate earnings reports, financial statements, and **analyst forecasts** (EPS, target prices, sector outlooks) are included at their official disclosure times. Since these variables evolve more slowly, they are interpolated into daily anchors that inform fundamental-based agents.

**Integration and routing.** Each data type is dispatched to specialized analytical agents according to its temporal resolution and structural format: high-frequency event streams to intraday news agents, scheduled macro releases to shock-sensitive agents, and low-frequency fundamentals to balance-sheet interpreters. This routing ensures that heterogeneous signals are jointly exploited for both event-driven trading and long-horizon adaptation.

| Source | Content | Resolution |
|---|---|---|
| GDELT / Reuters / Bloomberg | Market news, corporate events | Minute-level intraday |
| FRED / Macro releases | GDP, CPI, unemployment, yields | Intraday at release |
| Firm reports / Analyst forecasts | Earnings, targets | Quarterly (daily anchors) |
| Fed / FOMC | Policy statements, press conf. | Intraday at release |

Table 11: Data sources and temporal resolution aligned to the trading clock for intraday simulation.

### G.7 MOTIVATION AND RATIONALE OF SIMULATION SCENARIOS

We provide here the motivations and empirical grounding of the two scenarios used in Section 3.

**Scenario 1: Artificial Shocks (Controlled Mechanism Validation).** This scenario is designed as a controlled stress test of the FinEvo evolutionary dynamics. By injecting exogenous positive and negative shocks into an otherwise clean synthetic market, we can isolate how disturbances propagate through the selection, innovation, and perturbation components of the evolutionary SDE. This controlled setup allows for transparent tracing of shock amplification, ecological transitions, and re-stabilization patterns, which would be difficult to interpret if tested only under noisy real-world conditions. Such controlled perturbation tests are standard in the study of heterogeneous-agent models and systemic risk.

**Scenario 2: Real-World News (Empirical Relevance and Robustness).** After validating the mechanisms in a controlled environment, we evaluate FinEvo under news-driven real-market conditions using high-frequency GDELT/Reuters releases. This tests the framework's ability to remain stable and informative when exposed to unstructured, economically meaningful signals that influence short-horizon returns, order flow, and trading behavior. This scenario also assesses whether heterogeneous agents—in particular LLM-based traders—can extract useful signals from real news and survive evolutionary selection.

**Bull vs. Bear Market Regimes.** Within the Real-World News scenario, we further embed two major macro environments: a bear phase (January–October 2022) and a bull phase (October 2024–July 2025). These distinct regimes allow us to examine how evolutionary equilibria change across volatility and sentiment conditions. As shown in Figure 6, the stable ecological configurations differ substantially across market regimes, illustrating that FinEvo does not collapse to a single fixed structure, but responds to macroeconomic context in a realistic manner.

**Summary.** Together, the controlled Artificial Shocks scenario and the empirically grounded Real-World News scenario form a coherent validation path: from mechanism-level stress testing to real-market robustness analysis. This design demonstrates that FinEvo is capable of capturing both theoretical evolutionary dynamics and realistic market adaptations, providing a comprehensive evaluation of the framework.

G.8    ECONOMIC MOTIVATION FOR NEWS-DRIVEN SYNTHETIC MARKETS

This section provides additional clarification on the economic foundations of our news-driven experiments and the role of LLM-based agents in FinEvo.

**No Structural Bias Toward LLM Dominance.** FinEvo does not assume, nor structurally enforce, the dominance of LLM-based agents, and their performance is not hard-coded through the sentiment–return mapping. As shown in Figure 6, LLM agents are not uniformly superior: in bullish regimes, the Informer agent outperforms FinCon and Trading, while in bearish regimes DQN achieves higher returns than Trading and CoT. This indicates that relative performance is determined by the evolving ecology and market conditions, rather than by any built-in preference for a particular model class.

**Economic Basis for the Sentiment–Return Mapping.** The mapping from news sentiment to value/return shocks is grounded in established results in market microstructure and behavioral finance, rather than chosen heuristically. In the spirit of Kyle's model of information-based trading (Kyle, 1985), information shocks affect order flow and prices through informed and liquidity traders. Behavioral models show that sentiment and investor belief distortions can drive short-horizon predictability and crash risk (e.g., Barberis et al., 1998). Empirically, major news announcements are known to increase volatility, jump intensity, and variance dynamics (Engle & Ng, 1993; Andersen et al., 2003). In FinEvo, news shocks enter by shifting agents' perceived value processes $V_k(t)$, which is consistent with these mechanisms: sentiment does not directly dictate prices, but perturbs perceived fundamentals and thereby indirectly affects order flow and returns.

**Why Use Synthetic Endogenous Price Dynamics.** We intentionally use synthetic, endogenously generated price series in the news experiments, rather than replaying a fixed historical tape. Our goal is to study closed-loop evolutionary feedback between heterogeneous strategies and the market, which real historical prices cannot capture because the true price path was generated under an unknown and unmodeled ecology. In FinEvo, a news shock acts as an exogenous information input, while the resulting price dynamics are fully endogenous, jointly determined by all interacting agents (LLM, RL, DL, and rule-based). This design enables us to analyze how different strategy populations respond to the same news process and how evolutionary selection reshapes the ecology over time.

G.9    PURPOSE OF THE SIMULATION SECTION

The goal of Section 3 is to demonstrate that FinEvo provides a general dynamical framework for studying market-level evolutionary phenomena induced by interacting heterogeneous agents.

The simulations serve three purposes: (1) validating that the proposed evolutionary SDE coupled with a continuous double auction generates endogenous ecological dynamics (selection, dominance cycles, phase transitions); (2) illustrating system-wide responses to perturbations such as sentiment shocks or liquidity disturbances; and (3) enabling controlled experiments for market design and policy analysis.

## H  SCIENTIFIC INVESTIGATION VS. TRADING-MODEL CONSTRUCTION

In this appendix, we clarify what we mean by "scientific investigation of markets" in the context of FinEvo. While trading-model construction is one possible use of simulation, FinEvo is designed to support a broader research agenda focused on the emergent and systemic properties of market ecosystems. These questions cannot be addressed by optimizing a single agent's PnL in a replayed historical environment.

**Systemic Risk and Stability.**  FinEvo enables the study of how market-wide stability changes when the strategy composition evolves. For example, the emergence of a new "species" (e.g., LLM-based agents) may alter ecological stability, volatility persistence, or crowding-induced transitions such as flash-crash–like dynamics. These phenomena arise from endogenous selection, innovation, and perturbation within the evolutionary SDE and cannot be replicated by training a single agent.

**Policy and Regulation Analysis.**  Regulators focus on the behavior of the overall market rather than on individual trading profits. FinEvo provides a controlled virtual environment to evaluate how policies—such as transaction taxes, liquidity constraints, or rule changes—affect innovation rates, species survival, and ecological diversity. This positions FinEvo as a *policy laboratory*.

**Market Design.**  FinEvo also enables the analysis of how auction formats, matching rules, or latency structures influence ecosystem-level coexistence among heterogeneous agents (rule-based, RL, LLM). These are structural questions about how the market functions as an adaptive game, rather than how to construct a profitable strategy.

**Summary.**  Thus, the primary research use of FinEvo is to study the adaptive, emergent, and systemic dynamics of market ecosystems—a long-standing objective in market-microstructure and financial-economics research.

## I  FUTURE WORK: ENHANCING EMPIRICAL REALISM

This appendix outlines several natural extensions to increase the empirical realism of FinEvo. Our current contribution focuses on establishing the evolutionary formalism—the FinEvo SDE and its selection–innovation–perturbation dynamics—together with a microstructure-aware CDA engine. The framework was deliberately designed to support the realism enhancements described below.

**L2-Level Microstructure Calibration.**  A key direction is incorporating empirical limit-order-book statistics (e.g., LOBSTER) to calibrate depth profiles, spread dynamics, order-arrival processes, and queueing effects. FinEvo's modular separation between the CDA mechanism and evolutionary dynamics makes this integration straightforward.

**Hybrid Replay + Endogenous Evolution.**  We plan to introduce a hybrid regime in which external liquidity is replayed from historical LOB data, while strategy populations evolve endogenously under the FinEvo SDE. This provides both empirical fidelity and evolutionary tractability.

**Richer Economic Feedback Mechanisms.**  Additional realism can be achieved by introducing inventory constraints, market-impact feedback, transaction taxes, throttling rules, and adaptive market-design modules. These components allow FinEvo to be used not only as an evolutionary model but also as a platform for evaluating policy interventions and systemic risk.

**High-Fidelity Interactive Environment for Agent Research.**  A more realistic market ecology benefits research on RL agents, LLM agents, and hybrid systems. An enhanced FinEvo environment provides non-stationary competitive pressure, realistic order-book interactions, and emergent collective behaviors that cannot be obtained from isolated backtests.

**Scientific Applications.**  Improved realism enables deeper investigation of systemic fragility, shock propagation, strategy concentration, and regulatory effects on long-term ecological equilibria. This aligns FinEvo with broader market-microstructure and financial-economics research agendas.

**Open-Ended Strategy Evolution.**  An additional extension is to allow the strategies themselves to evolve over time, rather than keeping the strategy set fixed.  This could be achieved through genetic algorithms, neuroevolution, or policy hybridization mechanisms that mutate or recombine existing strategies to create new ones. While such open-ended evolution substantially enriches the expressive power of the ecosystem, it also leads to a rapidly expanding strategy space that complicates theoretical analysis.  In this work, we focus on establishing the ecological game formalism and its population-level dynamics, but the FinEvo framework is fully compatible with incorporating strategy-level evolution in future research.

**Network-Based Propagation and Local Diffusion.**  While FinEvo models population-level diffusion through its innovation and perturbation terms—consistent with the mean-field formulation of evolutionary game theory )—an important future direction is to incorporate micro-level propagation mechanisms based on local interactions. Such dynamics include diffusion of strategies through adjacent nodes in an agent network, local imitation on a graph, and contagion driven by neighborhood structures. Introducing network-based propagation would allow FinEvo to capture spatially heterogeneous diffusion patterns, clustered adoption, and localized cascades, complementing the current mean-field evolutionary dynamics. The modular design of the FinEvo SDE makes it straightforward to extend the evolutionary mechanism with graph-based transition operators or network-weighted innovation kernels, enabling richer models of behavioral contagion and microstructural propagation in future work.

## J    LIMITATIONS

While FinEvo provides a unified ecological game formalism with endogenous market dynamics, several modeling assumptions introduce limitations that also serve as directions for future improvement.

**Scope of the Evolutionary Mechanism.**  The evolutionary dynamics are formulated at the population level following the mean-field tradition of evolutionary game theory.  This provides analytical clarity but abstracts away from network-based local propagation. As discussed in Appendix I, the framework can be naturally extended with graph-based transition operators to model diffusion through adjacent nodes.

**External Knowledge of LLM Agents.**  LLM-based agents may have encountered textual descriptions of certain historical events during pretraining.  Since FinEvo generates endogenous market dynamics and the LLMs do not access future prices or internal microstructure, this overlap is unlikely to materially distort the comparative ecological outcomes we report.

## K    USE OF LARGE LANGUAGE MODELS

Large language models (LLMs), such as ChatGPT, were employed solely for minor language polishing. All conceptual development, technical contributions, experiments, and analyses were conducted exclusively by the authors.

