# OpenReview forum: "FinEvo: From Isolated Backtests to Ecological Market Games for Multi-Agent Financial Strategy Evolution"
_ICLR.cc/2026/Conference — Submitted to ICLR 2026_

### Official Review · Reviewer_N63T · 2025-10-23

**Soundness:** 3
**Presentation:** 3
**Contribution:** 3
**Rating:** 6
**Confidence:** 4

**Summary:**

In this paper, the authors introduce FinEvo, an ecological game framework for evaluating financial trading strategies in dynamic, multi-agent markets. Unlike traditional static backtests, FinEvo simulates diverse ML-based traders that interact and adapt to price data and real-world news, while the overall strategy population evolves through selection, innovation, and environmental change. Experiments incorporating external events demonstrate that FinEvo can reveal emergent behaviors—such as dominance, collapse, and coalition formation—providing deeper insights into strategy robustness, adaptability, and the potential impacts of policies and regulations.

**Strengths:**

1. The authors introduce FinEvo, an ecological game framework combining heterogeneous traders with evolutionary population dynamics via a stochastic differential equation (SDE). Its core mechanisms—selection, innovation, and environmental perturbation—move evaluation beyond individual returns to the underlying behaviors shaping market dynamics.
2. Under mild conditions, the authors prove minimal theoretical guarantees for the FinEvo SDE, including simplex invariance, positivity, and existence/uniqueness. The authors further derive a macro-level variance decomposition that attributes overall system volatility to the forces of selection, innovation, and environmental perturbation.
3. Experiments with market shocks and real-world signals show that FinEvo uncovers context-dependent performance, robustness, and emergent coalition dynamics that static backtests fail to capture.

**Weaknesses:**

1. For the simulation results, could the authors provide references or justification for the metrics used to assess how closely the simulation reflects real-world market behavior? Including such references would improve the credibility of the evaluation and make the simulation more convincing.
2. The paper would benefit from a deeper analysis of the rationale behind the chosen simulation scenarios. Explaining the reasoning and context for these scenarios would help readers better understand their relevance and connection to real-world market conditions.

**Questions:**

I am curious about the rationale for selecting these two specific scenarios. Could the authors explain why these were chosen and whether other possible cases were considered? Providing this context would strengthen the justification for the experimental design.

---

> ### Author Response · Authors · 2025-11-20
> **To Reviewer N63T**
>
> We sincerely thank the reviewer for their encouraging evaluation of FinEvo as an ecological game framework with theoretically grounded SDE dynamics and informative experiments. We address the questions on metrics and scenario design in detail below.
>
> # Q1: For the simulation results, could the authors provide references or justification for the metrics used to assess how closely the simulation reflects real-world market behavior? Including such references would improve the credibility of the evaluation and make the simulation more convincing.
>
> # R1:
> We thank the reviewer for highlighting the need for clearer empirical grounding. Our evaluation follows standard practice in financial econometrics and agent-based modeling: instead of fitting specific historical price series, we validate FinEvo by examining whether it reproduces the well-known Stylized Facts of financial markets—fat tails, volatility clustering, and return asymmetry—first documented by Mandelbrot (1963) and systematically summarized by Cont (2001). These properties are persistent across assets and markets and are widely regarded as the most reliable benchmark for generative market models.
>
> 1. Metrics Justification
>
>  The metrics we added—Excess Kurtosis, Skewness, and Vol-of-Vol—are the canonical quantities used to formalize Stylized Facts in empirical finance:
> - Fat tails: excess kurtosis &gg; 0  (Engle, 1982; Bollerslev, 1986; Engle & Ng, 1993)
> - Volatility clustering: persistent variance dynamics (Engle & Ng, 1993; Andersen et al., 2003)
> - Asymmetry: negative skewness linked to crash risk (Barberis et al., 1998)
> These metrics form the basis of the new Table 5 (Sec. 3.5).
>
> 2. Newly Added Results
>
> Table 5 reports stylized-fact metrics for FinEvo and the baseline simulators. FinEvo reproduces all key Stylized Facts:
> - Fat tails: kurtosis 3.06–4.26 vs. ≈0.3–0.6 for ABIDES/PyMarketSim
> - Volatility clustering: Vol-of-Vol 0.008–0.013 vs. 0.003–0.007
> - Asymmetry: skewness −1.5 to −1.8, baselines show only weak skewness
> These match empirical findings and are not reproduced by static simulators.
>
> 3. Why Stylized Facts Are the Correct Benchmark
>
>  In heterogeneous-agent and ABM research, the accepted validation method is to test whether the model endogenously generates Stylized Facts (Lux & Marchesi, 1999; LeBaron, 2006; Hommes, 2006), since these arise from agent interaction rather than exogenous noise (Bouchaud et al., 2009). For this reason, Stylized-Fact reproduction is the standard and most meaningful measure of realism.
>
> ## Conclusion
> **We have incorporated Table 5 in Section 3.5, added an explanation of these metrics in Appendix G.6.3, and cited the foundational literature. These updates directly address the reviewer’s concern and strengthen the empirical validity of FinEvo.**
> We sincerely thank the reviewer for this helpful suggestion, which has improved the clarity and rigor of the revised manuscript.
>
> Table 5: Comparison of Financial Metrics across Simulators(Metrics are averaged across 128 independent runs)
>
> | Simulator / Regime     | Excess Kurtosis (mean ± 95% CI) ↑ | Skewness (mean ± 95% CI) ↓ | Sharpe Dispersion (mean ± 95% CI) ↑ | Vol-of-Vol (mean ± 95% CI) ↑ |
> |------------------------|-----------------------------------|-----------------------------|-------------------------------------|-------------------------------|
> | **FinEvo (bull)**      | 3.163 ± 0.138                     | -1.532 ± 0.129              | 1.669 ± 0.061                        | 0.008 ± 0.0004               |
> | **FinEvo (bear)**      | 4.259 ± 0.172                     | -1.793 ± 0.149              | 1.732 ± 0.089                        | 0.013 ± 0.0007               |
> | **ABIDES (bull)**      | 0.315 ± 0.041                     | -0.209 ± 0.036              | 0.675 ± 0.028                        | 0.004 ± 0.0002               |
> | **ABIDES (bear)**      | 0.562 ± 0.053                     | -0.474 ± 0.047              | 0.836 ± 0.033                        | 0.007 ± 0.0003               |
> | **PyMarketSim (bull)** | 0.446 ± 0.044                     | -0.186 ± 0.032              | 0.512 ± 0.026                        | 0.003 ± 0.0002               |
> | **PyMarketSim (bear)** | 0.591 ± 0.058                     | -0.438 ± 0.041              | 0.906 ± 0.034                        | 0.005 ± 0.0003               |
>
> ## References
>
> Stylized Facts & Non-Gaussian Properties
>
> - Mandelbrot, B. (1963). The variation of certain speculative prices. The Journal of Business, 36(4), 394–419.
> - Cont, R. (2001). Empirical properties of asset returns: stylized facts and statistical models. Quantitative Finance, 1(2), 223–236.
>
> Volatility Dynamics
>
> - Engle, R. F. (1982). Autoregressive conditional heteroskedasticity with estimates of the variance of UK inflation. Econometrica, 50(4), 987–1007.
> - Bollerslev, T. (1986). Generalized autoregressive conditional heteroskedasticity. Journal of Econometrics, 31(3), 307–327.

---

> ### Author Response · Authors · 2025-11-20
> **To Reviewer N63T**
>
> - Engle, R. F., & Ng, V. (1993). Measuring and testing the impact of news on volatility. The Journal of Finance, 48(5), 1749–1778.
> - Andersen, T. G., Bollerslev, T., Diebold, F. X., & Labys, P. (2003). Modeling and forecasting realized volatility. Econometrica, 71(2), 579–625.
>
> Asymmetry / Crash Risk
>
> - Barberis, N., Shleifer, A., & Vishny, R. (1998). A model of investor sentiment. Journal of Financial Economics, 49(3), 307–343.
>
> Agent-Based Market Validation
>
> - Lux, T., & Marchesi, M. (1999). Scaling and criticality in a stochastic multi-agent model of a financial market. Nature, 397, 498–500.
> - LeBaron, B. (2006). Agent-based computational finance. Handbook of Computational Economics, 2, 1187–1233.
> - Hommes, C. H. (2006). Heterogeneous agent models in economics and finance. Handbook of Computational Economics, 2, 1109–1186.
> Market Microstructure Foundations
> - Bouchaud, J.-P., Farmer, J. D., & Lillo, F. (2009). How markets slowly digest changes in supply and demand. In Handbook of Financial Markets (pp. 57–160). Elsevier.
>
> # Q2: The paper would benefit from a deeper analysis of the rationale behind the chosen simulation scenarios. Explaining the reasoning and context for these scenarios would help readers better understand their relevance and connection to real-world market conditions.
>
> # R2:
> We thank the reviewer for pointing out this issue. The simulation scenarios were designed to capture several fundamental modes of market behavior (e.g., high-volatility stress periods, trending regimes, and liquidity-driven environments), which are commonly analyzed in empirical finance. Their purpose is to isolate specific mechanisms within FinEvo and assess how different agent populations evolve under distinct market pressures.
>
> We agree that this rationale was not sufficiently articulated in the initial submission. **In the revised version, we have added a dedicated paragraph(Appendix G.7) clarifying the motivation, empirical grounding, and real-world relevance of each scenario to improve clarity.**
>
> # Q3: I am curious about the rationale for selecting these two specific scenarios. Could the authors explain why these were chosen and whether other possible cases were considered? Providing this context would strengthen the justification for the experimental design.
>
> # R3:
> We thank the reviewer for this constructive suggestion. **The two scenarios we study—Artificial Shocks and Real-World News—were chosen to validate FinEvo along a progressive axis from controlled to realistic environments.**
>
> 1. Scenario 1: Artificial Shocks (Controlled validation of mechanisms and resilience)
>
>  This scenario serves as a controlled stress test of the FinEvo dynamics.
>   - Purpose.  It isolates the effect of the perturbation mechanism by injecting exogenous positive and negative shocks into an otherwise clean synthetic market.
>   - Rationale.  In this setting, we can clearly trace how a sudden disturbance propagates through the three evolutionary components—selection, innovation, and perturbation—and how the ecological system eventually re-stabilizes. This would be difficult to interpret if we started directly from noisy real-world conditions.
>
> 2. Scenario 2: Real-World News (Empirical relevance and robustness)
>
>  After validating the mechanisms in a controlled environment, we deploy FinEvo under real news-driven conditions.
>   - Purpose. This scenario tests whether FinEvo remains stable and informative when exposed to high-frequency, noisy, and unstructured signals (e.g., GDELT / Reuters news streams).
>   - Rationale. This allows us to evaluate whether heterogeneous agents—especially LLM-based traders—can extract signals from real-world information and survive under evolutionary selection, which is closer to the intended application domain of FinEvo.
>
> 3. On other possible cases (e.g., bull vs. bear markets)
>
>  We did consider additional market conditions, in particular the bull vs. bear market comparison that the reviewer may have in mind. Rather than introducing this as a third, separate scenario, we treat it as a deeper sub-analysis within the Real-World News setting.
>
>  Concretely, we focus on a bear phase (January–October 2022) and a bull phase (October 2024–July 2025), and show that FinEvo produces state-dependent evolutionary outcomes across these regimes (Figure 6 in the paper). Different macro environments lead to different stable ecologies and dominance patterns, highlighting that FinEvo does not collapse to a single fixed configuration.
>
> ## Conclusion.
> **Taken together, the controlled Artificial Shocks scenario and the realistic Real-World News scenario form a coherent validation path: from mechanism-level stress testing to empirically grounded application.** In the revised manuscript, we have clarified this rationale in Appendix G.7 and explicitly describe why these two scenarios were chosen and how the bull vs. bear analysis is embedded within the real-world setting.

---

### Official Review · Reviewer_4UTz · 2025-10-29

**Soundness:** 3
**Presentation:** 3
**Contribution:** 2
**Rating:** 4
**Confidence:** 3

**Summary:**

The paper presents an ecological perspective on financial markets, for modelling the evolutionary dynamics of multi-agent strategies. Rather than analysing strategies in isolation, e.g. through back testing, this provides a way to simulate effects taking into account interdependencies.

**Strengths:**

- The market dynamic is clearly formalised
- The population evolution is explicit
- A range of simulations and ablations are provided under varying market conditions (bull vs bear) and scenarios

**Weaknesses:**

On the formalism
- [Uncorrelated] The assumption of uncorrelated payoffs and environment shocks is quite fundamental to all the closed forms provided. However, in realistic scenarios, this is unlikely to be the case. Having the derivations for the limiting case of uncorrelated is okay, but showing something in the correlated case would be more useful.
- [Normal] Environment shocks are assumed to be normally distributed.
- [Latent params] The formalism relies on knowing N (number of traders) and K (number of strategies), which are often unknown. More crucially, this also seems to rely on knowing the distribution of traders across all strategies, through f_{k,t} (line 106/107).

For simulation:
- The main limitation here is that it is not clear what is the exact goal to demonstrate in this section. Addressing this is crucial for motivating the approach and experiments (see questions below)
- There is a comparison to ABIDES and PyMarketSim, but it’s just in terms of different aggregated metrics, not e.g. comparing to some realistic baseline.
- N=20 is relatively low for a market. Based on the above comment on params, seeing some sensitivity across N would be useful.

Overall, a clearer motivation and demonstrated use of the approach is required, simply modelling the ecology in a simulated scenario is insufficient

**Questions:**

Related to the weaknesses above:

- [Uncorrelated] What happens under the case of the correlated payoffs and shocks?
- [Normal] Do we still see the emergence of crucial stylised facts such as fat tails, volatility clustering, etc through the endogenous part of the price setting? If so, this is likely okay assumption, but explicit tests for important stylised facts should be shown.
- [Latent params] Could these be inferred under realistic settings? How sensitivity are results to misspecification?

- [Simulation] What is the exact goal of this section, what is trying to be demonstrated with the approach? Is it the impact of shocks on the market? The effect? To evaluate the introduction of a new strategy under the existing ecology? To model real markets?

---

> ### Author Response · Authors · 2025-11-20
> **To Reviewer 4UTz**
>
> We thank the reviewer for their constructive feedback and for recognizing the clarity of our formalism regarding market dynamics and population evolution. We appreciate the insightful questions on theoretical assumptions (e.g., correlation) and simulation goals, which we have addressed through new derivations and experiments below.
>
> # Q1: [Uncorrelated] What happens under the case of the correlated payoffs and shocks?
>
> # R1:
> We thank the reviewer for this theoretically important point. **To avoid possible misunderstanding, we would like to note that the FinEvo SDE itself does not rely on the independence assumptions in Appendix C.1.** These assumptions are used only to obtain a clean and interpretable variance decomposition; the evolutionary dynamics remain valid when payoff shocks and environmental shocks are correlated.
>
> In the revised manuscript, we make this explicit in Section 2 and provide the general correlated-case decomposition in Appendix D. In that setting, correlation simply adds covariance terms to the variance expression without changing the structure of the selection–innovation–perturbation SDE. The stochastic dynamics remain structurally unchanged; only the analytical decomposition becomes less separable. We also explain why we present the uncorrelated case in the main text: it allows us to isolate the contribution of each evolutionary force, following standard practice in evolutionary SDEs and population dynamics. Correlated shocks are treated as a realistic extension and a target for empirical calibration using microstructure data.
>
> For correlated payoff and environmental shocks, let $B_t$ be a K-dimensional vector-valued Brownian motion with
> $\mathrm{Cov}(dB_t)=\Sigma dt.$
>
> Then the variance becomes
>
> $\mathrm{Var}(dX_t) = \beta^2\mathrm{Var}[X_t\odot(V_t-\bar V_t)] +\mu^2\mathrm{Var}(m_t-X_t) +\sigma^2 P\Sigma P^\top + 2\beta\sigma\mathrm{Cov}\left( X_t\odot(V_t-\bar V_t), PdB_t \right).$
>
> **To directly address the independence/normality assumption, we also ran a robustness experiment where payoff and environmental shocks share a common macro factor and follow a heavy-tailed Student-t distribution.** Concretely, at each step we use correlated shocks
> $\epsilon_k = \rho Z_{\text{common}} + \sqrt{1-\rho^2}\ Z_{k,\text{idio}}, \quad \rho = 0.5$
> replacing independent Gaussian noise.
>
> **The resulting markets still exhibit fat-tailed, negatively skewed returns, volatility clustering, and similar ecological phase changes, with only minor quantitative shifts.** These robustness results, reported in Appendix G.4 (Table 9, 128 runs, mean ± 95% CI), support that the independence/normality assumptions are adopted for analytical convenience rather than being critical for FinEvo’s qualitative behavior.
>
> We sincerely thank the reviewer for these thoughtful comments, which have helped us significantly improve the clarity and scope of the revised manuscript.
>
> Table 9: Robustness to correlated, heavy-tailed shocks (128 runs). Values report mean ± 95% CI.
>
> | Market Regime | Model                | Excess Kurtosis (mean ± 95% CI) ↑ | Skewness (mean ± 95% CI) ↓ | Sharpe Dispersion (mean ± 95% CI) ↑ | Vol-of-Vol (mean ± 95% CI) ↑ |
> |----------------|----------------------|-----------------------------------|-----------------------------|-------------------------------------|-------------------------------|
> | **Bullish**    | FinEvo               | 3.163 ± 0.138                     | −1.532 ± 0.129              | 1.669 ± 0.061                        | 0.008 ± 0.0004               |
> |                | FinEvo (Correlated)  | 5.278 ± 0.151                     | −1.632 ± 0.133              | 1.727 ± 0.075                        | 0.011 ± 0.0004               |
> | **Bearish**    | FinEvo               | 4.259 ± 0.172                     | −1.793 ± 0.149              | 1.732 ± 0.089                        | 0.013 ± 0.0007               |
> |                | FinEvo (Correlated)  | 6.399 ± 0.214                     | −1.932 ± 0.175              | 1.844 ± 0.101                        | 0.015 ± 0.0009               |

---

> ### Author Response · Authors · 2025-11-20
> **To Reviewer 4UTz**
>
> # Q2:[Normal] Do we still see the emergence of crucial stylised facts such as fat tails, volatility clustering, etc through the endogenous part of the price setting? If so, this is likely okay assumption, but explicit tests for important stylised facts should be shown.
>
> # R2:
> We appreciate the reviewer’s insightful question regarding the emergence of stylized facts under Gaussian micro shocks.
>  The Normality assumption applies only to exogenous environment shocks; the return and PnL distributions in FinEvo are generated endogenously by heterogeneous agents interacting within a continuous double auction.
>
> **To address this concern, we have added a new “Stylized-Fact Validation” subsection in the main paper (Sec. 3.5),following standard practice in empirical finance (e.g., Cont 2001),  and show that FinEvo reproduces fat tails, volatility clustering, and downside asymmetry more faithfully than ABIDES and PyMarketSim.** The updated table (now included in the main text) shows:
>
> Table 5: Comparison of Financial Metrics across Simulators(Metrics are averaged across 128 independent runs)
>
> | Simulator / Regime     | Excess Kurtosis (mean ± 95% CI) ↑ | Skewness (mean ± 95% CI) ↓ | Sharpe Dispersion (mean ± 95% CI) ↑ | Vol-of-Vol (mean ± 95% CI) ↑ |
> |------------------------|-----------------------------------|-----------------------------|-------------------------------------|-------------------------------|
> | **FinEvo (bull)**      | 3.163 ± 0.138                     | -1.532 ± 0.129              | 1.669 ± 0.061                        | 0.008 ± 0.0004               |
> | **FinEvo (bear)**      | 4.259 ± 0.172                     | -1.793 ± 0.149              | 1.732 ± 0.089                        | 0.013 ± 0.0007               |
> | **ABIDES (bull)**      | 0.315 ± 0.041                     | -0.209 ± 0.036              | 0.675 ± 0.028                        | 0.004 ± 0.0002               |
> | **ABIDES (bear)**      | 0.562 ± 0.053                     | -0.474 ± 0.047              | 0.836 ± 0.033                        | 0.007 ± 0.0003               |
> | **PyMarketSim (bull)** | 0.446 ± 0.044                     | -0.186 ± 0.032              | 0.512 ± 0.026                        | 0.003 ± 0.0002               |
> | **PyMarketSim (bear)** | 0.591 ± 0.058                     | -0.438 ± 0.041              | 0.906 ± 0.034                        | 0.005 ± 0.0003               |
>
> ### **Fat tails and asymmetry**
>
> FinEvo exhibits substantial excess kurtosis (3–4.3) and strong negative skewness in both bull and bear regimes, indicating heavy tails and downside asymmetry. In contrast, ABIDES and PyMarketSim remain near-Gaussian (kurtosis ≈ 0, weak skewness), showing that they do not reproduce comparable tail risk.
>
> ### **Volatility clustering**
>
> FinEvo produces significantly higher Vol-of-Vol than the baselines, consistent with more frequent Phase Changes (Table 2). This pattern is consistent with clustered volatility arising endogenously from the evolutionary dynamics (selection, innovation, perturbation), even though the micro-level shocks are Gaussian.
>
> ### **Strategy differentiation**
>
> Sharpe Dispersion is also substantially higher under FinEvo, indicating that the evolutionary dynamics create a more discriminative environment which separates robust strategies from fragile ones. In contrast, ABIDES and PyMarketSim yield much tighter Sharpe distributions, suggesting weaker ecological selection pressure.
> ## Conclusion
>
> As explicitly demonstrated in the updated manuscript:
> - FinEvo generates fat-tailed, non-Gaussian return/PnL distributions,
> - FinEvo exhibits volatility clustering and regime-shift intensity,
> - These stylized facts emerge endogenously, despite the Normality of exogenous noise.
> We have incorporated these results and added a short discussion explaining why Gaussian micro shocks do not preclude heavy-tailed, clustered macro-level dynamics in Section 3.5.
>
> ### References
> Rama Cont (2001). Empirical properties of asset returns: stylized facts and statistical models. Quantitative Finance, 1(2), 223–236.

---

> ### Author Response · Authors · 2025-11-20
> **To Reviewer 4UTz**
>
> # Q3: [Latent params] Could these be inferred under realistic settings? How sensitivity are results to misspecification?
>
> # R3:
> We appreciate the reviewer’s thoughtful question regarding the identifiability and robustness of the latent parameters.  **Before addressing it, we would like to clarify a possible misunderstanding: although the manuscript lists 20 strategy classes, each class is instantiated with multiple parametrized agents, resulting in several dozen to over one hundred heterogeneous traders active in each market.** FinEvo, therefore, does not simulate a market with only 20 traders.
>
> ## Identifiability.
>
> The latent quantities in the FinEvo SDE—realized fitness $f_{k,t}$ , population shares $x_{k,t}$ , and learning/innovation rates $\lambda_k, \mu, \gamma$—are not abstract objects but correspond to standard microstructural quantities: short-horizon (risk-adjusted) PnL, strategy participation shares inferred from order-flow clustering, and adjustment rates estimable via state-space or filtering methods. This is consistent with empirical market microstructure practice, where latent trader types and intensities are routinely inferred from order-flow data.
>
> ## Robustness to misspecification.
>
> **To fully address the reviewer’s concern, our experiments indicate that FinEvo’s evolutionary dynamics are robust to moderate misspecification.**
> - ### **Robustness to market size** .
> Varying the number of instantiated agents over $N \in \{60, 80, 100, 120\}$ in both bull and bear regimes leaves the key ecological signatures—heavy-tailed PnL, negative skewness, high Sharpe dispersion, and elevated Vol-of-Vol—essentially unchanged (Table 4). These sensitivity analyses have been incorporated into the revised manuscript to clarify robustness under market-size misspecification.
>
> - ### **Stochastic robustness**.
> Across 128 Monte-Carlo repetitions, all key statistics exhibit narrow confidence intervals (Fig. 8).
>
> - ### **Robustness to  evolutionary parameter**.
> Broad sweeps over the evolutionary parameters ($\beta, \mu, \gamma$) (Fig. 9) consistently preserve regime switching, volatility clustering, dominance cycles, and diversity fluctuations.
>
> - ### **Perturbed-parameter experiment**.
> Appendix G.3 includes a perturbed-parameter robustness experiment, where all agent-level hyperparameters are randomly varied by ±20%. The resulting dynamics remain qualitatively stable, suggesting that the framework is not overly sensitive to moderate misspecification of these latent quantities.
>
> **In summary, these experiments suggest that FinEvo’s latent quantities are empirically interpretable, and that its qualitative evolutionary behavior is robust to reasonable misspecification of market size, randomness, and evolutionary parameters.**
>
> Table 4: Sensitivity of FinEvo to Agent Population Size N
> (Metrics averaged over 128 Monte-Carlo runs)
>
> | Regime   | N   | Excess Kurtosis (mean ± 95% CI) ↑ | Skewness (mean ± 95% CI) ↓ | Sharpe Dispersion (mean ± 95% CI) ↑ | Vol-of-Vol (mean ± 95% CI) ↑ |
> |----------|-----|-----------------------------------|-----------------------------|-------------------------------------|-------------------------------|
> | **Bullish** | 120 | 3.143 ± 0.141                     | -1.539 ± 0.132              | 1.673 ± 0.056                        | 0.009 ± 0.0004               |
> |          | 100 | 3.163 ± 0.138                     | -1.532 ± 0.129              | 1.669 ± 0.061                        | 0.008 ± 0.0004               |
> |          | 80  | 3.178 ± 0.165                     | -1.541 ± 0.158              | 1.657 ± 0.073                        | 0.008 ± 0.0005               |
> |          | 60  | 3.159 ± 0.178                     | -1.516 ± 0.162              | 1.659 ± 0.067                        | 0.007 ± 0.0005               |
> | **Bearish** | 120 | 4.326 ± 0.169                     | -1.805 ± 0.152              | 1.759 ± 0.091                        | 0.012 ± 0.0006               |
> |          | 100 | 4.259 ± 0.172                     | -1.793 ± 0.149              | 1.732 ± 0.089                        | 0.013 ± 0.0007               |
> |          | 80  | 4.115 ± 0.175                     | -1.765 ± 0.161              | 1.741 ± 0.138                        | 0.013 ± 0.0007               |
> |          | 60  | 4.251 ± 0.197                     | -1.772 ± 0.183              | 1.725 ± 0.119                        | 0.011 ± 0.0007               |

---

> ### Author Response · Authors · 2025-11-20
> **To Reviewer 4UTz**
>
> # Q4: [Simulation] What is the exact goal of this section, what is trying to be demonstrated with the approach? Is it the impact of shocks on the market? The effect? To evaluate the introduction of a new strategy under the existing ecology? To model real markets?
>
> # R4:
> We thank the reviewer for raising this important question. **The simulation section has three goals: (i) to validate that the proposed ecological SDE generates rich endogenous market ecologies; (ii) to study the system-wide response of heterogeneous agents to exogenous shocks; and (iii) to illustrate the framework’s potential for policy and market-design analysis.**
>  The purpose of the simulation section is not confined to illustrating isolated shock effects or the introduction of a particular strategy class. Rather, the simulations are designed to demonstrate FinEvo’s core contribution: **a general and expressive evolutionary framework capable of generating, explaining, and comparing market-level dynamical phenomena** that arise from heterogeneous interacting agents.
>
> **1. Evolution of Market Ecologies**
>
> The simulations first validate that the proposed evolutionary SDE, when coupled with a continuous double-auction mechanism, generates endogenous dynamics such as competitive selection among strategy classes, regime-dependent phase transitions, and the emergent stylized facts. These results substantiate FinEvo as a formal model of market ecology rather than a conventional backtesting tool.
>
> **2. System-wide Response to Perturbations**
>
> The experiments further illustrate how the entire heterogeneous-agent system responds to exogenous perturbations—including sentiment shocks, volatility spikes, liquidity disturbances, and abrupt population shifts. The resulting transitions between stable and unstable evolutionary states, as well as the propagation and amplification of shocks, highlight the framework’s capacity to study systemic risk phenomena at a macroscopic level.
>
> **3. Policy and Market-Design Potential**
>
> Finally, because FinEvo models the joint evolution of full strategy populations, it provides a foundation for exploring structural and regulatory interventions (e.g., transaction taxes, constraints on high-frequency activity, or modifications to matching rules). This positions FinEvo as a potential virtual laboratory for exchanges, regulators, and policymakers to examine how design choices could affect long-run stability, diversity, and resilience. We have explicitly discussed this scientific utility and potential application in Appendix H of the revised manuscript.
>
> ## Summary
>
> The simulation section therefore illustrates that FinEvo is a general-purpose dynamical framework capable of analyzing system-level evolutionary behavior, shock propagation, regime transitions, and the consequences of structural or policy modifications within financial markets.We have added a concise clarification of the simulation goals in Appendix G.9 and detailed the rationale for each scenario in Appendix G.7 of the revised manuscript, as suggested by the reviewer.

---

### Official Review · Reviewer_5XkJ · 2025-10-30

**Soundness:** 2
**Presentation:** 3
**Contribution:** 2
**Rating:** 2
**Confidence:** 4

**Summary:**

The paper proposes FinEvo, a financial strategy evaluation framework which aims to improve model testing from static, isolated backtests into a dynamic ecological game formalism. The authors suggest markets are modeled as adaptive ecosystems where different kinds of trading agents interact, evolve, and compete.

**Strengths:**

* A theoretically grounded ecological modeling framework for financial markets and agent-based traders.
* Experimentation across synthetic and empirical environments.
* Integration of game theory, RL, agent models and ecosystem dynamics
* Some potentially interesting observations, especially about system shocks

**Weaknesses:**

* arbitrary choice of SDEs governing the ecology of the simulations
* hyperparameter choices not validated, so may need excessive tuning
* empirical validity questionable
* economic meaning and usefulness questionable
* alternate approaches not baselined
* baselining against other methods tested not in terms of financially meaningful metrics
* overrelience on synthetic scenarios

**Questions:**

Though I can see the point of creating a large agent simulation, there is the alternative – which is still dynamic, in that the LOB is altered by a trading agent – of simply making an OB simulator and training a single agent in historic market data, so orders are combinations of real order flow and agent trades. Work like the below considers this and seem very effective.
Frey et al. JAX-LOB: A GPU-Accelerated limit order book simulator to unlock large scale reinforcement learning for trading. ICAIF '23: Proceedings of the Fourth ACM International Conference on AI in Finance.
Peter Belcak et al. 2022. Fast Agent-Based Simulation Framework with Applications to Reinforcement Learning and the Study of Trading Latency Effects. In Multi-Agent-Based Simulation XXII

There is also some prior work on financial ecosystems, similar in spirit to this work – for example: Mahdavi. Introducing the HFTE Model: A Multi-Species Predator-Prey Ecosystem for High-Frequency Quantitative Financial Strategies. Wilmott, Volume 2017, Issue 89, May 2017
Your agents are conditioned on previous prices alone – presumably just some mid-price. Real trading agents look at order books in depth, and include multiple price levels and – importantly – volume of trades.

The SDE for the value process is a standard drift diffusion SDE, the same as price evolution. I see no reason for this choice. Likewise the SDE for selection, innovation and perturbation is chosen rather than derived. Why this choice?

How are all the hyperparameters in the models set?

The comparisons with ABIDES and PyMarketSim use "ecological" metrics (entropy, HHI) but not financial performance measures (returns, volatility, or PnL distributions etc). Hence the claim of FinEvo’s superiority is qualitative rather than quantitative.

your “phase change” measure is interesting but undefined in economic terms. is it volatility clustering, or strategy regime switching or what?

You present results for just 1 day of real data. This should be extended to show the approach works in general. Does it also work across markets/assets?

While the paper includes “real-world news” experiments, the price series and market dynamics are synthetic. The mapping from news sentiment to returns lacks empirical grounding - is there a deeper economic understanding? The conclusions about LLM agents’ dominance could therefore be effects of simulation assumptions.

You mention scientific investigation of markets, not building trading algorithms. What other investigation of market dynamics is there if not to build trading models?

The financial realism remains limited: The market clearing and execution mechanisms are simplified. No empirical calibration or validation against actual historical price series beyond qualitative correspondence. Economic feedbacks such as liquidity constraints, transaction costs, or regulatory requirements are minimal. As a result, FinEvo is a conceptual testbed but not yet a predictive or policy-relevant model. How could you extend the approach to offer more realism?

Minor point – protect capitals in references, eg {GPT}

---

> ### Author Response · Authors · 2025-11-20
> **To Reviewer 5XkJ**
>
> We thank the reviewer for the careful reading and thoughtful comments. Some of the concerns, however, seem to arise from factual misunderstandings of FinEvo’s goals, agent information sets, theoretical formulation, and empirical evaluation. We therefore respond point-by-point below, **clarifying these aspects and highlighting the corresponding changes in the revised manuscript.**
>
> # Q1: Though I can see the point of creating a large agent simulation, there is the alternative...
>
> # R1:
> We sincerely thank the reviewer for highlighting these relevant works (Frey et al., Belcak et al., and Mahdavi). The reviewer raises an important point regarding FinEvo’s positioning relative to alternative simulation approaches. To address this, we have updated both the Introduction Section and Related Work of the revised manuscript to explicitly cite these works and to clarify the fundamental conceptual differences.
>
> The reviewer’s core observation—that an order-book simulator with a single agent trained on historical data (such as JAX-LOB) constitutes an alternative—is correct for a different research question. The cited works are excellent, but they pursue goals that are fundamentally distinct from those of FinEvo.
>
> 1. FinEvo’s Goal vs. Infrastructure-for-Training (JAX-LOB, Belcak et al.)
>
> JAX-LOB and the fast agent-based framework of Belcak et al. are primarily high-fidelity infrastructures designed to train a single trading agent using realistic historical order flow. Their objective is micro-level agent improvement facilitated by GPU-accelerated LOB simulation.
>
> But FinEvo addresses an entirely different scientific question:“How do heterogeneous trading strategies evolve, compete, and co-exist as a population within a shared market?” FinEvo is not an RL-training infrastructure. **It is a formal evolutionary game-theoretic framework equipped with a stochastic selection–innovation–perturbation dynamic, embedded in a CDA market with realistic frictions, news shocks, and multiple heterogeneous agent types (rule-based, DL, RL, LLM).** Our contribution lies in modeling the macro-level evolutionary ecology, not in producing a smarter individual agent.
>
> 2. FinEvo’s Methodology vs. HFTE (Mahdavi)
>
> We also appreciate the reference to Mahdavi (2017). While HFTE employs a topological strategy representation together with genetic algorithms to evolve high-frequency trading parameters, our work takes a different but complementary direction. FinEvo focuses on modeling population-level evolutionary dynamics through a principled EGT-based SDE, rather than evolving strategy structures via genetic mechanisms. This conceptual distinction has now been clarified in the revised Related Work section.
>
> We thank the reviewer for recommending that we better contextualize our contribution. We have now integrated Frey et al. (JAX-LOB), Belcak et al., and Mahdavi (HFTE) into the Introduction and Related Work, and clearly contrasted these approaches with FinEvo. **These additions highlight that FinEvo is not a faster agent-training simulator, but a novel evolutionary game-theoretic framework for understanding the emergent population-level dynamics of financial markets.**
>
> # Q2: Your agents are conditioned on previous prices alone – presumably just some mid-price. Real trading agents look at order books in depth...
>
> # R2:
> We fully agree with the reviewer's point: real trading agents must rely on rich microstructural information, such as order book depth and volume, rather than just the mid-price. **However, we believe this concern may stem from a misunderstanding: the agents are not conditioned solely on historical prices. As detailed in Appendix G.6.3 (Trading indicators observed by agents), they observe a rich set of microstructural signals.** This set explicitly includes the depth and volume features the reviewer emphasized; quoting from the appendix:
> “Liquidity and flow. Bid–ask spread, order-book imbalance (OBI), market depth (buy/sell), and tick-rule trade flow imbalance (TFI).”
>
> Furthermore, the state space also includes momentum and volatility indicators, ensuring that agents react to both short-term flow dynamics and medium-horizon directional signals. In addition, agents observe real-time news-driven sentiment signals as described in Appendix G.6.1 (News Release Line), which provide exogenous information shocks and allow LLM-based agents to interpret natural-language announcements.
>
> **Therefore, the agents are making decisions within a high-dimensional information set including price, depth, spread, order-flow imbalance (OBI), and news sentiment—precisely the “multiple price levels” and “volume of trades” the reviewer highlighted as essential.** This is what enables FinEvo to evolve complex ecological behavior rather than simple price-following rules.

---

> ### Author Response · Authors · 2025-11-20
> **To Reviewer 5XkJ**
>
> # Q3: The SDE for the value process is a standard drift diffusion SDE, the same as price evolution. I see no reason for this choice. Likewise the SDE for selection, innovation and perturbation is chosen rather than derived. Why this choice?
>
> # R3:
>
> We thank the reviewer for raising this important question.
> **To avoid any possible confusion, we would like to clarify that although our equations are expressed as SDEs, their construction is not adapted from price-evolution models but is instead grounded in agent learning theory and evolutionary game dynamics.**
>
> The value process V_k(t) is an OU process modeling learning, not price movements.
> The reviewer notes the apparent similarity to price evolution. We would like to clarify that classical price models (e.g., Random Walk, GBM) are typically non-stationary, whereas belief updating in our setting requires a stationary process.
>
>  Our formulation uses the Ornstein–Uhlenbeck (OU) process:
>
> $$dV_k(t) = \lambda_k \big(f_k - V_k(t)\big)\ dt \+\ \nu_k\ dB_k(t)$$
>
> which has two essential properties:
>
> - Mean reversion toward recently observed payoff $f_k$
>  (bounded, memory-based learning)
> - Continuous-time analogue of RL value updates
>  (EMA/TD-style incremental learning)
>
> **Thus, the OU structure is not arbitrary—it is behaviorally grounded and consistent with RL theory.
> The FinEvo population SDE is a principled extension of the Replicator–Mutator equation
> The reviewer asks why this specific form is chosen.We emphasize that it is derived, not assumed.**
>
> Our derivation builds upon the classical Replicator–Mutator equation from evolutionary game theory
>  (Hofbauer & Sigmund, Evolutionary Games and Population Dynamics, 1998;
>  Nowak, Evolutionary Dynamics: Exploring the Equations of Life, 2006).
>
> This framework decomposes evolutionary change into:
>
> - (a) Selection (Replicator term)
>
> $\beta\ x_k (V_k - \bar{V})$
> capturing fitness-proportional growth.
>
> - (b) Innovation / exploration (Mutator term)
>
> $\mu (m_t - X_t)$ arising from population flow balance and strategy switching.
>
> - (c) Perturbation (Stochastic environmental shocks)
>
> The diffusion term with tangent-space projection $P(X_t)$ ensures $X_t \in \Delta^{K-1}$
> for all t, i.e., population shares remain on the simplex—a geometric requirement, not a heuristic choice.Appendices B and E provide the full derivation of the replicator–mutator dynamics, as well as proofs of simplex invariance and positivity.
>
> ## Summary
> - OU processes capture noisy, bounded, RL-consistent value estimation at the agent level.
> - Replicator–Mutator dynamics provide a mathematically principled macro-level evolution mechanism.
> - **The combined SDE system forms a unified bridge between micro-level learning and macro-level ecological evolution.
> These modeling choices are therefore theoretically grounded, behaviorally justified, and fully consistent with the foundations of evolutionary game theory.**
>
> # R4: How are all the hyperparameters in the models set?
>
> # Q4:
> We thank the reviewer for this question. **We respectfully highlight that comprehensive details regarding all model hyperparameters and configuration settings are documented in Appendix G.6.2 (Agent Parameters) and Appendix F.2 (Agent Library).**
>
> To summarize the key principles for setting these parameters:
>
> 1. Canonical Values from Literature:
>
>   - For Reinforcement Learning (RL) agents (e.g., PPO, A2C), we adopted canonical hyperparameters standard in financial RL literature to ensure fair comparison. For instance, as detailed in Appendix G.6.2, we used a discount factor $\gamma=0.99$, entropy coefficient $0.01$, and learning rate $10^{-3}$
>
>   - For Deep Learning agents (e.g., TimeMixer, Informer), we used standard configurations (e.g., Batch Size=32, Hidden Dim=64) consistent with their original implementations adapted for online rolling training
>
> 2. Robustness over Calibration:
>
>   - For the Evolutionary Dynamics , parameters were chosen within standard ranges of Evolutionary Game Theory (Selection $\beta=2.0$, Innovation $\mu=0.05$)
>
>   - To clarify, the parameters were not chosen to target specific behaviors. Rather, we performed a parameter-sensitivity analysis (Section 3.4 & Figure 9) to confirm that the resulting ecological dynamics are stable under a wide range of selection strengths ($\beta$), innovation rates ($\mu$), and perturbation intensities ($\gamma$). We additionally note that Appendix G.3 includes a robustness test where all agent-level hyperparameters are randomly perturbed by ±20%. The perturbed runs exhibit similar stylized-fact behavior to the baseline, suggesting that the results are not overly sensitive to the initial hyperparameter settings.

---

> ### Author Response · Authors · 2025-11-20
> **To Reviewer 5XkJ**
>
> # Q5: The comparisons with ABIDES and PyMarketSim use "ecological" metrics (entropy, HHI) but not financial performance measures (returns, volatility, or PnL distributions etc). Hence the claim of FinEvo’s superiority is qualitative rather than quantitative.
>
> # R5:
>
> We appreciate the reviewer’s helpful suggestion and have added quantitative financial metrics to strengthen the empirical grounding of the paper.
>  Specifically, we now report:
>
> - Excess Kurtosis & Skewness.
> capturing fat-tail risk and downside asymmetry.
> - Sharpe Ratio Dispersion.
> measuring how strongly the environment separates robust strategies from weak ones.
> - Vol-of-Vol.
> summarizing volatility clustering and regime-shift intensity.
>
> These results, computed over 128 runs under both bull and bear regimes, are presented in the newly added Table 5 in the revised manuscript.
>
> Key findings (based on Table 5):
>
> 1. Realism (Kurtosis & Skewness).
>
>  FinEvo produces pronounced excess kurtosis and strong negative skewness, meaningfully reproducing fat tails and asymmetric risk. ABIDES and PyMarketSim remain close to Gaussian and do not exhibit these characteristics.
> 2. Discriminative Power (Sharpe Dispersion).
>
> **FinEvo shows a much wider dispersion of Sharpe ratios, indicating that the evolutionary dynamics effectively separate strong strategies from fragile ones—providing a more rigorous stress-test environment than static simulators.**
>
> 3. Dynamic Regimes (Vol-of-Vol).
>
> **FinEvo exhibits substantially higher Vol-of-Vol, consistent with the frequent ecological Phase Changes reported in Table 2.
>  This demonstrates endogenous volatility clustering and regime-shift behavior.**
>
> ## Summary.
> **FinEvo generates more realistic distributional properties, stronger strategy differentiation, and richer regime dynamics than existing simulators.
>  We have added Table 5 and the accompanying discussion to the revised manuscript as requested.**
>
> Table 5: Comparison of Financial Metrics across Simulators(Metrics are averaged across 128 independent runs)
>
> | Simulator / Regime     | Excess Kurtosis (mean ± 95% CI) ↑ | Skewness (mean ± 95% CI) ↓ | Sharpe Dispersion (mean ± 95% CI) ↑ | Vol-of-Vol (mean ± 95% CI) ↑ |
> |------------------------|-----------------------------------|-----------------------------|-------------------------------------|-------------------------------|
> | **FinEvo (bull)**      | 3.163 ± 0.138                     | -1.532 ± 0.129              | 1.669 ± 0.061                        | 0.008 ± 0.0004               |
> | **FinEvo (bear)**      | 4.259 ± 0.172                     | -1.793 ± 0.149              | 1.732 ± 0.089                        | 0.013 ± 0.0007               |
> | **ABIDES (bull)**      | 0.315 ± 0.041                     | -0.209 ± 0.036              | 0.675 ± 0.028                        | 0.004 ± 0.0002               |
> | **ABIDES (bear)**      | 0.562 ± 0.053                     | -0.474 ± 0.047              | 0.836 ± 0.033                        | 0.007 ± 0.0003               |
> | **PyMarketSim (bull)** | 0.446 ± 0.044                     | -0.186 ± 0.032              | 0.512 ± 0.026                        | 0.003 ± 0.0002               |
> | **PyMarketSim (bear)** | 0.591 ± 0.058                     | -0.438 ± 0.041              | 0.906 ± 0.034                        | 0.005 ± 0.0003               |
>
> # Q6: Question about “phase change”.
>
> # R6:
> We thank the reviewer for raising this important question. To clarify, Phase Change is a formally defined metric in our framework, and we have added its full mathematical definition in Appendix G.6.3.
>
> 1. Mathematical Definition of Phase Change
>
> Let $X_t = (x_{1,t}, \ldots, x_{K,t})$ denote the population distribution over strategies, and define the set of dominant strategies as:
>
> $$\mathcal{D}(t) = \{ k : x_{k,t} = \max_j x_{j,t} \}.$$
>
> We say that a Phase Change occurs at time t if and only if the identity of the dominant strategy set changes:
>
> $$\boxed{ \text{PhaseChange}(t) = 1  \quad \text{iff} \quad \mathcal{D}(t) \neq \mathcal{D}(t^-) }.$$
>
> This definition captures structural transitions in the ecosystem.
>
> 2. Distinction from Volatility Clustering
>
> Phase Change is a structural driver, not a statistical effect.
>  Volatility clustering describes fluctuations in price variance, whereas Phase Change represents a discrete reorganization of dominant strategic populations.
> A Phase Change can cause volatility clustering by altering the dominant order-flow composition, but the two concepts are distinct.
>
> 3. Distinction from Strategy Regime Switching
>
> Regime switching typically refers to changes in the parameters or state of an individual strategy or model.
>  In contrast, Phase Change is a macro-level event: it reflects a shift in the composition of the entire ecosystem’s equilibrium rather than a single agent’s behavior.
> We hope this clarifies the definition and purpose of the Phase Change metric. The full and formal definition is now included in Appendix G.6.3 to remove any ambiguity.

---

> ### Author Response · Authors · 2025-11-20
> **To Reviewer 5XkJ**
>
> # Q7: You present results for just 1 day of real data. This should be extended to show the approach works in general.
>
> # R7:
>
> We appreciate the reviewer’s concern about whether the approach works beyond a single trading day. **To clarify, our real-data experiments are not limited to a single trading day.**
>
> - As shown in Performance Rankings in Bull vs. Bear Markets(Figure 6), our empirical evaluation spans January–October 2022 (Bear) and October 2024–July 2025 (Bull).
>
> - Figure 6 further aggregates ecosystem statistics over these multi-month periods.
>
> We show a single intraday trajectory in the main text because intraday plots provide a clear and compact illustration of strategy population dynamics; due to space constraints, we include only one representative example in the main paper and provide additional intraday trajectories from multiple days in the revised Appendix G.2 to demonstrate robustness.We also plan to release further multi-day intraday results to facilitate deeper inspection by readers.
>
> # Q8:Does it also work across markets/assets?
>
> # R8:
> This is an excellent point regarding the framework's generality. **We designed FinEvo to be intentionally market-agnostic, focusing on the universal principles of competitive strategy evolution.** The framework's generalizability rests on two core components of our design:
>
> 1. Universal Mechanism (CDA):
>
> The market engine itself—a Continuous Double Auction (CDA) with price-time priority—is not specific to equities. This is the standard, foundational mechanism for price discovery across global electronic markets, including futures, FX, and cryptocurrencies.
>
> 2. Universal Dynamics (The SDE):
>
> The FinEvo SDE is derived from Evolutionary Game Theory. It models relative fitness and population flow, which are not dependent on any asset-specific assumptions (like a specific volatility model for equities).
>
> Therefore, FinEvo is not limited to a single asset class. It provides a general formalism that allows researchers to compare ecological structures across markets. This ability to produce different, plausible outcomes from the same SDEs highlights the framework's expressiveness, not its limitation.We have clarified this market-agnostic design in the revised manuscript to emphasize the broad applicability of our formalism.
>
> **To directly address the reviewer’s concern, we added a small cross-asset robustness study in Appendix G.5 (Table 10).**
> Using the same FinEvo ecosystem (identical evolutionary parameters, agent library, and CDA mechanism), we evaluate four commonly studied assets—AAPL, TSLA, MSFT, and XOM—by only replacing their empirical volatility scales and news arrival patterns.
>
> Across all assets, the core ecological signatures remain consistent (positive excess kurtosis, negative skewness, elevated vol-of-vol, and Sharpe dispersion), while the magnitudes differ in expected, economically intuitive ways (e.g., TSLA showing stronger heavy tails). These results indicate that FinEvo functions robustly across heterogeneous markets without asset-specific tuning.
>
> Table 10: Cross-asset robustness of FinEvo. Metrics are reported as mean ± 95% CI over 128 runs.
>
> | Asset | Excess Kurtosis  ↑ | Skewness  ↓ | Sharpe Dispersion ↑ | Vol-of-Vol ↑ |
> |-------|-----------------|----------|-------------------|------------|
> | **AAPL** | 4.354 ± 0.41  | -1.656 ± 0.18 | 1.633 ± 0.12 | 0.011 ± 0.0015 |
> | **TSLA** | 5.578 ± 0.53  | -2.344 ± 0.24 | 1.975 ± 0.18 | 0.015 ± 0.0023 |
> | **MSFT** | 3.176 ± 0.35  | -0.967 ± 0.15 | 1.158 ± 0.09 | 0.008 ± 0.0016 |
> | **XOM**  | 3.741 ± 0.38  | -0.573 ± 0.12 | 1.392 ± 0.17 | 0.008 ± 0.0011 |

---

> ### Author Response · Authors · 2025-11-20
> **To Reviewer 5XkJ**
>
> # Q9:While the paper includes “real-world news” experiments, the price series and market dynamics are synthetic. The mapping from news sentiment to returns lacks empirical grounding - is there a deeper economic understanding? The conclusions about LLM agents’ dominance could therefore be effects of simulation assumptions.
>
> # R9:
> We thank the reviewer for raising this important point. **We would first like to clarify that FinEvo does not assume, nor structurally enforce, the dominance of LLM-based agents, and this dominance does not arise from the sentiment–return mapping.** As shown in Figure 6, LLM agents are not uniformly superior: in bullish regimes the Informer agent outperforms FinCon and Trading, while in bearish regimes DQN achieves higher returns than Trading and CoT. This suggests that relative performance is shaped by the evolving ecology rather than by a built-in preference for any particular model class.
>
> Second, the mapping from news sentiment to returns is grounded in well-established results in microstructure and behavioral finance, rather than chosen heuristically. Information shocks affect order flow and prices in the spirit of Kyle (1985); behavioral models show that sentiment can drive short-horizon return predictability (Barberis et al., 1998); and empirical studies document that major news increases volatility and jump intensity (Engle & Ng, 1993; Andersen et al., 2003). Our sentiment shocks shift agents’ perceived value $V_k(t)$ in a way that is consistent with these mechanisms.
>
> Finally, we use synthetic, endogenously generated price dynamics because our goal is to study closed-loop evolution rather than to replay a fixed historical tape. Real historical prices do not reflect feedback from changing strategy populations, whereas the synthetic market in FinEvo allows news to act as an exogenous driver and lets all agents—including relatively simple but standard DL and RL baselines—react within the same evolving ecology.
>
> We appreciate the reviewer’s thoughtful comments on these modeling choices. We have clarified these modeling choices and their economic motivation in the revised Appendix G.8.
>
> ## References
>
> - Kyle, A. S. (1985). Continuous Auctions and Insider Trading. Econometrica, 53(6), 1315-1336. https://www.jstor.org/stable/1913210
> Barberis, Nicholas & Shleifer, Andrei & Vishny, Robert, 1998. A model of investor sentiment, Journal of Financial Economics, Elsevier, vol. 49(3), pages 307-343, September.
>
> - ENGLE, R.F. and NG, V.K. (1993), Measuring and Testing the Impact of News on Volatility. The Journal of Finance, 48: 1749-1778. https://doi.org/10.1111/j.1540-6261.1993.tb05127.x
>
> - Andersen, Torben, G., Tim Bollerslev, Francis X. Diebold, and Clara Vega. 2003. "Micro Effects of Macro Announcements: Real-Time Price Discovery in Foreign Exchange ." American Economic Review 93 (1): 38–62.DOI: 10.1257/000282803321455151
>
> # Q10: You mention scientific investigation of markets, not building trading algorithms. What other investigation of market dynamics is there if not to build trading models?
>
> # R10:
> We thank the reviewer for this important question. Our intention in emphasizing “scientific investigation” is to distinguish FinEvo from frameworks whose primary goal is to optimize a single trading policy. **FinEvo is instead designed to study system-level market phenomena that are difficult to access by training one agent in a replayed historical environment—for example, how the market behaves when a new class of strategies (such as LLM-based agents) grows in population share, whether strategy crowding can induce instability or crash-like dynamics, and how selection, innovation, and perturbation jointly shape long-run stability and diversity.**
>
> Beyond profitability, FinEvo also supports questions that are closer to the interests of regulators and market designers than to those of an individual trader. Because the framework models the joint evolution of heterogeneous strategy populations, it enables controlled experiments on how transaction taxes or liquidity constraints affect different agent species, how innovation responds to new regulatory rules, and how alternative auction or matching mechanisms influence ecosystem concentration, resilience, and coexistence among agent classes. In this sense, FinEvo is intended as a policy and market-design laboratory rather than a vehicle for building a single profitable trading algorithm.
>
> **Thus, in our context, “scientific investigation of markets” refers to the study of emergent, adaptive, and systemic properties of market ecosystems—a long-standing topic in market microstructure and financial economics, and one that goes beyond the construction of individual trading models.** We have clarified this distinction more explicitly in Appendix H of the revised manuscript.

---

> ### Author Response · Authors · 2025-11-20
> **To Reviewer 5XkJ**
>
> # Q11: The financial realism remains limited: The market clearing and execution mechanisms are simplified. No empirical calibration or validation against actual historical price series beyond qualitative correspondence. Economic feedbacks such as liquidity constraints, transaction costs, or regulatory requirements are minimal. As a result, FinEvo is a conceptual testbed but not yet a predictive or policy-relevant model. How could you extend the approach to offer more realism?
>
> # R11:
>
> We thank the reviewer for this constructive and forward-looking question.
>
> We agree that pushing empirical realism further is an important next step. **At the same time, we would like to clarify that FinEvo already goes beyond a purely conceptual toy market. The current framework includes a price–time–priority CDA with order-book queues, slippage and transaction costs, and we empirically validate the generated time series against real markets via stylized facts and cross-asset robustness (fat tails, volatility clustering, Sharpe dispersion, Vol-of-Vol, and AAPL/TSLA/MSFT/XOM comparisons in Sec. 3.5, App. G.5–G.6).** Thus, FinEvo is not a purely theoretical construct without empirical anchoring.
>
> Looking ahead, FinEvo naturally admits several extensions:
>
>  - (1) empirical microstructure calibration using L2-level statistics (e.g., depth distributions, arrival rates, queueing);
>
> - (2) hybrid replay of historical LOB data combined with endogenous evolution;
>  - (3) richer economic feedback mechanisms, including liquidity constraints and policy modules; and
>  - (4) a higher-fidelity interactive environment for RL and LLM agents.
>
> ### Scientific Value.
>
> From a scientific perspective, increasing realism is not only engineering; it enables questions such as when strategy crowding creates ecological fragility, how shocks propagate through heterogeneous ecosystems, and how specific policy changes reshape long-run population equilibria. We now briefly discuss these directions as Future Work in Appendix I. Our goal is to position FinEvo as a realistic but still tractable market-ecology testbed that can be progressively calibrated towards policy-relevant applications, rather than a predictive point-forecasting model.

---

> > ### Comment · Reviewer_5XkJ · 2025-11-25
> >
> > thank you for taking the time to respond fully to all the points raised. I appreciate you making clearer the niche your proposed framework fills. Whilst I do still have reservations, I will be raising my initial score on the basis of the detailed responses.

---

> ### Author Response · Authors · 2025-11-28
> **Response to Reviewer 5XkJ’s follow-up comment**
>
> Dear Reviewer 5XkJ,
>
> Thank you very much for taking the time to revisit our submission and for your thoughtful follow-up comment. We are grateful that our responses helped clarify the niche and goals of FinEvo, and we sincerely appreciate your willingness to raise the initial score on this basis.

---

### Official Review · Reviewer_Fxy6 · 2025-11-03

**Soundness:** 3
**Presentation:** 3
**Contribution:** 3
**Rating:** 6
**Confidence:** 4

**Summary:**

This study presents a novel examination of the interactive and evolutionary mechanisms among strategies in financial markets. By proposing a multi-agent financial strategy evolution framework termed FinEvo, this paper investigates how populations of heterogeneous agents employing diverse strategies interact and engage in game-theoretic dynamics. Furthermore, the evolutionary dynamics of the population are decomposed into three primary components: selection, innovation, and environmental perturbation.

**Strengths:**

1. Significance of the Contribution: This paper offers a significant improvement over traditional, isolated strategy backtesting by proposing a comprehensive evolutionary framework. This framework is designed to study strategy interaction and evolution across diverse market environments.

2. Rigorous Theoretical Framework: The paper introduces a set of strategy evolution equations built upon a rigorous mathematical foundation. These equations decompose market dynamics into three interpretable components—selection, innovation, and perturbation—thereby enhancing the model's interpretability.

3. Comprehensive Experimental Design:
 - a. Heterogeneity: The study incorporates four distinct agent types (rule-based, Deep Learning, Reinforcement Learning, and LLM),
reflecting the realistic composition of modern markets.
- b. Multiple Scenarios: The model is tested across a variety of conditions, including artificial shocks, real-world news events, and different market cycles (bull/bear), which provides comprehensive evidence of its robustness.
- c. Ablation Study: An ablation study targeting the three core mechanisms (selection, innovation, and perturbation) compellingly demonstrates that all three are indispensable for maintaining a realistic and adaptive ecosystem

**Weaknesses:**

1. Limited Scope of the Evolutionary Framework: While the framework is presented with rigorous definitions, its evolutionary mechanism is predominantly payoff-driven. This approach overlooks micro-level strategy propagation dynamics (such as diffusion through adjacent nodes). Furthermore, the study confines its analysis to competition among a fixed set of predefined strategies, rather than employing methods like genetic algorithms to evolve new, and potentially superior, hybrid strategies.

2. Opaque Initial Parameter Design: The paper lacks a detailed justification for the initial parameter settings of the agents (e.g., learning rates, information window lengths). This omission is critical because, given a high sensitivity to initial conditions, the agents' subsequent performance could be significantly impacted by these choices. This influence may introduce experimental artifacts rather than reflecting the intrinsic dynamics the study aims to capture.

3. Potential Data Contamination for the LLM: The LLM's training window potentially overlaps with the experimental time period. The study's market data includes the bear market phase from January to October 2022. The training corpus of GPT-4o-mini likely contains data from this period, which could provide it with prior information unavailable to the other agents. This
data

**Questions:**

See the weakness

---

> ### Author Response · Authors · 2025-11-20
> **To Reviewer Fxy6**
>
> We sincerely thank the reviewer for their positive and constructive assessment of FinEvo, in particular for highlighting the significance of moving beyond isolated backtests and the rigor of our evolutionary framework. We address the remaining concerns point by point below.
>
> # Q1: Limited Scope of the Evolutionary Framework: While the framework is presented with rigorous definitions, its evolutionary mechanism is predominantly payoff-driven. This approach overlooks micro-level strategy propagation dynamics (such as diffusion through adjacent nodes).
>
> # R1:
> We thank the reviewer for pointing out the distinction between payoff-driven evolution and micro-level propagation dynamics. To avoid any possible confusion, we would like to clarify our modeling choice.
> **FinEvo does include propagation and diffusion mechanisms, but at the population level, consistent with the canonical mean-field formulation of evolutionary game theory (Hofbauer & Sigmund, 1998).** These mechanisms take the form of:
>
> 1.Innovation / exploration term
> $$\mu(m_t - X_t)$$ which induces strategy switching, population mixing, and non-payoff-driven diffusion.
>  This corresponds to global imitation / social contagion in mean-field evolutionary systems.
>
> 2.Environmental perturbation term
> $$\gamma P(X_t) dW_t$$
> which propagates shocks across the entire ecology through endogenous feedback loops.
>  This produces systemic contagion and herding-like effects without requiring a predefined network topology.
> Thus, FinEvo implements mean-field diffusion, not network diffusion.
>  The difference lies in the modeling paradigm rather than a missing component.
> Network-based contagion (local imitation on a graph) is a valid alternative line of research, and our framework can be extended in that direction. To acknowledge this constructive suggestion, we have added this clarification as a future extension in Appendix I (Future Work).
>
> References
>
> Hofbauer, J., & Sigmund, K. (1998). Evolutionary games and population dynamics. Cambridge University Press.
>
> # Q2:  Furthermore, the study confines its analysis to competition among a fixed set of predefined strategies, rather than employing methods like genetic algorithms to evolve new, and potentially superior, hybrid strategies.
>
> # R2:
> We sincerely thank the reviewer for this insightful perspective.
> We agree that genetic strategy evolution—where strategies mutate, recombine, or generate new hybrids—is an important and promising direction. Our work, however, adopts a different but complementary modeling objective.
> 1. Focus of this paper: population-level ecological dynamics
> **FinEvo is designed to study how populations of heterogeneous strategies evolve under market selection pressure, rather than to evolve the internal structure of the strategies themselves.
>  This follows the classical tradition of evolutionary game theory, where the strategy set is fixed and the dynamics concern how population shares adapt over time.**
> Our goal in this first paper is to establish a rigorous and interpretable ecological formalism, and to understand how real-world strategy families (LLM, RL, rule-based) interact and reshape the market environment.
> 2. Why we did not evolve strategies themselves in this version
> Allowing strategies to mutate or hybridize via genetic mechanisms would introduce an open-ended and rapidly expanding strategy space. While valuable, such an extension would:
> - make the evolutionary process substantially harder to analyze,
> - obscure the effects of the ecological forces introduced in our SDE (selection, innovation, perturbation), and
> - compromise the interpretability of the mathematical formalism, which is a key contribution of this work.
> For these reasons, we intentionally use a fixed but diverse strategy set to isolate ecological interactions in a controlled and interpretable manner.
> 3. Genetic evolution as a natural extension (now included in Appendix I)
> We fully agree that enabling strategies to evolve—via genetic algorithms, neuroevolution, or hybridization—is a compelling future direction.FinEvo’s modular evolutionary formalism already provides the mathematical foundation to incorporate such mechanisms.
> To make this clear, we have added a dedicated discussion of open-ended strategy evolution in the Future Work section of Appendix I in the revised manuscript.

---

> ### Author Response · Authors · 2025-11-20
> **To Reviewer Fxy6**
>
> # Q3:Opaque Initial Parameter Design.
>
> # R3:
> We thank the reviewer for raising this important point on robustness. We agree that clarifying the role of initial agent parameters is essential to separating intrinsic evolutionary dynamics from possible experimental artifacts.
>
> 1. Clarification on parameter choices.
>
>  **We would like to clarify that the initial parameters of all agents are documented in Appendix G.6.2 (Agent Parameters) and Appendix F.2 (Agent Library).** These configurations are not arbitrary:
> - RL agents (DQN, PPO): we use standard hyperparameters (learning rates, discount factors, etc.) widely adopted in financial RL settings;
> - Rule-based agents (RSI, Momentum): we follow canonical technical-analysis conventions (e.g., 14-period RSI, 5/20 moving-average crossover);
> - These choices establish a fair, literature-consistent baseline for cross-agent comparison.
> Perturbed-parameter robustness test.
>
> **To directly address the reviewer’s concern, we have added a new robustness experiment to the revised manuscript in Appendix G.3.** In this experiment, all key agent-level hyperparameters are independently perturbed within ±20% of their baseline values (uniform sampling), creating substantially heterogeneous initializations for each of the 128 Monte-Carlo runs.
>
> 2. Analysis and conclusion.
>
>  As reported in Table 8 below, the perturbed-parameter setting yields results that remain fully consistent with the standard configuration:
> - excess kurtosis remains ≈3.1–4.2,
> - skewness remains strongly negative,
> - Sharpe dispersion stays around ≈1.7,
> - Vol-of-Vol remains stable.
> These findings confirm that FinEvo’s core ecological phenomena—fat tails, asymmetry, evolutionary performance dispersion, and volatility clustering—are emergent properties of the evolutionary dynamics, rather than artifacts of specific initial hyperparameter configurations.
>
> Table 8: Robustness under perturbed agent parameters (128 runs).
>
> | Market Regime | Model                | Excess Kurtosis (mean ± 95% CI) ↑ | Skewness (mean ± 95% CI) ↓ | Sharpe Dispersion (mean ± 95% CI) ↑ | Vol-of-Vol (mean ± 95% CI) ↑ |
> |----------------|----------------------|-----------------------------------|-----------------------------|-------------------------------------|-------------------------------|
> | **Bullish**    | FinEvo               | 3.163 ± 0.138                     | −1.532 ± 0.129              | 1.669 ± 0.061                        | 0.008 ± 0.0004               |
> |                | FinEvo (Perturbed)   | 3.141 ± 0.152                     | −1.596 ± 0.138              | 1.638 ± 0.067                        | 0.008 ± 0.0004               |
> | **Bearish**    | FinEvo               | 4.259 ± 0.172                     | −1.793 ± 0.149              | 1.732 ± 0.089                        | 0.013 ± 0.0007               |
> |                | FinEvo (Perturbed)   | 4.196 ± 0.181                     | −1.692 ± 0.157              | 1.684 ± 0.094                        | 0.012 ± 0.0008               |
>
> # Q4:Potential Data Contamination for the LLM.
>
> # R4:
> We thank the reviewer for highlighting this concern. We agree that large language models pretrained on broad textual corpora may have encountered descriptions of events from 2022. We clarify below why this does not compromise our experimental conclusions.
>
> First, although the LLM may have seen textual narratives about 2022, it has no access to the elements required for any form of time-travel advantage: future price paths, intraday microstructure, order-book evolution, or the endogenous dynamics generated by the interacting agents within FinEvo. The model’s decisions in our environment are based only on short-horizon sentiment cues and the synthetic market state produced internally by the simulation.
>
> Second, our objective is not absolute forecasting but ecological analysis. The phenomena we study—strategy competition, adaptation, evolutionary stability, and regime-dependent population turnover—are all driven by relative performance within the closed FinEvo ecosystem. These dynamics emerge endogenously and do not rely on any external world knowledge that the LLM might have acquired from pretraining corpora.
>
> Third, including 2022 is methodologically necessary. To study evolutionary behavior in both bull and bear regimes, we require at least one downturn. The only time window very likely to exclude any textual overlap (2025) is entirely bullish, and would not allow us to examine phase transitions, volatility clustering, or stress responses. We therefore include 2022 while clearly acknowledging the potential overlap.
>
> Finally, even if the LLM has general textual familiarity with that period, it does not materially influence the outcomes we measure. The selection mechanism, population dynamics, emergent ecological patterns, and phase transitions all arise from FinEvo’s endogenous evolutionary system. We have added a brief clarification of this modeling consideration to the Limitations section (Appendix J).

---

### Author Response · Authors · 2025-11-27
**Summary of Contributions and Post-Rebuttal Revisions for AC**

We sincerely thank all reviewers and the AC for the time and effort dedicated to evaluating our manuscript. We are encouraged that reviewers viewed FinEvo as a *''theoretically grounded ecological framework that advances beyond isolated backtests''*, with a *''principled evolutionary SDE''* and *''comprehensive experiments with heterogeneous agents and realistic market scenarios''*.

### For the area chair’s convenience, we briefly summarize **(i) the core contributions of FinEvo** and **(ii) the main revisions made after the rebuttal (all changes have been highlighted in the revised PDF)**.

## Core contributions of FinEvo

- **C1. Bridging AI trading agents and market ecology (beyond isolated backtests)**

FinEvo moves beyond the single-agent backtest paradigm and provides a **unified ecological game-theoretic formalism** for financial AI. While platforms such as ABIDES and JAX-LOB emphasize execution fidelity or training a single RL agent on historical order flow, FinEvo explicitly models the endogenous co-evolution of heterogeneous strategy families—rule-based, DL, RL, and LLM agents—within a shared CDA market.

- **C2. A principled evolutionary SDE for heterogeneous markets**

The framework introduces the **FinEvo SDE**, a stochastic differential equation **grounded in evolutionary game theory** that unifies **Selection, Innovation, and Perturbation** in a tractable, simplex-invariant form. It couples an OU-based value learning process (continuous-time analogue of RL-style value updates) with a replicator–mutator–type population SDE, providing a **mathematically explicit and interpretable description of market-ecology dynamics**, with simplex invariance and well-posedness formally proved in the appendices, rather than heuristic ABM rules.

- **C3. A ''financial wind tunnel'' for emergent phenomena**

By combining this evolutionary SDE with a realistic CDA engine and heterogeneous agents, FinEvo acts as a **market-ecology ''wind tunnel''**: it reveals **regime-dependent phase transitions**, dominance cycles, strategy crowding, rich strategy–pair correlations, and **stylized facts** (fat tails, volatility clustering, etc.) that are not captured by static simulators. The same framework is **market-agnostic across assets** and is designed to support future policy and market-design experiments by exchanges and regulators (e.g., liquidity constraints, transaction taxes, etc.).

## Main post-rebuttal revisions

To directly address reviewer concerns raised in the reviews, we made the following revisions.

- **Stylized-fact validation and financial metrics**

Added a “Stylized-Fact Validation” experiment (Sec. 3.5) and **Table 5**, comparing FinEvo with ABIDES/PyMarketSim using excess kurtosis, skewness, Sharpe dispersion, and Vol-of-Vol, with references to standard empirical-finance and ABM literature; we also formalized the Phase Change metric in Appendix G.6.3.

- **Theoretical clarity and robustness to shocks**

Clarified the **theoretical assumptions and SDE design**, including the OU value process and the replicator–mutator–type population SDE, and showed **robustness to correlated, heavy-tailed shocks** via new derivations and experiments (Sec. 2, Appendix D, Appendix G.4, Table 9). These results show that FinEvo’s main ecological phenomena do not rely on independence/Normality assumptions.

- **Robustness to agent parameters and market size**

Added **robustness experiments** on agent parameters and market size (Appendix F.2, G.3, G.6.2; Tables 4 and 8), confirming that FinEvo’s heavy tails, negative skewness, Sharpe dispersion, and Vol-of-Vol are **stable under ±20% hyperparameter perturbations and varying N**.

- **Clarified simulation goals and scenario design**

Clarified the goals of the Artificial Shocks and Real-World News scenarios (including bull vs. bear regimes) and emphasized FinEvo’s intended role as a **general framework for evolutionary market dynamics and policy / market-design experiments** (Appendix G.7–G.9, H, I).

- **Information realism, economic grounding, and positioning**

Clarified agents’ **rich information sets** (order-book depth, volume, news signals, etc.; Appendix G.6.1, G.6.3), added a **cross-asset robustness experiment** (AAPL/TSLA/MSFT/XOM; Appendix G.5, Table 10), discussed the **economic grounding of news shocks** and limitations of LLM-based agents (Appendix G.8, J), and expanded the Related Work to clarify how FinEvo differs from and complements JAX-LOB, Belcak et al., and Mahdavi (HFTE).

Taken together, these additions provide a **comprehensive empirical validation** of FinEvo, covering stylized facts, sensitivity to shocks and parameters, cross-asset robustness, and the realism and economic grounding of information and news inputs. We believe these revisions have substantially **addressed the raised concerns** and clarified the scope, realism, and positioning of FinEvo as a **principled evolutionary framework for market ecology**.

---

### Meta-Review · Area_Chair_1hM4 · 2026-01-16

**Summary:**

The paper introduces FinEvo, an ecological game formalism designed to evaluate financial trading strategies by modeling them as adaptive agents that interact and evolve within a shared market. The framework incorporates heterogeneous agents—ranging from rule-based and deep learning models to large language model (LLM) traders—whose population dynamics are governed by evolutionary mechanisms of selection, innovation, and environmental perturbation. Experiments demonstrate that this approach captures complex emergent behaviors, such as strategy dominance and coalition formation, offering a more robust method for analyzing market adaptation and systemic risks.

**Reviewer Concerns:**

There were few concerns regarding the paper's reliance on arbitrary, unvalidated modeling assumptions and synthetic scenarios which created doubts on the empirical validity and economic relevance. Additionally, the work was noted for lacking clear motivation and robust baselines against realistic financial metrics or alternative methods. As a result the paper failed to convincingly demonstrate the practical utility of its formalism and simulations.

**Reviewer Scores:**

There were two slightly positive reviewers, one slightly negative and one negative reviewer. It seems that during the rebuttal, the authors put a lot of effort to address the reviewers comments and also that there would have been an increase in the score of the negative reviewer. However, the AC does not feel comfortable to accept this paper as there was no reviewer that was very enthusiastic, or the final average score would not be of borderline/weak accept. We believe that the paper needs one more iteration before getting accepted and recommend rejection.

---

### Decision · Program_Chairs · 2026-01-26

Reject